# A Multi-Power Law for Loss Curve Prediction Across Learning Rate Schedules

**Kairong Luo**[1]   **Haodong Wen**[2]   **Shengding Hu**[1]   **Zhenbo Sun**[1]
**Zhiyuan Liu**[1]   **Maosong Sun**[1†]   **Kaifeng Lyu**[3†]   **Wenguang Chen**[1,4†]

[1]Department of Computer Science and Technology, Tsinghua University
[2]Qian Xuesen College, Xi'an Jiaotong University
[3]Simons Institute, University of California, Berkeley
[4]Peng Cheng Laboratory

```
{luokr24,sunzb20}@mails.tsinghua.edu.cn
{herrywenh,shengdinghu}@gmail.com
kaifenglyu@berkeley.edu
{liuzy,sms,cwg}@tsinghua.edu.cn
```

## Abstract

Training large models is both resource-intensive and time-consuming, making it crucial to understand the quantitative relationship between model performance and hyperparameters. In this paper, we present an empirical law that describes how the pretraining loss of large language models evolves under different learning rate schedules, such as constant, cosine, and step decay schedules. Our proposed law takes a multi-power form, combining a power law based on the sum of learning rates and additional power laws to account for a loss reduction effect induced by learning rate decay. We extensively validate this law on various model sizes and architectures, and demonstrate that after fitting on a few learning rate schedules, the law accurately predicts the loss curves for unseen schedules of different shapes and horizons. Moreover, by minimizing the predicted final pretraining loss across learning rate schedules, we are able to find a schedule that outperforms the widely used cosine learning rate schedule. Interestingly, this automatically discovered schedule bears some resemblance to the recently proposed Warmup-Stable-Decay (WSD) schedule (Hu et al., 2024) but achieves a slightly lower final loss. We believe these results could offer valuable insights for understanding the dynamics of pretraining and designing learning rate schedules to improve efficiency.[1]

## 1 Introduction

Large Language Models (LLMs) can achieve strong performance if pretrained with an appropriate configuration of hyperparameters, such as model width, depth, number of training steps, and learning rate. However, tuning these hyperparameters at scale is extremely costly since one pretraining run can take weeks or even months.

To reduce the cost of hyperparameter tuning, various scaling laws have been proposed to predict pretraining loss or downstream performance by capturing empirical relationships between key hyperparameters and model performance. A notable example is the Chinchilla scaling law (Hoffmann et al., 2022), which approximates the final pretraining loss as a simple function of the model size $N$ and total training steps $T$ (or total training tokens), $\mathcal{L}(N, T) = L_0 + A \cdot T^{-\alpha} + B \cdot N^{-\beta}$. By fitting parameters $L_0, A, B, \alpha, \beta$ from a few training runs with varying $N$ and $T$, one can use the formula to infer the optimal choice of $N$ and $T$ given a fixed compute budget $C \propto NT$.

A key challenge that existing scaling laws have not addressed is how to set the **Learning Rate (LR)** optimally over time. LR is arguably the most critical hyperparameter in optimization, as it can significantly affect the training speed and stability. A large LR can quickly reduce the training loss, but in the long term, it may cause overshooting and oscillation along sharp directions on the loss

---

[†]Corresponding authors.
[1]Code Implementation: https://github.com/thu-yao-01-luo/MultiPowerLaw

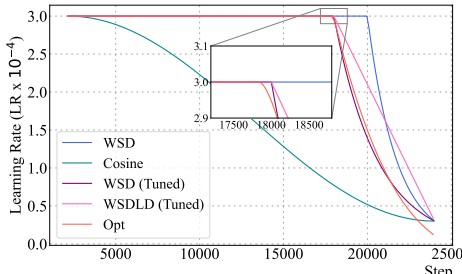 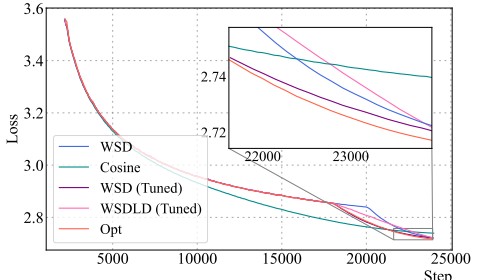

(a) Comparison of Optimized LR Schedules  (b) Loss Curves for Different LR Schedules

Figure 1: Optimizing the LR schedule induces a schedule (Opt) better than cosine and WSD schedules. We conduct evaluation experiments on a 400M Llama-2 (Touvron et al., 2023) model trained over 12B tokens. Zoom-in regions facilitate the readers who are interested in the local details. **(a)** Our optimized schedule comprises constant and decay stages post-warmup, aligning with WSD (Hu et al., 2024). **(b)** Loss curves demonstrate that our optimized schedule outperforms cosine schedules and two major variants of WSD with tuned hyperparameters (WSD with exponential decay and WSDLD with linear decay).

landscape. In contrast, a small LR ensures a more stable training process but also slows down the convergence. Practitioners often balance these trade-offs by starting training with a large LR and then gradually reducing it over time, following a *Learning Rate schedule* (LR schedule) (Bengio, 2012). These LR schedules sometimes include a warmup phase at the beginning, where the LR linearly increases from zero to a large value over a few thousand steps, and only after this warmup phase does the LR start to decay. The most commonly used LR schedule in LLM pretraining is the cosine schedule (Loshchilov & Hutter, 2017), which decays the LR following a cosine curve. Other schedules include the cyclic (Smith, 2017), Noam (Vaswani et al., 2017), and Warmup-Stable-Decay (WSD) schedules (Hu et al., 2024), but there is no consensus on the optimal choice.

Existing scaling laws sidestep the complexity of LR schedules by fitting parameters on a fixed family of LR schedules. For instance, Hoffmann et al. (2022) fitted the parameters in the Chinchilla scaling law for training runs that have gone through the entire cosine LR schedule. As a result, it does not generalize well to other LR schedules, or even to the same schedule with early stopping. Moreover, existing scaling laws lack a term to account for LR schedules, limiting their ability to provide practical guidance on setting the LRs. This issue can become even more pronounced when scaling up training to trillions of tokens (Dubey et al., 2024; Liu et al., 2024), where the extreme cost of training makes it impractical to experiment with multiple LR schedules.

In this paper, we aim to quantify how LR schedules influence the evolution of training loss in LLM pretraining through empirical analysis. More specifically, we study the following problem, which we call the *schedule-aware loss curve prediction* problem: *Can we use a simple formula to accurately predict the training loss curve $\mathcal{L}(t)$ ($1 \leq t \leq T$) given a LR schedule $E := \{\eta_1, \eta_2, \ldots, \eta_T\}$ for $T$ steps of training?* To align with standard practices in LLM pretraining and to enable a more precise analysis tailored to this setting, we impose the following reasonable restrictions on the problem. First, we take fresh samples from a data stream at each training step, so there is no generalization gap between the training and test loss. Second, we focus on LR schedules that decay the LR over time, i.e., $\eta_1 \geq \eta_2 \geq \eta_3 \geq \cdots$. Finally, as most LR schedules used in practice start with a warmup phase before the LR decays, we make a minor modification to the problem and include a fixed warmup phase before the decay phase we are interested in. We assume that the shape and the peak LR $\eta_{\max}$ of the warmup phase have been carefully picked, potentially through a series of short training runs, and we are only interested in understanding how different LR decay schedules after warmup affect the training loss curve. For convenience, we shift the time index so that $t = 1$ corresponds to the first step after the warmup phase.

In contrast to most existing scaling laws that rely on only two or three hyperparameters (Kaplan et al., 2020; Hoffmann et al., 2022; Muennighoff et al., 2023; Goyal et al., 2024), solving the above problem poses unique challenges, as it requires predicting the loss curve based on the entire LR schedule, which is inherently high-dimensional. This complexity necessitates a more sophisticated approach to understand and quantify the relationship between the LR schedule and the loss curve.

**Our Contribution: Multi-Power Law.**  In this paper, we propose the following empirical law (1) for schedule-aware loss curve prediction:

$$\mathcal{L}(t) = L_0 + A \cdot (S_1(t) + S_W)^{-\alpha} - \mathrm{LD}(t), \quad \text{where} \quad S_1(t) := \sum_{\tau=1}^{t} \eta_\tau. \qquad (1)$$

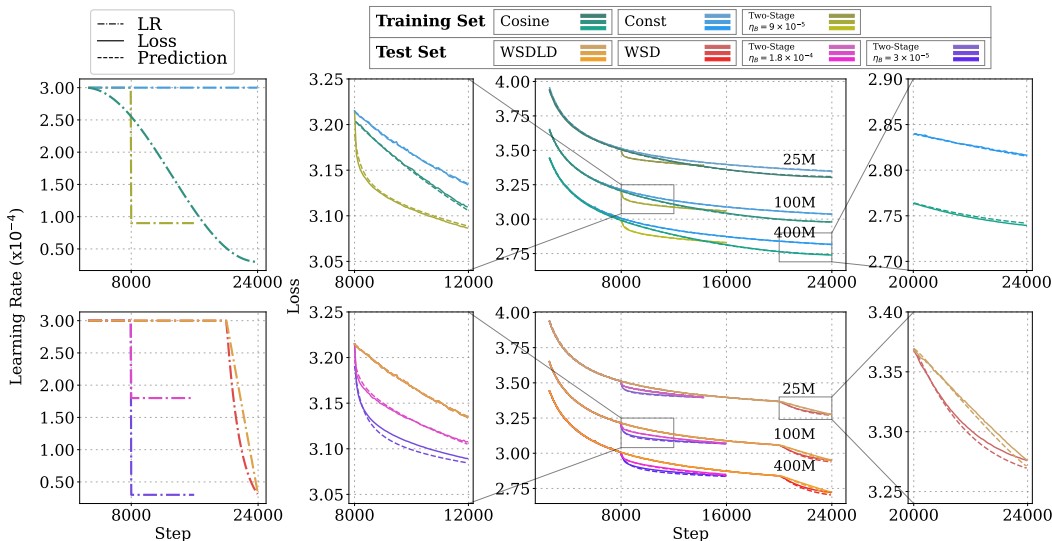

Figure 2: The Multi-Power Law (MPL) with parameters fitted on cosine, constant, and two-stage schedules can accurately predict the loss curves of unseen schedules, including WSDLD, WSD, and two-stage schedules with a different LR in the second stage. See Table 1 for evaluation metrics.

Here, $S_W$ denotes the sum of learning rates used in the warmup phase. The first two terms $L_0 + A \cdot (S_1(t) + S_W)^{-\alpha}$ can be viewed as an extension of the Chinchilla scaling law by replacing the number of steps $T$ with the cumulative sum of learning rates up to step $t$, while neglecting the dependence on the model size. While this alone provides a crude approximation of the loss curve by linearizing the contribution of the LR at each step (see Section 3.1 for further discussion), it does not account for the specific shape of the LR decay. The additional term $\text{LD}(t)$ serves as a correction term, which captures the effect of LR decay in further reducing the loss:

$$\text{LD}(t) := B \sum_{k=1}^{t} (\eta_{k-1} - \eta_k) \cdot G(\eta_k^{-\gamma} S_k(t)), \quad S_k(t) := \sum_{\tau=k}^{t} \eta_\tau, \quad G(x) := 1 - (Cx+1)^{-\beta}. \quad (2)$$

More specifically, $\text{LD}(t)$ is linear with a cumulative sum of the LR reductions $\eta_{k-1} - \eta_k$ over time, scaled by a nonlinear factor $G(\eta_k^{-\gamma} S_k(t))$. This factor gradually saturates to a constant as the training progresses, which follows a power law in a scaled sum of learning rates $\eta_k^{-\gamma} S_k(t)$.

We call this law of $\mathcal{L}(t)$ the *Multi-Power Scaling Law (MPL)* as it consists of multiple power-law forms. $L_0, A, B, C, \alpha, \beta, \gamma$ are the parameters of the law and can be fitted by running very few pretraining experiments with different LR schedules. Our main contributions are as follows:

1. We propose the Multi-Power Law (1) for schedule-aware loss curve prediction, and empirically validate that after fitting the parameters of the law on at most 3 pretraining runs, it can predict the loss curve for unseen LR schedules with remarkable accuracy (see Figure 2). Unlike the Chinchilla scaling law, which relies solely on the final loss of each training run to fit its parameters, our approach utilizes the entire loss curve of each training run to fit the parameters, thus significantly reducing the number of training runs and compute resources needed for accurate predictions (Figure 5). Extensive experiments are presented for various model architectures, sizes, and training horizons (Section 4).

2. Our Multi-Power Law is accurate enough to be used to search for better LR schedules. We show that by minimizing the predicted final loss according to the law, we can obtain an optimized LR schedule that outperforms the standard cosine schedule. Interestingly, the optimized schedule has a similar shape as the recently proposed WSD schedule (Hu et al., 2024), but its shape is optimized so well that it outperforms WSD with grid-searched hyperparameters (Section 5).

3. We use a novel "bottom-up" approach to empirically derive the Multi-Power Law. Starting from two-stage schedules, we conduct a series of ablation studies on LR schedules with increasing complexity, which has helped us to gain strong insights into the empirical relationship between the LR schedule and the loss curve (Section 3).

4. We provide a theoretical analysis for quadratic loss functions and demonstrate that the Multi-Power Law emerges when the Hessian and noise covariance matrices exhibit certain types of power-law structures (Appendix B).

## 2 PRELIMINARY

**Learning Rate Schedule.** A learning rate (LR) schedule is a sequence $E := \{\eta_1, \ldots, \eta_T\}$ that specifies the LR at each step of the training process. For language model pretraining, the cosine LR schedule (Loshchilov & Hutter, 2017) is the most popular schedule, which can be expressed as $\eta_t = \frac{1+\alpha}{2} \eta_{\max} + \frac{1-\alpha}{2} \eta_{\max} \cos(\frac{\pi t}{T})$. Here, $\eta_{\max}$ is the peak LR and $\alpha$ is usually set to 0.1. The Warmup-Stable-Decay (WSD) schedule (Hu et al., 2024) is a recently proposed LR schedule. This schedule first goes through a warmup phase, then maintains at a stable LR $\eta_{\max}$ with $T_{\text{stable}}$ steps, and finally decays in the form of $f(t - T_{\text{stable}})\eta_{\max}$ for $T_{\text{stable}} \leq t \leq T_{\text{total}}$. Here $f(x) \in (0, 1)$ can be chosen as linear or exponential decay functions. We visualize these two LR schedules in Figure 1(a).

**Warmup Phase.** Many LR schedules, such as WSD, include a warmup phase in which the LR gradually increases from 0 to the peak LR $\eta_{\max}$ over a few thousand steps. We denote the number of warmup steps as $W$. By default, the LR increases linearly, so the total LR sum during warmup is given by $S_W = \frac{1}{2}\eta_{\max}W$. Our analysis focuses on the training process after the warmup, where the LR is decaying in almost all LR schedules. We count training steps starting from the end of warmup and set $t = 1$ as the first step after warmup. Accordingly, $\{\eta_1, \ldots, \eta_T\}$ represents the post-warmup schedule, and the LR at the last warmup step $\eta_0 = \eta_{\max}$ is the peak LR of the entire schedule.

**Power Law of Data Scaling** Prior studies (Hoffmann et al., 2022; Kaplan et al., 2020) demonstrate that, for a fixed model size, the final loss follows a power law of the data size or, equivalently, the total training step number $T$ in a constant-batch-size setting. This relationship is expressed as:

$$\mathcal{L}(T) \approx \hat{\mathcal{L}}(T) := L_0 + \tilde{A} \cdot T^{-\alpha}, \tag{3}$$

where $L_0, \tilde{A}, \alpha$ are parameters to fit. This law is typically fitted over the final losses of a set of training curves generated from a specific LR schedule family, such as a cosine schedule with a given peak LR ($\eta_{\max}$), ending LR ($\alpha\eta_{\max}$) and warmup steps ($W$). However, applying (3) directly to intermediate steps ($t < T$) introduces bias, as the LR schedule up to $t$ bears insufficient decay compared to the full schedule over $T$, resulting in different loss trajectories. This discrepancy is confirmed in Figure 5(b). We refer to (3) as the Chinchilla Data Scaling Law (abbreviated as CDSL) throughout the paper since it is simplified from the Chinchilla scaling law (Hoffmann et al., 2022) to highlight the data dimension.

## 3 EMPIRICAL DERIVATION OF THE MULTI-POWER LAW

In this section, we present the empirical derivation of the Multi-Power Law (MPL) for schedule-aware loss curve prediction. Our key insights are summarized as follows:

1. If two training runs share the same sum of learning rates, $\sum_{t=1}^{T} \eta_t$, then their final losses tend to be similar, though a non-negligible discrepancy remains (Section 3.1).

2. In particular, for a training run with a given LR schedule, the final loss $\mathcal{L}(T)$ is similar to that of another training run using a constant learning rate schedule with the same total LR sum. This motivates us to decompose $\mathcal{L}(T)$ into two components: (1) the final loss of the corresponding constant LR run; and (2) a residual term that captures the effect of LR decay, defined as the difference between the final loss of the target run and the constant LR run. (Section 3.1)

3. Empirically, we observe that training runs with constant learning rates exhibit a Chinchilla-like power-law behavior in the loss curve and can thus be well approximated by a simple power law. (Section 3.2.1)

4. To approximate the residual term, instead of analyzing it directly, we imagine a sequence of training runs with schedules that gradually transition from a constant LR to the target schedule, all while maintaining the same total LR sum. Using a novel "bottom-up" approach, we derive an approximation formula for the loss difference introduced by each incremental change in the LR schedule, first by analyzing simple two-stage schedules and then extending the results to more complex schedules. (Sections 3.2.2 and 3.3)

Finally, we sum up all the approximation terms above, leading to our MPL. Below, we elaborate on our approach in detail.

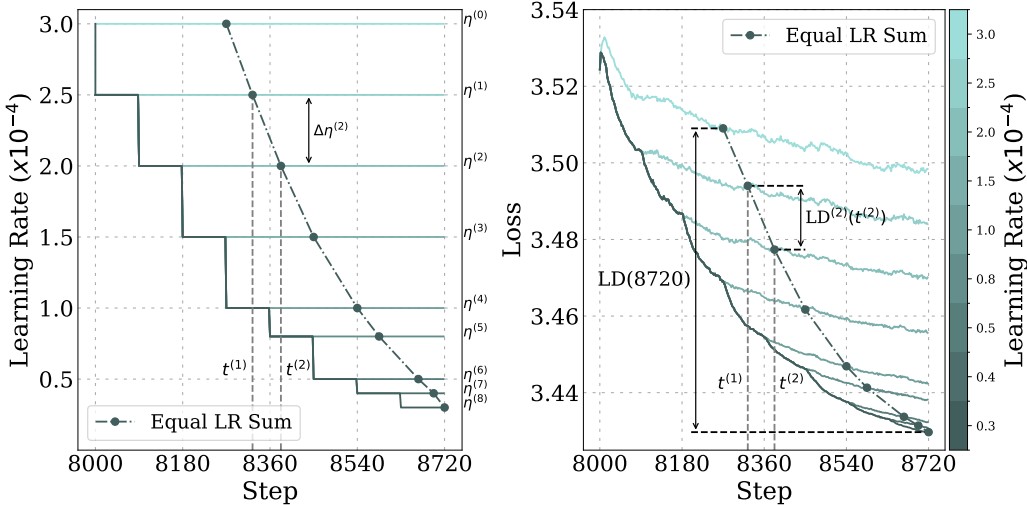

Figure 3: A multi-stage schedule (Appendix A.2) example to illustrate the learning rate (LR) sum matching (Section 3.1) and fine-grained loss reduction decomposition (Section 3.2.2). The steps with equal LR sum as the final step $T_9 = 8720$ are marked and linked with the dash-point line. Each stage spans 90 steps. $T_1 = 8000$, $T_2 = 8090$, $t^{(1)} = Z_{T_2}(T_9)$, $t^{(2)} = Z_{T_3}(T_9)$. See Appendix F.3 for experiment details. **Left:** The actual multi-stage schedule and schedules for auxiliary processes. LR gap between adjacent points denotes the LR reduction $\Delta\eta^{(i)} = \eta^{(i-1)} - \eta^{(i)}$. **Right:** Corresponding training curves for the multi-stage schedule and the auxiliary processes. The total loss reduction is $\mathrm{LD}(T_9)$ and can be decomposed as the intermediate loss reduction sum. The loss gap between adjacent points denotes the stage-wise loss reduction $\mathrm{LD}^{(i)}(t^{(i)})$.

## 3.1 OUR APPROACH: LEARNING RATE SUM MATCHING

**Auxiliary Training Process.** As introduced above, we construct a series of auxiliary training runs with LR schedules gradually changing from a constant LR schedule to the target schedule $E := \{\eta_1, \ldots, \eta_T\}$. Our construction is detailed as follows. We define the $k$-th auxiliary process shares the first $k$ steps of learning rates, $\{\eta_1, \ldots, \eta_k\}$, with the actual training process with LR schedule $E$, and continues with the constant LR $\eta_k$ afterwards. The corresponding loss curve for the $k$-th auxiliary process is denoted as $\mathcal{L}_k(t)$. In particular, the 0-th auxiliary process shares only the warmup phase with the actual training process and uses a constant LR $\eta_0 = \eta_{\max}$ after warmup. We especially call it the *constant process* and use $\mathcal{L}_{\mathrm{const}}(t)$ to represent its loss curve. The $T$-th auxiliary process coincides with the actual training run with the target LR schedule, so $\mathcal{L}_T(t) = \mathcal{L}(t)$.

**Learning Rate Sum Matching Decomposition** The Multi-Power Law (MPL) approximates the loss curve $\mathcal{L}(t)$ of the actual training process through the following decomposition. We define $Z(t)$ as the equivalent step in a constant LR process that shares the same cumulative LR sum as the actual process up to step $t$, where $Z(t) = \frac{S(t)}{\eta_0}$ and $S(t) = \sum_{\tau=1}^{t} \eta_\tau$ represents the sum of post-warmup LRs. The loss at step $t$ is then decomposed as:

$$\mathcal{L}(t) = \mathcal{L}_{\mathrm{const}}(Z(t)) - \underbrace{(\mathcal{L}_{\mathrm{const}}(Z(t)) - \mathcal{L}(t))}_{=:\ \mathrm{LD}(t)}, \tag{4}$$

where $\mathcal{L}_{\mathrm{const}}(Z(t))$ interpolates the loss for non-integer $Z(t)$ in the constant LR process. We first approximate $\mathcal{L}(t)$ using the training loss $\mathcal{L}_{\mathrm{const}}(Z(t))$ at step $Z(t)$, and then write the residual term $\mathrm{LD}(t)$ representing the approximation error. We call $\mathrm{LD}(t)$ the **Loss reDuction term**, as it quantifies the loss reduction due to LR decay. We will approximate these two terms by parts in Section 3.2, with $\mathcal{L}_{\mathrm{const}}(Z(t))$ detailed in Section 3.2.1 and $\mathrm{LD}(t)$ in Section 3.2.2.

**Motivation: Continuous Approximations of the Training Dynamics.** The rationale behind this approach is that two training runs with the same LR sum should result in similar training losses, thus making it natural to decompose the loss curve into a major term corresponding to the loss of a run with the same LR sum and a small residual term. To see this, we use SGD as an example. If the learning rates $\eta_1, \ldots, \eta_T$ are small, then SGD can be seen as a first-order approximation of its continuous counterpart, gradient flow, under mild conditions (Li et al., 2017; Cheng et al., 2020; Elkabetz & Cohen, 2021). Here gradient flow describes a continuous-time process in which the parameters $\boldsymbol{\theta}(\tau)$ evolve according to the differential equation $\frac{d\boldsymbol{\theta}(\tau)}{d\tau} = -\nabla\mathcal{L}(\boldsymbol{\theta}(\tau))$, where $\nabla\mathcal{L}(\boldsymbol{\theta})$ is the gradient at $\boldsymbol{\theta}$, and $\tau$ denotes the continuous time. In this approximation, the $t$-th step of SGD

corresponds to evolving $\boldsymbol{\theta}(\tau)$ over a small time interval of length $\eta_t$. When the learning rates are sufficiently small, the parameters after $t$ steps of SGD are close to $\boldsymbol{\theta}(\tau)$ at time $\tau = \sum_{k=1}^{t} \eta_k$. This connection naturally motivates us to compare the losses of two training runs with the same LR sum. While we use SGD for illustration, other optimization methods such as Adam can be similarly approximated by their continuous counterparts (Ma et al., 2022).

## 3.2 APPROXIMATION BY PARTS

### 3.2.1 CONSTANT PROCESS LOSS APPROXIMATION

Motivated by the continuous approximation of the training dynamics, we hypothesize that losses of constant LR processes with identical LR sums are closely aligned. This insight inspires us to represent $\mathcal{L}_{\mathrm{const}}(Z(t))$ as a function of $S(t) + S_W$, where $S(t) + S_W$ represents the cumulative LR sum up to step $t$, including the warmup phase part $S_W$. Analogous to (3), we propose that $\mathcal{L}_{\mathrm{const}}(Z(t))$ follows a power law over the LR sum:

$$\hat{\mathcal{L}}_{\mathrm{const}}(Z(t)) = L_0 + A \cdot (S(t) + S_W)^{-\alpha}, \tag{5}$$

where $A$ is a parameter counterpart of $\tilde{A}$. We perform extensive empirical validation and ablation studies across different model sizes, training horizons, and learning rates to confirm the robustness of (5), as detailed in Appendix F.1 and illustrated in Figure 11.

### 3.2.2 LOSS REDUCTION APPROXIMATION

Now we turn to the loss reduction term $\mathrm{LD}(t)$. We start by proposing a simple yet effective linear approximation as a warmup, then we further break down the term with a finer-grained LR sum matching approach.

**Warmup: A Crude Linear Approximation.** We first generate training loss curves across various LR schedule types, including cosine and WSD schedules, alongside the loss curves of their corresponding constant processes. Then we can compute the loss reduction $\mathrm{LD}(t)$ for different LR schedules and analyze their dependency. As demonstrated in Figure 10, $\mathrm{LD}(t)$ is approximately proportional to the LR reduction, $\Delta\eta_t = \eta_0 - \eta_t$ across different schedules. This leads to the following approximation:

$$\mathrm{LD}(t) \approx B(\eta_0 - \eta_t), \tag{6}$$

where $B$ is a constant. This finding highlights a strong correlation between the loss gap and the LR gap at equivalent LR sum points on the loss landscape. However, while the linear approximation offers insights into the shape of $\mathrm{LD}(t)$, deviations from the actual loss reduction remain. Notably, when the LR decreases abruptly (e.g., in step-wise schedules), it predicts an instant loss drop at the stage switch, whereas the true loss decline remains smoother during the training process. See Appendix C for further discussion.

**Fine-Grained LR Sum Matching Decomposition.** In practice, the loss reduction term $\mathrm{LD}(t)$ can have a more complex dependency on the LR schedule. To provide a more accurate approximation than the linear approximation above, we employ LR sum matching between adjacent auxiliary processes and decompose the loss reduction $\mathrm{LD}(t)$ into a telescoping sum of *intermediate loss reductions* between adjacent auxiliary processes.

More specifically, consider the step $t$ in the actual training process. Similar to $Z(t)$, we define $t_k := Z_k(t)$ as the equal-LR-sum step in the $k$-th auxiliary process, which is given by

$$t_k := Z_k(t) := k - 1 + \frac{1}{\eta_k} S_k(t), \tag{7}$$

where $S_k(t) = \sum_{\tau=k}^{t} \eta_\tau$. Then, for the $k$-th and $(k+1)$-th processes, we define the intermediate loss reduction as:

$$\mathrm{LD}_k(t_{k+1}) := \mathcal{L}_k(t_k) - \mathcal{L}_{k+1}(t_{k+1}). \tag{8}$$

Intuitively, this term compares the loss at step $t_{k+1}$ in the $(k+1)$-th process with the loss at the equal-LR-sum step in a process that stops decaying the LR after the first $k$ steps, i.e., the $k$-th process. We then decompose the loss reduction term as a telescoping sum of intermediate loss reductions:

$$\mathrm{LD}(t) = \mathcal{L}_{\mathrm{const}}(Z(t)) - \mathcal{L}(t) = \mathcal{L}_0(Z_0(t)) - \mathcal{L}_t(Z_t(t)) = \sum_{k=0}^{t-1} \mathrm{LD}_k(t_{k+1}). \tag{9}$$

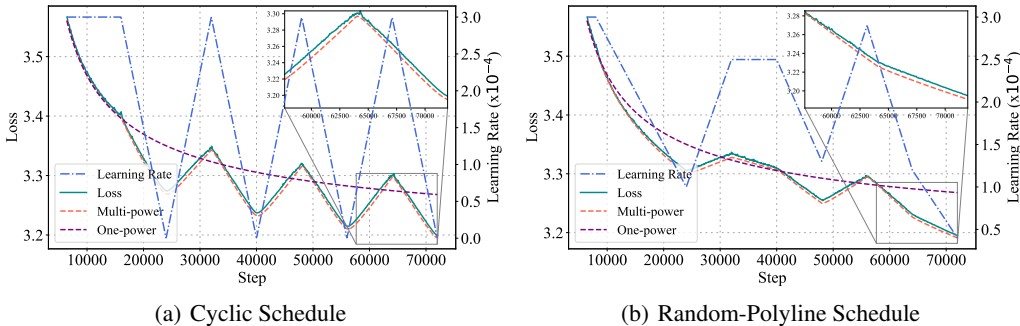

(a) Cyclic Schedule          (b) Random-Polyline Schedule

Figure 4: The examples of long-horizon non-monotonic schedules. The one-power line represents the constant process prediction. **(a)** The cyclic schedule with 72000 steps, where each half-cycle spans 8000 steps, and the first decay begins after 16000 steps. **(b)** The random-polyline schedule, consisting of piecewise linear interpolation between randomly selected intermediate learning rates in the range of $3 \times 10^{-5}$ to $3 \times 10^{-4}$, with LR milestones occurring at intervals of 8000 steps.

By leveraging this fine-grained decomposition, a good estimation of $\mathrm{LD}_k(t_{k+1})$ can lead to a more accurate approximation of $\mathrm{LD}(t)$. Where the context is clear, we simplify notation by omitting subscripts and denoting intermediate loss reduction as $\mathrm{LD}_k(t)$.

### 3.3 BOTTOM-UP DERIVATION: TWO-STAGE, MULTI-STAGE, AND GENERAL SCHEDULES

The challenges in approximating the intermediate loss reduction $\mathrm{LD}_k(t)$ are twofold. First, for commonly used schedules, the learning rate (LR) reduction at intermediate steps is often too small to induce a measurable loss reduction. Second, $\mathrm{LD}_k(t)$ may depend intricately on all previous learning rates $\{\eta_1, \ldots, \eta_k\}$, which we refer to as the *LR prefix* in this section.

To address these issues, we derive the form of $\mathrm{LD}_k(t)$ using a "bottom-up" approach regarding schedule structures. First, we propose its form through schedules comprising two constant LR stages, leveraging significant LR reductions. Next, we examine its dependency on the LR prefix using schedules of multiple stages. Our main finding is that $\mathrm{LD}_k(t)$ depends weakly on the LR prefix and can be approximated by the following form:

$$\mathrm{LD}_k(t) \approx \widehat{\mathrm{LD}}_k(t) := B(\eta_k - \eta_{k+1})\left(1 - \left(C\eta_{k+1}^{1-\gamma}(t-k)+1\right)^{-\beta}\right), \tag{10}$$

with LR-prefix independent constants $B$, $C$, $\gamma$ and $\beta$. Due to space constraints, we refer readers to Appendices A.1 and A.2 for detailed derivations of (10).

For general LR schedules, we extrapolate this findings and propose to approximate the total loss reduction term as:

$$\widehat{\mathrm{LD}}(t) := \sum_{k=0}^{t-1}\widehat{\mathrm{LD}}_k(t_{k+1}) = \sum_{k=0}^{t-1}B(\eta_k - \eta_{k+1})\left(1 - \left(C\eta_{k+1}^{1-\gamma}(t_{k+1}-k)+1\right)^{-\beta}\right).$$

By the definition of $t_{k+1}$ (7), we have $t_{k+1} - k = \frac{S_{k+1}(t)}{\eta_{k+1}}$. Therefore, we can conclude

$$\mathrm{LD}(t) \approx \widehat{\mathrm{LD}}(t) = \sum_{k=1}^{t}B(\eta_{k-1} - \eta_k)\left(1 - (C\eta_k^{-\gamma}S_k(t)+1)^{-\beta}\right), \tag{11}$$

where we also change the subscript indices from $k+1$ to $k$. Combining the above ansatz for the loss reduction term with the power-law ansatz for the auxiliary loss in (5) leads to our Multi-Power Law:

$$\mathcal{L}(t) \approx L_0 + A \cdot (S_1(t) + S_W)^{-\alpha} - \sum_{k=1}^{t}B(\eta_{k-1} - \eta_k)\left(1 - (C\eta_k^{-\gamma}S_k(t)+1)^{-\beta}\right). \tag{12}$$

See Appendix C for the ablation studies on different components of the Multi-Power Law.

## 4 EMPIRICAL VALIDATION OF THE MULTI-POWER LAW

The Multi-Power Law (MPL) comes from our speculations based on our experiments with special types of LR schedules. Now we present extensive experiments to validate the law for common LR schedules used in practice. Our experiments demonstrate that MPL requires only two or three LR

Table 1: Evaluation metrics for the Momentum Law and Multi-Power Law on predicting the loss curves of 25M, 100M, and 400M models with unseen schedules. $R^2$, MAE, RMSE, PredE, and WorstE are the coefficient of determination, Mean Absolute Error, Root Mean Square Error, Prediction Error, and Worst-case Error, respectively.

| Model Size | Method | $R^2 \uparrow$ | MAE $\downarrow$ | RMSE $\downarrow$ | PredE $\downarrow$ | WorstE $\downarrow$ |
|---|---|---|---|---|---|---|
| **25M** | Momentum Law | 0.9904 | 0.0047 | 0.0060 | 0.0014 | 0.0047 |
| | Multi-Power Law (Ours) | **0.9975** | **0.0039** | **0.0046** | **0.0012** | **0.0040** |
| **100M** | Momentum Law | 0.9959 | 0.0068 | 0.0095 | 0.0022 | 0.0094 |
| | Multi-Power Law (Ours) | **0.9982** | **0.0038** | **0.0051** | **0.0013** | **0.0058** |
| **400M** | Momentum Law | 0.9962 | 0.0071 | 0.0094 | 0.0025 | 0.0100 |
| | Multi-Power Law (Ours) | **0.9971** | **0.0053** | **0.0070** | **0.0019** | **0.0070** |

schedules and their corresponding loss curves in the training set to fit the law. The fitted MPL can then predict loss curves for test schedules with different shapes and extended horizons.

## 4.1 RESULTS

**Generalization to Unseen LR Schedules.** MPL accurately predicts loss curves for LR schedules outside the training set. As illustrated in Figure 2 and Table 1, despite the absence of WSD schedules in the training set and the variety of decay functions, MPL successfully predicts their loss curves with high accuracy. Furthermore, MPL generalizes to two-stage schedules with different $\eta_B$ values from the training set, effectively extrapolating curves for both continuous and discontinuous cases.

**Generalization to Longer Horizons.** MPL demonstrates the ability to extrapolate loss curves for horizons exceeding three times the training set length. In our runs, the training set contains approximately 22000 post-warmup steps, while the test set includes curves with up to 70000 post-warmup steps. These results validate MPL's capability to generalize to longer horizons. Notably, the data-to-model ratio for a 25M-parameter model trained over 72000 steps (36B tokens) is comparable to Llama2 pretraining (70B model, 2T tokens), consistent with trends favoring higher data volumes for fixed model sizes (Dubey et al., 2024).

**Generalization to Non-monotonic Schedules.** MPL extends effectively to complex non-monotonic schedules, although derived for monotonic decay schedules. We test the fitted MPL over challenging cases such as cyclic schedules and the *random-polyline schedule*, where LR values are randomly selected at every 8000 steps and connected by a polyline. These experiments, conducted on a 25M-parameter model over 72000 steps, also represent a demanding long-horizon scenario. As shown in Figure 4, MPL accurately predicts these long-horizon non-monotonic schedules.

## 4.2 COMPARISON WITH BASELINES

**Comparison with Chinchilla Law.** While Chinchilla-style data scaling laws, which we abbreviate as CDSLs, are widely adopted (Muennighoff et al., 2023; Hoffmann et al., 2022), MPL offers several distinct advantages: (1) MPL incorporates LR dependency, unlike CDSLs, and (2) MPL predicts the entire loss curve, whereas CDSLs are limited to estimating only the final loss. These advantages enable MPL to achieve higher sample efficiency than CDSLs. Notably, we demonstrate that a single constant and cosine schedule curve suffices to fit MPL with strong generalization. As illustrated in Figure 5(a), MPL reduces final loss prediction to less than $1/3$ that of CDSLs while requiring about $1/5$ compute budget. Furthermore, MPL excels in fitting the open-source 7B OLMo (Groeneveld et al., 2024), as shown in Figure 5(b). Additional details of the comparison with Chinchilla Law are provided in Appendix G.2.

**Comparison with Momentum Law.** The MPL outperforms the recently proposed Momentum Law(MTL) (Tissue et al., 2024) in both accuracy and applicability to discontinuous learning rate schedules. While MTL incorporates LR annealing effects by modeling loss reduction through the momentum of LR decay, it indicates an exponential loss reduction for two-stage LR schedules, inconsistent with our observations (see Appendix A.1). Across the diverse schedules in the test set, MPL consistently outperforms MTL in both average and worst-case prediction accuracy, as summarized in Table 1. Additionally, for WSD schedules with linear LR decay, MPL more accurately captures the loss reduction trend during the decay stage, as highlighted in Figure 14(b), compared to MTL. Further details on MTL and its relationship to MPL can be found in Appendix C, with fitting specifcs provided in Appendix G.2.

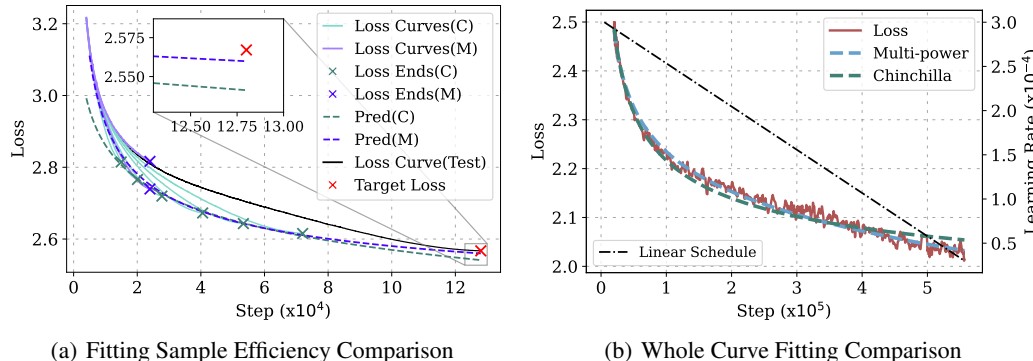

(a) Fitting Sample Efficiency Comparison      (b) Whole Curve Fitting Comparison

Figure 5: **(a)** Target loss predictions at 128000-step for a cosine schedule using MPL and CDSL fitting, with a 400M model. CDSL fitting requires six cosine losses (Loss Curve(C)) from 14,960 steps to 72000 steps but relies solely on their final losses (Loss Ends(C)). In contrast, MPL leverages the entire 24000-step constant and cosine loss curves (Loss Curves(M)). Final loss predictions are denoted as Pred(C) for CDSL and Pred(M) for MPL respectively. **(b)** Comparison of MPL and CDSL fittings on the whole loss curve of the open-source 7B OLMo model, trained with a linear schedule.

Table 2: Downstream performance comparison for the cosine and our optimized schedules. Percentage changes ($\uparrow$ or $\downarrow$) indicate relative improvements or regressions compared to the cosine schedule.

| Schedule | LAMBADA | HellaSwag | PIQA | ARC-E | $C^3$ | RTE |
|---|---|---|---|---|---|---|
| Cosine | 46.54 | 37.12 | **65.13** | 43.56 | 48.44 | 52.71 |
| **Optimized** | **48.71** | **37.74** | 65.07 | **44.09** | **50.30** | **53.79** |
| | ($\uparrow 2.17\%$) | ($\uparrow 0.62\%$) | ($\downarrow 0.06\%$) | ($\uparrow 0.53\%$) | ($\uparrow 1.86\%$) | ($\uparrow 1.08\%$) |

## 5 THE MULTI-POWER LAW INDUCES BETTER LR SCHEDULES

Due to the high cost of each pretraining run and the curse of dimensionality for LR schedules, it is generally impractical to tune the LR for every training step. To address this, we propose leveraging the Multi-Power Law (MPL) to predict the final loss as a surrogate function to optimize the entire LR schedule, achieving a lower final loss and outperforming the cosine schedule and WSD variants.

### 5.1 METHOD

The Multi-Power Law (MPL) provides an accurate loss estimation, enabling its final loss prediction to serve as a surrogate for evaluating schedules. We represent the learning rate (LR) schedule as a $T$-dimensional vector $E = (\eta_1, \ldots, \eta_T)$, with the final loss denoted as $\mathcal{L}(E)$ under given hyperparameters. Our goal is to find the optimal LR schedule $E^* = \arg\min_E \mathcal{L}(E)$. Using MPL, we parameterize the predicted final loss as $\mathcal{L}_\Theta(E)$ with parameters $\Theta = \{L_0, A, B, C, \alpha, \beta, \gamma\}$, estimated as outlined in Section 4. We approximate $E^*$ by optimizing the surrogate loss $\mathcal{L}_\Theta(E)$ subject to monotonicity constraints:

$$\min_E \ \mathcal{L}_\Theta(E) \quad \text{s.t.} \quad 0 \leq \eta_t \leq \eta_{t-1}, \forall\, 1 \leq t \leq T. \tag{13}$$

This optimization induces an "optimal" schedule under the MPL approximation. In practice, we set the peak LR $\eta_0 = 3 \times 10^{-4}$ We view $E$ as a high-dimensional vector and optimize it using the Adam optimizer. Further details are provided in Appendix H. Results for a 400M model are shown in Figure 1, with additional experiments for 25M and 100M models in Figure 18.

### 5.2 RESULTS

**Optimized LR Schedule Exhibits Stable-Decay Pattern.** Our optimized LR schedule follows a Warmup-Stable-Decay (WSD) pattern, comprising two main post-warmup phases: a stable phase with a constant peak LR, and a decay phase ending with a lower LR, as illustrated in Figures 1 and 18. By contrast, the momentum law (Tissue et al., 2024) theoretically yields a collapsed learning rate schedule, which we will prove in Appendix I.

**Optimized LR Schedule Outperforms Cosine Schedules.** Across comparison experiments of different model sizes and training steps, our optimized schedules consistently outperform the co-

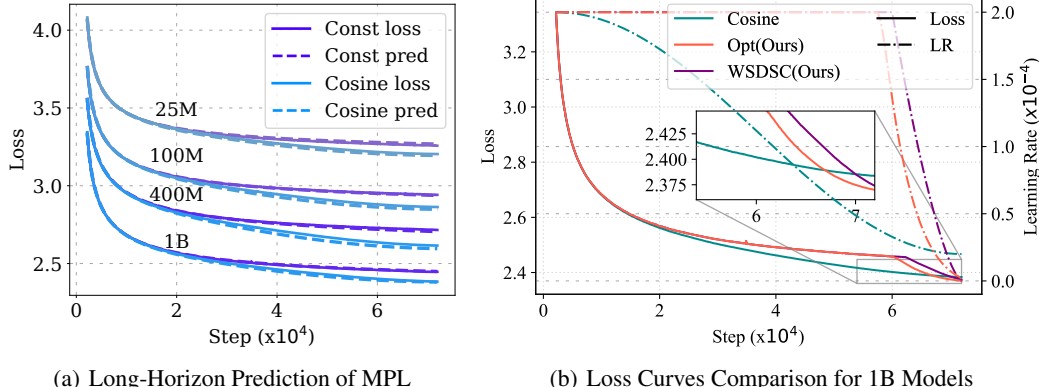

(a) Long-Horizon Prediction of MPL  (b) Loss Curves Comparison for 1B Models

Figure 6: **(a)** Long-horizon loss predictions using MPL for cosine and constant schedules, with model sizes ranging from 25M to 1B (top to bottom). **(b)** Loss curve comparison for 1B models across the optimized schedule (Opt), cosine schedule (Cosine), and simplified optimized schedule (WSDSC, see Section 5.2), featuring a WSD schedule with sqrt-cube decay.

sine schedules, achieving a margin exceeding 0.02. Notably, no WSD-like schedule is present in the training set, highlighting MPL's extrapolation capability. Figure 19 extends this comparison to longer training horizons and Figure 6(b) validates the superiority for 1B model. we further validate the effectiveness of our optimized schedules by evaluating the downstream task performance. As shown in Table 2, our optimized schedule leads to overall improvements in downstream tasks against the cosine schedules, showing practical gains from loss improvements. Ablation details for longer horizons and larger models are in Appendix H.

**Optimized LR Schedule Outperforms Tuned WSD Variants.** Our optimized schedules lead to smaller final loss than the WSD and WSDLD schedules proposed in Hu et al. (2024). For a 400M model, we find that the decay step of a 24000-step optimized schedule (Figure 1) is close to the optimally tuned step ($\sim$6000) for these WSD schedules, determined via grid search over $\{3000, 4000, 5000, 6000, 7000\}$. However, even when the decay ratios of WSD and WSDLD schedules are optimally tuned, our optimized schedule still outperforms them. Further, we analyze key differences between our optimized schedule and these two WSD schedules as follows. The optimized schedule decays to below $1/20$ of the peak LR, even approaching to zero, while WSD schedules decay linearly or exponentially to $1/10$ of the peak LR. However, simply adjusting the ending LR to near-zero (Appendix H) does not close the gap. Another key difference is the decay function: we find through symbolic regression that in the decay phase, the optimized schedule roughly follows a power decay function rather than a linear or exponential decay: $\eta_t \approx \eta_{\max} \cdot (1-\tau)^{1.5}$, where $\tau$ is the step number in the decay phase, normalized to $[0, 1]$. Motivated by this, we propose a WSD variant with sqrt-cube decay (WSDSC), which decays the LR exactly as $\eta_t = \eta_{\max} \cdot (1-\tau)^{1.5}$. WSDSC is effective across various model sizes and architectures and outperforms the WSD schedule, as shown in Figures 6(b) and 13(a), though it still falls behind our optimized schedule. See Appendix H for details.

## 6 CONCLUSIONS

This paper proposes the Multi-Power Law (MPL) to capture the relationship between loss and LR schedule. The fitted MPL accurately predicts the entire loss curve while requiring much fewer training runs compared to existing scaling laws. Furthermore, our MPL is accurate enough to be used for optimizing schedules, and we extensively validate the superiority of our optimized schedules over commonly used ones. However, we do observe slight deviations between our predictions and actual training curves, especially for long-horizon and high peak LR cases like in Figures 15 and 16. likely due to several simplifications in our derivation: (1) the coefficient $\beta$ remains constant across different LR scales; (2) the intermediate loss reduction does not depend on the LR prefix; (3) variations in LR during the warm-up phase are ignored.

In future work, we aim to (1) further explore the theoretical foundation of our MPL to uncover its underlying mechanisms; (2) investigate empirical laws for schedule-aware loss curve prediction with varying peak LRs and other hyperparameters; and (3) refine our MPL to further enhance its prediction accuracy and generalizability.

## ACKNOWLEDGMENT

We would like to thank Kaiyue Wen, Huanqi Cao, and all anonymous reviewers, for their insightful comments and feedback. We also thank Hongzhi Zang for improving figure readability. This work is supported by the National Natural Science Foundation of China under Grant Number U20B2044.

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

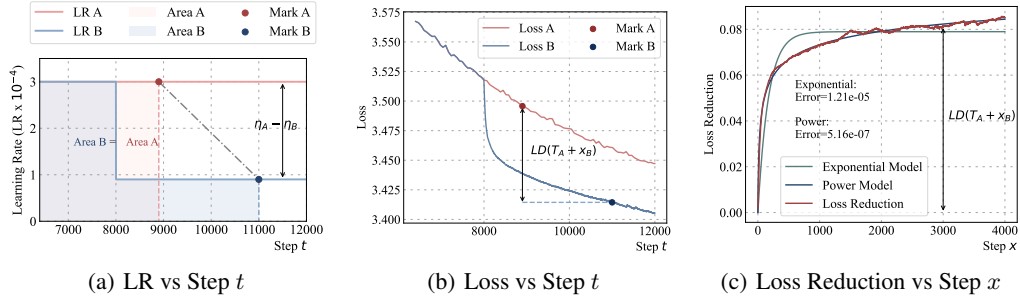

(a) LR vs Step $t$      (b) Loss vs Step $t$      (c) Loss Reduction vs Step $x$

Figure 7: Loss reduction (LD) of two-stage schedule exhibits a power law. Example setting: $t_B = 11000$, $x_B = 3000$, $\eta_B = 9 \times 10^{-5}$, $\eta_A = 3 \times 10^{-4}$, $T_A = 8000$. **(a)** A and B have the equal LR sums: $x_A = 900$, $t_A = 8900$. **(b)** Loss reduction at B: $\text{LD}(T_A + x_B) = \mathcal{L}_A(t_A) - \mathcal{L}_B(t_B)$. **(c)** Fitting loss reduction $\widehat{\text{LD}}(T_A + x_B)$ with power form results in $0.13(1 - (1 + 0.21x)^{-0.15})$; Fitting with exponential form results in $0.0790(1 - e^{-0.01x})$. The shape of loss reduction is closer to a power form than exponential.

# A   BOTTOM-UP DERIVATION: TWO-STAGE, MULTI-STAGE (SECTION 3.3)

## A.1   CASE 1: TWO-STAGE LEARNING RATE SCHEDULE

The two-stage schedule keeps learning rates at $\eta_A$ for $T_A$ steps, directly drops to $\eta_B$, and continues for $T_B$ steps. Then the LR reduction $\eta_A - \eta_B$ could be significant enough to induce $\text{LD}_{T_A}(t)$, which is also the loss reduction $\text{LD}(t)$ for step $t$ on Stage 2. See Appendix F.2 for experiment details.

**Loss Reduction Term Follows a Power Law.**   As shown in Figure 7, the number of steps $x := t - T_A$ in Stage 2 increases, $\text{LD}(T_A + x)$ monotonically rises from 0 to around 0.09 and eventually saturates. This motivates us to approximate $\text{LD}(T_A + x)$ in the form $\tilde{B} \cdot (1 - U(\eta_B x))$, where $\tilde{B}$ is a parameter and $U(s)$ is a function that decreases from 1 to 0 as $s = \eta_B x$ increases from 0 to infinity. The reason we choose $\eta_B x$ instead of $x$ as the argument of $U$ will be clear in the general case.

But at what rate should $U(s)$ decrease? After trying different forms of $U(s)$ to fit $\text{LD}(T_A + x)$, we find that the power-law form $U(s) = (\tilde{C} \cdot s + 1)^{-\beta}$ for some $\tilde{C}, \beta > 0$ fits most properly as shown in Figure 7, which leads to the following power-law form for the loss reduction term:

$$\text{LD}(T_A + x) \approx \widehat{\text{LD}}(T_A + x) := \tilde{B}(1 - (\tilde{C} \cdot \eta_B x + 1)^{-\beta}). \tag{14}$$

Appendix A.1 shows that this power law aligns well with the actual loss reduction term $\text{LD}(T_A + x)$. In contrast, the exponential form $U(s) = e^{-Bs}$ (so $\text{LD}(T_A + x) \approx A(1 - e^{-B\eta_B x})$) struggles to match the slow and steadily increase of $\text{LD}(T_A + x)$ when $x$ is large.

**Parameter Pattern of Power Law.**   We further investigate how to estimate the parameters $\tilde{B}, \tilde{C}, \beta$ in the power law. Based on our preliminary experiments, we set $\beta = 0.4$, a constant that works well. Then we conduct experiments to understand how the best parameters $\tilde{B}, \tilde{C}$ to fit $\text{LD}(t)$ depend on $\eta_A, \eta_B, T_A$, where we set default values $\eta_A = 3 \times 10^{-4}$, $\eta_B = 3 \times 10^{-5}$, $T_A = 8000$ and change one variable at a time. The details of ablation experiments can refer to Appendix F.2. The observations are summarized as follows.

(1) $\tilde{B}$ **is Linear to LR Reduction**. As shown in the first row of Figure 8, $\tilde{B}$ linearly decreases with $\eta_B$ and approximately increases linearly with $\eta_A$, especially when $\eta_A$ is not too large. Moreover, the slope of $\tilde{B}$ over $\eta_A$ and $\eta_B$ are approximately opposite to each other. This motivates us to hypothesize that $\tilde{B} \propto \eta_A - \eta_B$ and reparameterize $\tilde{B}$ as $\tilde{B} = B(\eta_A - \eta_B)$, where $B$ is a constant.

(2) $\tilde{C}$ **Follows a Power Law of $\eta_B$.** As shown in the second row of Figure 8, $\tilde{C}$ is very sensitive to $\eta_B$ but much less dependent on $\eta_A$. We hypothesize that $\tilde{C}$ follows a power law $\tilde{C} \propto \eta_B^{-\gamma}$, and reparameterize $\tilde{C}$ as $\tilde{C} = C\eta_B^{-\gamma}$, where $C > 0$ and $\gamma > 0$ are constants.

(3) **LR Reduction Term Depends Less on $T_A$.** We also find that $\tilde{B}$ and $\tilde{C}$ are less sensitive to $T_A$, relatively stable as $T_A$ varies, as shown in the last column in Figure 8. This suggests that the loss reduction has a weak dependency of loss reduction on LR prefix length.

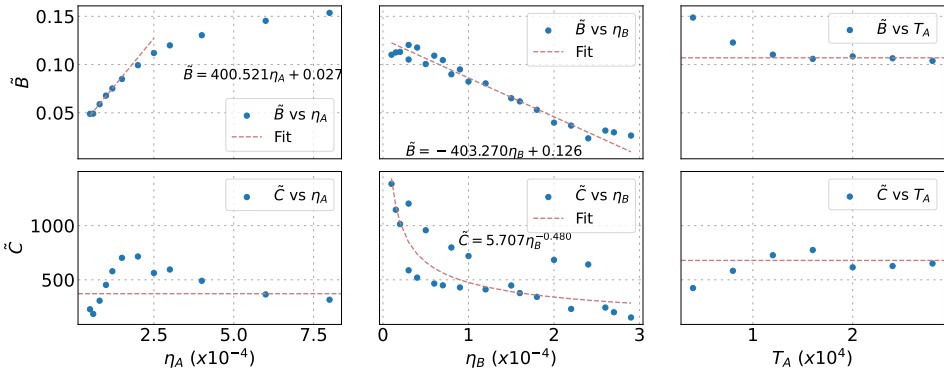

Figure 8: The dependency patterns of $\tilde{B}$, $\tilde{C}$ over $\eta_A$, $\eta_B$ and $T_A$ in the two-stage cases. $\tilde{B}$ is approximately proportional to $\eta_A - \eta_B$, and $\tilde{C}$ manifests power-law pattern over $\eta_B$. The dependency of $\eta_A$ over $\tilde{C}$ and the impacts of $T_A$ on $\tilde{B}$, $\tilde{C}$ are unpredictable or negligible, which are approximately ignored in our discussion.

**Approximation Form.** Putting all the above observations together, we have the final approximation form for the loss reduction term in the two-stage schedule:

$$\text{LD}(T_A + x) \approx \widehat{\text{LD}}(T_A + x) := B(\eta_A - \eta_B)\left(1 - (C\eta_B^{1-\gamma}x + 1)^{-\beta}\right). \tag{15}$$

## A.2 CASE 2: MULTI-STAGE LEARNING RATE SCHEDULE

In the two-stage case, the LR prefix is constant at $\eta_A$, leaving uncertainty about whether the intermediate loss reduction conforms to the power form when the LR prefixes vary. To investigate this, we analyze the multi-stage step decay schedule. Consider an $n$-stage LR schedule $E = \{\eta_1, \ldots, \eta_T\}$, where the $i$-th stage spans from step $T_i + 1$ to $T_{i+1}$ and uses the LR $\eta^{(i)}$ ($0 \leq T_1 < \cdots < T_{n+1} = T$, with $\eta_0 = \eta^{(0)} > \eta^{(1)} > \cdots > \eta^{(n)}$, $1 \leq i \leq n$). An example is illustrated in Figure 3.

**Stage-Wise Loss Reduction.** In the multi-stage schedule, given stage index $1 \leq i \leq n$, the stage-wise loss reduction is defined as $\text{LD}^{(i)}(t) = \text{LD}_{T_i}(t)$[2]. The LR reduction between stages, $\Delta\eta^{(i)} = \eta^{(i-1)} - \eta^{(i)}$, is also measurable. Using this, we estimate the shape of $\text{LD}^{(i)}(t)$ for different stages. Regard $T_i$ as $T_A$ in the two-stage case and define $x := t - T_i$. As shown in Figure 9(a), $\text{LD}^{(i)}(T_i + x)$ approximately conforms to a similar power law as (14) for the two-stage case:

$$\text{LD}^{(i)}(T_i + x) \approx \widehat{\text{LD}}^{(i)}(T_i + x) := \tilde{B}^{(i)}\left(1 - \left(\tilde{C}^{(i)} \cdot \eta^{(i)}x + 1\right)^{-\beta}\right), \tag{16}$$

where $\tilde{B}^{(i)}$ and $\tilde{C}^{(i)}$ are constants dependent on the LR prefix $\{\eta_1, \ldots, \eta_{T_i}\}$ for stage $i$.

**Intermediate Loss Reduction Weakly Depends on the LR Prefix Shape.** For stage $i$, the LR prefix is $\{\eta_1, \ldots, \eta_{T_i}\}$, which varies in length and scale across stages. To evaluate the effect of the LR prefix on the intermediate loss reduction form, we examine its impact on $\tilde{B}^{(i)}$ and $\tilde{C}^{(i)}$. Interestingly, as shown in Figure 9(b), we observe that $\tilde{B}^{(i)} \approx B(\eta^{(i-1)} - \eta^{(i)})$ and $\tilde{C}^{(i)} \approx C(\eta^{(i)})^{-\gamma}$, which align closely with the two-stage results. Here, $B$, $C$, and $\gamma$ are constants largely independent of the stage index. This suggests that intermediate loss reductions are relatively insensitive to the LR prefix compared to the LR reductions $\Delta\eta^{(i)}$ and the stage LR $\eta^{(i)}$. Moreover, this weak dependence on the LR prefix may extend to general schedules, indicating a broader applicability of the power-law form for intermediate loss reduction.

# B HOW MIGHT THE MULTI-POWER LAW ARISE?

In this section, we present a preliminary theoretical analysis to understand how the Multi-Power Law might arise. More specifically, we consider a simple setting where SGD optimizes a quadratic loss function with noisy gradients, and show that the Multi-Power Law naturally emerges when the Hessian and noise covariance matrices exhibit certain types of power-law structures. While this analysis does not fully capture the complexity of deep learning, we believe it offers insight into how the Multi-Power Law relates to underlying spectral properties in the optimization landscape.

---

[2]Note that $\text{LD}^{(i)}(t) = \text{LD}_{t^{(i)}}(t)$ for each $T_i + 1 \leq t^{(i)} \leq T_{i+1}$, as these auxiliary processes for a specific stage coincide.

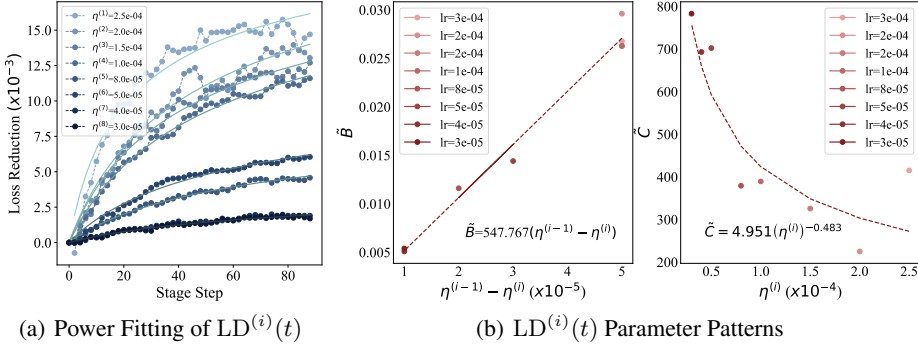

(a) Power Fitting of $\mathrm{LD}^{(i)}(t)$       (b) $\mathrm{LD}^{(i)}(t)$ Parameter Patterns

Figure 9: The intermediate loss reductions of a multi-stage schedule (Figure 3) and their shape patterns. **(a)** The loss reduction $\mathrm{LD}^{(i)}(t)$ between the adjacent stages of the multi-stage schedules still follows the power form. **(b)** $\tilde{B} \propto \eta^{(i-1)} - \eta^{(i)}$, $\tilde{C} \propto (\eta^{(i)})^{-\gamma}$. The parameter patterns in the two-stage setting hold in the multi-stage setting approximately. The shape of patterns is similar to the patterns in the two-stage experiments, as shown in Figure 8.

## B.1 SETUP

We consider a quadratic loss function $\mathcal{L}(\boldsymbol{\theta}) = \frac{1}{2}(\boldsymbol{\theta} - \boldsymbol{\theta}_*)^\top \boldsymbol{H}(\boldsymbol{\theta} - \boldsymbol{\theta}_*)$, where $\boldsymbol{\theta} \in \mathbb{R}^d$ represents the trainable parameters, $\boldsymbol{\theta}_*$ is the ground truth and $\boldsymbol{H} \in \mathbb{R}^{d \times d}$ is the Hessian matrix. Linear regression is a special case of this formulation. More generally, any sufficiently smooth loss function can be locally approximated by such a quadratic form near a minimizer.

We use SGD with LR schedule $E = \{\eta_1, \ldots, \eta_T\}$ to optimize the loss function, where the $t$-th iteration is given by $\boldsymbol{\theta}_t = \boldsymbol{\theta}_{t-1} - \eta_t \boldsymbol{g}_t$, with $\boldsymbol{g}_t$ being the stochastic gradient at step $t$. We assume that the stochastic gradient $\boldsymbol{g}_t$ equals the true gradient $\nabla \mathcal{L}(\boldsymbol{\theta}_{t-1}) = \boldsymbol{H}\boldsymbol{\theta}_{t-1}$ plus Gaussian noise $\mathcal{N}(\boldsymbol{0}, \boldsymbol{\Sigma})$, where $\boldsymbol{\Sigma} \in \mathbb{R}^{d \times d}$ is the covariance matrix. That is, $\boldsymbol{g}_t \sim \mathcal{N}(\boldsymbol{H}\boldsymbol{\theta}_{t-1}, \boldsymbol{\Sigma})$.

**From spectra to scaling law.** The scaling behavior of the loss during training is typically determined by the spectra of the Hessian matrix $\boldsymbol{H}$ and the noise covariance matrix (Canatar et al., 2021; Spigler et al., 2020; Maloney et al., 2022; Cui et al., 2021; Brandfonbrener et al., 2025). By carefully analyzing the training dynamics, we show that certain structures of Hessian and noise covariance matrices can lead to a scaling behavior similar to our empirical Multi-Power Law. In the following, we use $\lambda_i$ to denote the $i$-th eigenvalue of $\boldsymbol{H}$ and use $\Sigma_{ii}$ to denote the $i$-th diagonal entry of $\boldsymbol{\Sigma}$ in the eigenbasis of $\boldsymbol{H}$ (i.e., $\boldsymbol{v}_i^\top \boldsymbol{\Sigma} \boldsymbol{v}_i$, where $\boldsymbol{v}_i$ is the $i$-th eigenvector of $\boldsymbol{H}$). We initialize the parameters at $\boldsymbol{\theta}_0$ and use $\Delta_i$ to denote the $i$-th corrdinate of $\boldsymbol{\theta}_0 - \boldsymbol{\theta}_*$ in the eigenbasis of $\boldsymbol{H}$ (i.e., $\boldsymbol{v}_i^\top(\boldsymbol{\theta}_0 - \boldsymbol{\theta}_*)$). We consider a scenario where Hessian, noise covariance, and initial point are drawn from certain distributions before training, and make the following assumptions:

**Assumption 1.** *For all $1 \leq i \leq d$, the marginal distribution of $(\lambda_i, \Sigma_{ii}, \Delta_i)$ is a fixed distribution $p(\lambda, \mathcal{E}, \Delta)$ with following properties:*

- *$\lambda$ is supported on $[0, \Lambda]$ for some $\Lambda > 0$, and $p(\lambda) \propto \lambda^{-\nu}$ for some exponent $\nu \in [0, 1)$. That is, $p(\lambda) = \mathbb{1}_{\{\lambda \in [0, D]\}} \frac{\lambda^{-\nu}}{Z}$ for some normalization constant $Z > 0$.*

- *$\mathbb{E}_p[\mathcal{E} \mid \lambda] \propto \lambda^{-\rho} \exp(-r\lambda)$, for some $\rho < 1 - \nu$ and $r > 0$.*

- *$\mathbb{E}_p[\Delta^2 \mid \lambda] = D^2 \cdot \lambda^{-\kappa}$ for some $\kappa \in [0, 2 - \nu)$ and $D > 0$*

## B.2 MAIN THEOREM

Consider an SGD training process with an arbitrary LR schedule $E = \{\eta_1, \ldots, \eta_T\}$. Define $S_k(t) := \sum_{\tau=k}^{t} \eta_\tau$ for $1 \leq k \leq t \leq T$. Fix any $\eta_0 > 0$. We define the following function $\widehat{\mathcal{L}}(t)$ as an estimate of the expected loss at step $t$:

$$\widehat{\mathcal{L}}(t) := L_0 + A \cdot S_1(t)^{-\alpha} - \widehat{\mathrm{LD}}(t), \tag{17}$$

$$\widehat{\mathrm{LD}}(t) := B \sum_{k=1}^{t} (\eta_{k-1} - \eta_k) \cdot \widehat{G}(S_k(t)), \quad \widehat{G}(x) := 1 - \frac{\gamma(\beta, (2x + r)\Lambda)}{\gamma(\beta, r\Lambda)} \cdot (Cx + 1)^{-\beta}, \tag{18}$$

where the constants $L_0, A, \alpha, B, \beta$ above are given by

$$L_0 := \frac{d}{4}\eta_0 \mathbb{E}_p[\mathcal{E}], \; A := \frac{d \cdot \Gamma(\alpha)}{2^{\alpha+1}Z}D^2, \; \alpha := 2-\nu-\kappa, \; B := \frac{d}{4}\mathbb{E}_p[\mathcal{E}], \; \beta := 1-\nu-\rho, \; C := \frac{2}{r}. \quad (19)$$

Here, $\Gamma(x) := \int_0^{+\infty} t^{s-1}e^{-t}\mathrm{d}t$ denotes the gamma function, and $\gamma(s, x) := \int_0^x t^{s-1}e^{-t}\mathrm{d}t$ denotes the lower incomplete gamma function.

The theorem below establishes $\widehat{\mathcal{L}}(t)$ as a precise estimate of the expected loss curve of SGD. See Appendix J for the detailed proof.

**Theorem 1.** *Under Assumption 1, for all $1 \le t \le T$, if $\eta_{\max} := \max_{0 \le t < T}\{\eta_t\}$ is sufficiently small and $S_1(t)$ is sufficiently large, then we have the following estimate of $\mathbb{E}[\mathcal{L}(\boldsymbol{\theta}_t)]$:*

$$|\mathbb{E}[\mathcal{L}(\boldsymbol{\theta}_t)] - \widehat{\mathcal{L}}(t)| = O(\eta_{\max}S_1(t)^{-\min\{\alpha+1,\beta\}} + \eta_{\max}^2).$$

Here, Equation (17) is the same as Equation (1), and we can see that the exponent $\alpha$ is determined by the rate of eigenvalue decay of the Hessian and the rate of variance decay of the initial parameter. The coefficients $L_0 \propto \eta_0\mathbb{E}_p[\mathcal{E}], A \propto D^2$ depend on the learning rate, variance of gradient noise, and the initial distance to the optimal parameter.

Equation (18) gives the loss reduction term $\widehat{\mathrm{LD}}(t)$, which is similar to the loss reduction term in Equation (2). A change in the learning rate at step $k$ induces a loss reduction $B(\eta_{k-1}-\eta_k)\widehat{G}(S_k(t))$, where $B \propto \mathbb{E}_p[\mathcal{E}]$ depends on the variance of gradient noise. Similar to (2), this loss reduction saturates as $B(\eta_{k-1} - \eta_k)(1 - \Theta(S_k^{-\beta}))$ when $S_k(t)$ becomes large, where $\beta$ is determined by the rate of eigenvalue decay of the Hessian and the rate of variance decay of the gradient noise. But a slight misalignment between theory and practice here is that the form of $\widehat{G}(S_k(t))$ is not the same as $G(\eta_k^{-\gamma}S_k(t))$ in (2). The main discrepancy here is that $G(\eta_k^{-\gamma}S_k(t))$ has an explicit dependence on the current learning rate $\eta_k$, while $\widehat{G}(S_k(t))$ does not. We suspect that this is due to that in practice, changing the learning rate can also change the local loss landscape, such as the well-known Edge of Stability (EoS) phenomenon (Cohen et al., 2021; Damian et al., 2023; Lyu et al., 2022; Cohen et al., 2025), but the quadratic loss function we consider here is not complex enough to exhibit such a behavior. In addition, the function $G(x)$ is in a simple power form, while $\widehat{G}(x)$ also involves a lower incomplete gamma function $\gamma(\beta, (2x+r)\Lambda)$ and only approximately follows the power form. We argue that this is a minor difference since these functions are approximately the same especially for large $x$, as we plot in Figure 21. Further, if $\Lambda \to +\infty$, i.e., the maximum eigenvalue of the Hessian is very large, it is easy to see that $\widehat{G}(x) \to G(x)$ with the same $C$ and $\beta$.

Extending the analysis of our Multi-Power Law beyond quadratic cases is a key direction for future work. A deeper understanding of loss landscape properties in deep learning is crucial for this generalization. For example, a recent paper (Wen et al., 2025) conjectures that the loss landscape in LLM pretraining exhibits a river valley landscape, which is similar to a deep valley with a river at its bottom. Based on this conjecture, they further explained the success of WSD schedules. For future work, it would be interesting to extend our analysis to this river valley landscape or other frameworks that better capture the complex structure of the loss function in practical deep learning scenarios.

## C  FORMULA COMPONENT ABLATION

To understand and evaluate the role of each component in our Multi-Power Law (MPL; See (1)), we systematically simplify the MPL formula at various levels and explore alternative formulations. Table 3 summarizes the fitting performance of these simplified versions and variants of the MPL. The fitting experiments are conducted on 25M models, using the same experimental setup described in Appendix E.

**No Loss Reduction.**  The necessity of the loss reduction term $LD(t)$ can be assessed by fitting a One-Power Law (OPL), a simplified MPL where $LD(t) = 0$ or equivalently $B = 0$:

$$\mathcal{L}_{\mathrm{OPL}}(t) = L_0 + A \cdot (S_1(t) + S_W)^{-\alpha}, \quad S_1(t) := \sum_{\tau=1}^t \eta_\tau. \quad (20)$$

This formulation approximates the loss curve by matching the LR sum without correction term, as discussed in Section 3.1. The fitted results (first row of Table 3) exhibit significant degradation compared to the full MPL, demonstrating the critical role of $LD(t)$.

**Linear Approximation of Loss Reduction.** Based on the observation in Section 3.2.2, the loss reduction term $LD(t)$ (defined in Equation (2)) can be simplified by treating the scaling function $G(x)$ as a constant:

$$\text{LD}(t) \approx \sum_{k=1}^{t} B(\eta_{k-1} - \eta_k) = B(\eta_0 - \eta_t). \tag{21}$$

Despite its simplicity, we observe a near-linear relationship between $LD(t)$ and the LR reduction $(\eta_0 - \eta_t)$, regardless of the LR schedule type, as shown in Figure 10. This motivates the Linear Loss reDuction Law (LLDL):

$$\mathcal{L}_{\text{LLDL}}(t) = L_0 + A \cdot (S_1(t) + S_W)^{-\alpha} + B(\eta_0 - \eta_t). \tag{22}$$

As shown in Table 3, LLDL achieves significantly better accuracy than OPL, although it underperforms the full MPL. However, this formulation is unsuitable for optimizing schedules, as its results collapse to a trivial solution: $\eta_k = \eta_0$ when $k \le T - 1$ and $\eta_k = 0$ when $k = T$.

**Loss Reduction Without $\gamma$.** Next, we simplify $G(x)$ by setting $\gamma = 0$, yielding the No-$\gamma$ Law:

$$\mathcal{L}_{\text{No}-\gamma} = L_0 + A \cdot (S_1(t) + S_W)^{-\alpha} + B \sum_{k=1}^{t} (\eta_{k-1} - \eta_k) \cdot G(S_k(t)). \tag{23}$$

Results (third row of Table 3) show a slight performance drop, confirming that $\gamma$ enhances fitting accuracy with minimal additional computational cost. Thus, we retain $\gamma$ in the final MPL.

**Step-Based Approximation.** An alternative is to replace $G(\eta_k^{-\gamma} S_k(t))$ with a step-based formulation, $G(t - k + 1)$. This yields the Step Power Law (SPL):

$$\mathcal{L}_{\text{SPL}} = L_0 + A \cdot (S_1(t) + S_W)^{-\alpha} + B \sum_{k=1}^{t} (\eta_{k-1} - \eta_k) \cdot G(t - k + 1). \tag{24}$$

While simpler, this approximation reduces prediction accuracy and contradicts empirical results, because it implies loss reduction continues to increase even when LR reaches zero.

**Exponential Approximation.** Substituting $G(x)$ with an exponential function $G(x) = 1 - e^{-Cx}$ gives the Multi-Exponential Law (MEL):

$$\mathcal{L}_{\text{MEL}} = L_0 + A \cdot (S_1(t) + S_W)^{-\alpha} + B \sum_{k=1}^{t} (\eta_{k-1} - \eta_k) \cdot G(S_k(t)). \tag{25}$$

Results (fifth row of Table 3) show a performance drop compared to the power-based MPL, consistent with observations in Appendix A.1 that $\tilde{U}(t, \eta_k)$ takes a power form rather than an exponential form.

**Relation to Momentum Law.** The concurrently proposed MomenTum Law (MTL) is in the form of

$$\mathcal{L}_{\text{MTL}}(t) = L_0 + A \cdot (S_1 + S_W)^{-\alpha} + B \cdot S_2, \text{ where } S_1 = \sum_{i=1}^{t} \eta_i, \; S_2 = \sum_{i=1}^{t} \sum_{k=1}^{i} (\eta_{k-1} - \eta_k) \lambda^{i-k},$$

where $\lambda$ is a hyper-parameter of MTL and $\lambda < 1$. It is indeed a variant of MPL since

$$S_2 = \sum_{i=1}^{t} \sum_{k=1}^{i} (\eta_{k-1} - \eta_k) \lambda^{i-k} = \sum_{k=1}^{t} (\eta_{k-1} - \eta_k) \sum_{i=k}^{t} \lambda^{i-k} = \sum_{k=1}^{t} (\eta_{k-1} - \eta_k) \left( \frac{1 - \lambda^{t-k+1}}{1 - \lambda} \right).$$

Thus, MTL is a variant of MPL with an exponential step-based approximation:

$$\mathcal{L}_{\text{MTL}}(t) = L_0 + A \cdot (S_1(t) + S_W)^{-\alpha} + B' \cdot G(t - k + 1), \quad G(x) = 1 - e^{-C'x}.$$

Here, $B' = \frac{B}{1-\lambda}$, $C' = -\log \lambda$. MTL incorporates step-based decay and its performance (last second row of Table 3) even lags behind MEL, highlighting the limitations of step-based approximations.

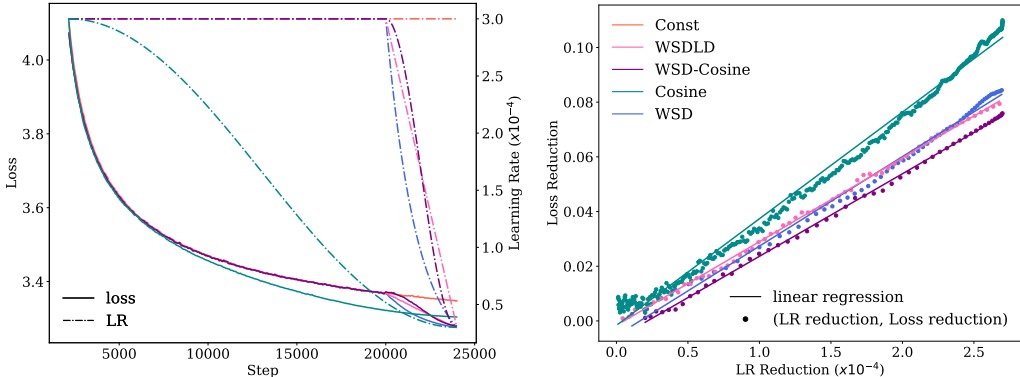

Figure 10: Linear regression of loss reduction versus LR reduction across different schedules for a 25M model over 24000 steps. Decay steps are set at 4000 for WSD and its variants, among which WSD-Cosine specifically denotes the WSD schedule with cosine decay function. **Left:** Visualization of learning rate schedules and their corresponding loss curves. **Right:** Scatter plot of loss reductions against LR reductions, accompanied by a linear regression fit (mean $R^2 = 0.9980$), demonstrating a strong linear relationship between the two variables.

Table 3: Summary of fitting results for simplified versions and variants of the MPL. Metrics include $R^2$, MAE, RMSE, PredE, and WorstE, where higher $R^2$ values and lower values of other metrics indicate better fitting performance. See Table 1 for metric definitions.

| Formula | Features | $R^2 \uparrow$ | MAE↓ | RMSE↓ | PredE↓ | WorstE↓ |
|---------|----------|------|------|-------|--------|---------|
| OPL | $LD(t) = 0\ (B = 0)$ | 0.8309 | 0.0378 | 0.0412 | 0.0111 | 0.0241 |
| LLDL | $G(x) = 1$ | 0.9797 | 0.0077 | 0.0101 | 0.0023 | 0.0108 |
| No-$\gamma$ | $\gamma = 0$ | 0.9961 | 0.0046 | 0.0053 | 0.0014 | 0.0041 |
| SPL | $x = t - k$ | 0.9921 | 0.0066 | 0.0075 | 0.0020 | 0.0069 |
| MEL | $G(x) = 1 - e^{-Cx}, \gamma = 0$ | 0.9934 | 0.0044 | 0.0057 | 0.0013 | 0.0047 |
| MTL | $G(x) = 1 - e^{-Cx}, x = t - k$ | 0.9904 | 0.0047 | 0.0060 | 0.0014 | 0.0047 |
| **MPL** **(Ours)** | $G(x) = 1 - (Cx + 1)^{-\beta}$, $x = \eta_k^{-\gamma} \sum_{\tau=k}^{t} \eta_\tau$ | **0.9975** | **0.0039** | **0.0046** | **0.0012** | **0.0040** |

## D  RELATED WORK

**Scaling Laws and Loss Curve Prediction.**   Scaling laws reveal empirical power-law relationships between training losses and various factors such as model size, dataset size, and computational resources. Hestness et al. (2017) initially observed that generalization errors in deep learning decrease predictably with larger model and dataset scales, following a power-law trend. Subsequently, Kaplan et al. (2020) investigated these scaling laws in the context of training Transformers (Vaswani et al., 2017), demonstrating a persistent power-law decay in loss and contributing to the rise of LLMs (Mann et al., 2020; Bi et al., 2024; Touvron et al., 2023; Dubey et al., 2024). Later studies refined these scaling laws by improving fitting and evaluation pipelines, refining parameter size metrics, ensuring consistent hyperparameter configurations across scales, and extending their applicability to broader training scenarios (Hoffmann et al., 2022; Henighan et al., 2020; Bi et al., 2024; Caballero et al., 2022; Alabdulmohsin et al., 2022).

Scaling laws have been used to predict the model performance in various settings, hyperparameter optimization (Kadra et al., 2023), multi-epoch training (Muennighoff et al., 2023), training sparsely-connected models (Frantar et al., 2023; Ludziejewski et al., 2024), length extrapolation (Liu et al., 2023), transfer learning (Hernandez et al., 2021), and data mixing (Hashimoto, 2021; Ge et al., 2024; Ye et al., 2024; Jain et al., 2024). However, most existing work neglects the impact of the learning rate, making their predictions unreliable for assessing model performance throughout training (Hoffmann et al., 2022). As a result, deriving scaling laws for final losses under specific schedules (e.g., the cosine schedule (Hoffmann et al., 2022; Muennighoff et al., 2023)) requires more than 10 full

training runs. Another line of works extrapolate loss curves with Bayesian methods (Klein et al., 2017; Domhan et al., 2015; Choi et al., 2018; Adriaensen et al., 2023) for hyperparameter optimization, but they also need a large number of training runs. In comparison, our work introduces a multi-power law that extends the existing power-law form of specific LR schedules by incorporating terms to capture LR schedule effects. Unlike prior work, our approach is LR-dependent and can predict full loss curves across various schedules using fewer than three training curves.

Concurrent to our work, Tissue et al. (2024) proposed a momentum law that also incorporates LR into scaling laws. However, our work outperforms theirs by identifying a power-law decay rather than an exponential decay in loss reduction relative to LR reduction (detailed in Section 4.2). This results in a more accurate formula capable of inducing optimized schedules, whereas the momentum law provably yields suboptimal collapsed schedules (detailed in Appendix I). In another concurrent work, Xie et al. (2024) derived a LR-dependent scaling law based on theoretical insights drawn from SDE-based continuous-time approximation of training dynamics. However, their scaling law relies on manually defined training phases and is limited to predicting the final loss value. In contrast, our proposed scaling law is arguably more general, as it can predict the entire loss curves for various LR schedules, even including discontinuous and non-monotonic ones. This also allows us to optimize the LR schedule by minimizing the predicted loss over all possible LR decay schedules, which may not be easily achievable using their method.

**Theoretical Insights on Scaling Laws.** Many works have explored theoretical explanations for the observed power-law behavior. Sharma & Kaplan (2022) attribute the power-law behavior to the intrinsic dimension of data manifold. Hutter (2021); Michaud et al. (2023) draw insights from a toy case where an infinite amount of distinct knowledge pieces need to be memorized, and the power law of the loss curve can arise when the frequency of knowledge exhibits a power-law distribution. Many other works analyze the scaling law of linear models (Spigler et al., 2020; Bordelon et al., 2020; Maloney et al., 2022; Bahri et al., 2024; Wei et al., 2022; Bordelon et al., 2024; Lin et al., 2024; Atanasov et al., 2024; Paquette et al., 2024), assuming certain power-law properties in the input data or ground truth functions. A few others examine how power law behaviors arise in simple neural networks (Nam et al., 2024; Bordelon et al., 2025; Lyu et al., 2025). Similar to these works, we also provide a theoretical explanation for our multi-power law assuming certain power-law properties in the optimization landscape, but our analysis is accurate enough to capture the effects of learning rate schedules on the loss curve.

**Optimal Learning Rate Schedule.** Setting a proper schedule for the learning rate is crucial for training deep neural networks. He et al. (2016) introduced the warmup strategy, which is now standard in modern schedules. Early work by Smith (2017) proposed cyclical LR schedules, featuring periodic linear decay with warmup restarts, later extended to cosine decay with warmup restarts by Loshchilov & Hutter (2017). Some works explored adaptive approaches, such as Bayesian or reinforcement learning-based methods (Xu et al., 2019; Teng et al., 2021), but they are computationally expensive for LLMs. Li & Arora (2020) demonstrated that training with one schedule accompanied by weight decay is equivalent to training with the same network with an exponentially increasing LR schedule without weight decay. Goyal et al. (2017); Hoffer et al. (2017); Malladi et al. (2022) studied how to scale the LR when increasing the batch size. Li et al. (2019); You et al. (2019) showed that LR decay can benefit generalization by suppressing the memorization of noisy data early in training and learning complex patterns late in training.

In the context of large model training, Hu et al. (2024) introduced the Warmup-Stable-Decay (WSD) schedule, which starts with a warmup phase, continues a main stable phase, and ends with a rapid decay phase. This schedule has shown strong performance in LLM pretraining and efficient continual training. Similar patterns have been adopted in other works (Zhai et al., 2022; Hägele et al., 2024). Ibrahim et al. (2024); Zhai et al. (2022); Raffel et al. (2020) adopt a reciprocal (inverse)-sqrt LR schedule in full process or as a component. Wen et al. (2025) analyze the benefits of WSD schedules by conjecturing that the loss landscape exhibits a river valley structure, and propose alternatives for WSD in continual training. Inspired by these works, recent open-source models advocate schedules with a slow-decay or stable phase followed by a rapid decay (Liu et al., 2024; OLMo et al., 2024).

Our resulting scaling law can induce optimized schedules share a similar pattern with the WSD schedule, even though we do not fit the law on WSD schedules. Concurrent to our work, other schedules have claimed optimality under specific conditions. Defazio et al. (2023) proposed linear

decay schedules as optimal based on worst-case analysis. Shen et al. (2024) introduced a power schedule for continual training, which outperforms WSD schedules. Schaipp et al. (2025) drew parallels between convex optimization and LR scheduling for LLMs, using simulation results to guide continual training strategies and peak LR selection. Bergsma et al. (2025) argued for a linear-to-zero LR schedule as optimal, ablating on peak LR, data size, and model size. Defazio et al. (2024) proposed a schedule-free approach using weight averaging techniques, but it underperforms WSD schedules (Hägele et al., 2024). Existing methods often optimize schedules under specific constraints, such as ending LR (Bergsma et al., 2025), decay ratio (Schaipp et al., 2025), continual training (Shen et al., 2024), or worst-case convergence bounds (Defazio et al., 2023). In contrast, our approach integrates LR schedules into scaling laws, which enables gradient-based optimization over all possible schedules.

## E  EXPERIMENT SETTING

Unless otherwise specified, the model training in the Section 3, 4 and 5 follows the following settings.

| Codename | Embedding Dimension | #Heads | #Layers | #Non-embeddings | #Params |
|----------|---------------------|--------|---------|-----------------|---------|
| 25M | 640 | 5 | 5 | 25 | 89 |
| 100M | 1024 | 8 | 8 | 101 | 205 |
| 400M | 1536 | 12 | 12 | 340 | 493 |
| 1B | 2048 | 32 | 16 | 822 | 1026 |
| GPT-2 | 768 | 12 | 12 | 85 | 162 |

Table 4: The model series run in all the experiments. Hoffmann et al. (2022) utilizes the number of non-embedding parameters (#Non-embeddings) to count model sizes, while Kaplan et al. (2020) counts the total number of parameters (#Params). The unit of the Parameter is M in this table.

| Default Hyperparameter | Value |
|------------------------|-------|
| Sequence Batch Size | 128 |
| Sequence Length | 4096 |
| Optimizer Type | AdamW |
| $\beta_1$ | 0.9 |
| $\beta_2$ | 0.95 |
| $\epsilon$ | $1 \times 10^{-8}$ |
| Weight Decay | 0.1 |
| Gradient Clipping | 1.0 |
| Peak Learning Rate | $3 \times 10^{-4}$ |
| Final Learning Rate | $3 \times 10^{-5}$ |
| Warmup Steps | 2160 |

Table 5: Hyperparameters related to model training.

Our validation contains two steps: (1) fitting schedule-curve pairs from the training set and (2) predicting the loss curves for schedules in the test set. The training set contains only a single 24,000-step constant and cosine schedule pair, alongside a 16,000-step two-stage schedule of $\eta_B = 0.3\eta_A$. The test set has one 72,000-step constant and cosine schedule, 24,000-step unseen WSD and WSDLD schedules, and 16,000-step two-stage schedules with $\eta_B = 0.1\eta_A$ and $\eta_B = 0.6\eta_A$. The details are provided in Table 6. We train Llama2 (Touvron et al., 2023) models of 25M, 100M, and 400M, and collect their loss curves, with model parameter details in Table 4. Training employs the AdamW optimizer, with a weight decay of 0.1, gradient clipping at 1.0, $\beta_1 = 0.90$, and $\beta_2 = 0.95$, consistent with the Llama2 training setup. Default hyperparameters include a peak LR of $3 \times 10^{-4}$, a warmup period of 2160 steps, and a batch size of 0.5M. Additional hyperparameters are detailed in Table 5. In ablation studies, we simplify the experiment to fit short constant and cosine schedules and predict the loss for a long-horizon cosine schedule. The MPL fitting employs Huber loss (Huber, 1992) as the objection function, aligning with prior work (Hoffmann et al., 2022; Muennighoff et al., 2023),

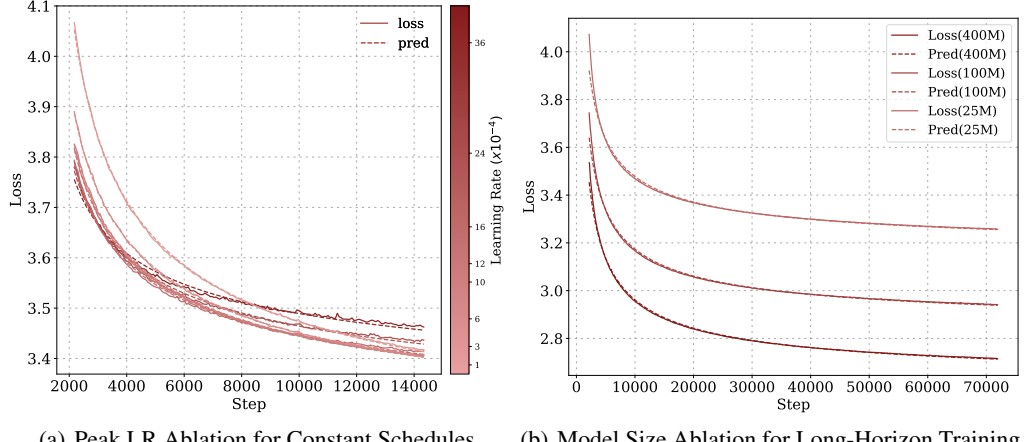

(a) Peak LR Ablation for Constant Schedules  (b) Model Size Ablation for Long-Horizon Training

Figure 11: Loss curves for constant LR schedules. *pred* denotes the fitted law prediction and *loss* represents the ground-truth loss curve. See Appendix F.1 for details.

and uses the Adam optimizer for optimization. Unless otherwise specified, we report validation loss. For fitting approaches and additional details see Appendix G.

# F DISCUSSIONS OF MULTI-POWER LAW DERIVATION (SECTION 3)

## F.1 CONSTANT PROCESS LOSS APPROXIMATION (SECTION 3.1)

The constant process employs a constant LR schedule with the same warmup phase and peak LR as the actual process schedule. We validate (5), the LR sum power law of the loss curves for constant schedules, through two series of experiments. First, we conduct ablation over the peak LR, ranging from $3.0 \times 10^{-4}$ to $3.6 \times 10^{-3}$ over 14,400 steps, achieving an MSE of $1.55 \times 10^{-5}$ and $R^2$ of 0.9976 (Figure 11(a)). Second, we validate the power form over long-horizon curves (72000 steps) for model sizes of 25M, 100M, and 400M, with a peak LR of $3.0 \times 10^{-4}$, yielding an average MSE of $8.04 \times 10^{-5}$ and $R^2$ of 0.9947 (Figure 11(b)).

## F.2 TWO-STAGE EXPERIMENTS (SECTION A.1)

In this section, we provide details on the investigation of the variation of coefficients in the power law for two-stage LR schedules.

**Experiment Setting and Law Fitting.** The experiment setting aligns with Appendix E. Default configuration uses $\eta_A = 3 \times 10^{-4}$, $\eta_B = 3 \times 10^{-5}$, $T_A = 8000$. In the ablation experiments, $\eta_A$ ranges from $5 \times 10^{-5}$ to $1 \times 10^{-3}$, $\eta_B$ ranges from $4 \times 10^{-5}$ to $2.9 \times 10^{-4}$, and $T_A$ ranges from 4000 to 28000. The second stage lengths spanning 1000 to over 6000 steps. Validation loss is sampled every 2 steps due to the rapid loss changes after the stage switch. Following Hoffmann et al. (2022), we fit the law utilizing Huber loss as the objection function (Huber, 1992),

$$\min_{\Theta} \sum_x \text{Huber}_\delta(\log \widehat{\text{LD}}_\Theta(T_A + x) - \log \text{LD}(T_A + x)), \tag{26}$$

where $\Theta = \{\tilde{B}, \tilde{C}, \beta\}$, and we set $\delta = 1 \times 10^{-2}$. For each experiment, we use the Adam optimizer with a learning rate at $1 \times 10^{-4}$ and total steps of 20000. Here we do not conform to the L-BFGS algorithm like Hoffmann et al. (2022) due to its sensitivity to the initialization. In our fitting, the parameters are initialized based on the loss reduction curve shape: $\tilde{B}$ corresponds to the estimation of asymptotic values of loss reduction and $\tilde{C}$ can be estimated according to the slope at $x = 0$ step (Equation (14)).

**Fixed $\beta$ Experiments for Parameter Patterns.** We fit the power-law form in Equation (14) across ablation experiments to identify the loss curve shape and power-law parameter patterns. For the sake of further derivation, we fix the exponent $\beta$ as LR-independent parameter 0.4 based on the warmup

experiments. Then we re-fit the loss curves fixing $\beta = 0.4$ to confirm the validity of the power form. Figure 12 includes the re-fitted curves and ground truths for the ablation experiments over $\eta_A$ and $\eta_B$, showing feasible error margins for further derivation despite fixed $\beta$. We further investigate the dependency of different parameters on the $\eta_A$, $\eta_B$, and $T_A$, with pair-wise relations presented in Figure 8 and summarized in Appendix A.1.

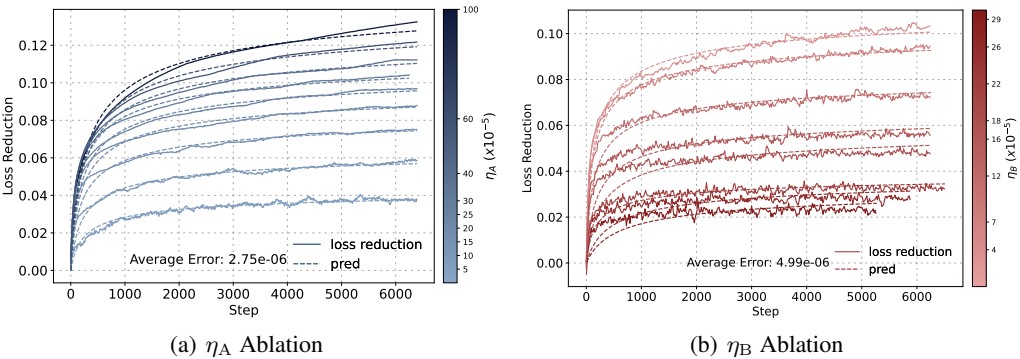

(a) $\eta_A$ Ablation  (b) $\eta_B$ Ablation

Figure 12: Power-law fitting of loss reductions versus steps $x$ for two-stage LR schedules.

### F.3 MULTI-STAGE EXPERIMENTS (SECTION A.2)

We analyze intermediate loss reduction dependency on the LR prefix, through experiments of a multi-stage schedule and its auxiliary (intermediate) processes. As shown in Figure 3, our multi-stage schedule consists of 9 stages, with a first stage of 8000 steps at $3 \times 10^{-4}$, followed by eight 90-step stages. The validation interval is also set to 2 steps. For adjacent stages $i-1$ and $i$ ($x \le 90$), We compute $\mathrm{LD}^{(i)}(T_i + x)$, as defined in Appendix A.2 and fit it going through the same law fitting process as Equation (26). The fitting of loss reductions for different stages is presented in the left panel of Figure 9. Moreover, parameter trends, analogous to the two-stage findings, reveal $\tilde{B}^{(i)}$ changing with $\eta^{(i-1)} - \eta^{(i)}$ and $\tilde{C}^{(i)}$ changing with $\eta^{(i)}$, shown in the right two sub-figures of Figure 9.

## G   DETAILS OF VALIDATION EXPERIMENTS (SECTION 4)

### G.1   TRAINING SET AND TEST SET

| Set | Schedule Type | Total Lengths | $\eta_B/\eta_A$ |
|---|---|---|---|
| Training | Constant | 24000 | |
| | Cosine | 24000 | |
| | Two-stage | 16000 | 0.3 |
| Test | WSD | 24000 | |
| | WSDLD | 24000 | |
| | Two-stage | 16000 | 0.1 |
| | Two-stage | 16000 | 0.6 |
| | Constant | 72000 | |
| | Cosine | 72000 | |

Table 6: Summary of training and test sets.

Our validation frames the Multi-Power Law (MPL) fitting as a machine learning task, training on schedule-loss curve pairs from the training set and predicting loss curves for the test set. The training set contains a 24000-step constant and cosine schedule pair, and a 16000-step two-stage schedule with $\eta_B = 0.3\eta_A$. The test set includes a 72000-step constant and cosine schedule, a 24000-step unseen WSD and WSDLD schedule, and 16000-step two-stage schedules with $\eta_B = 0.1\eta_A$ and

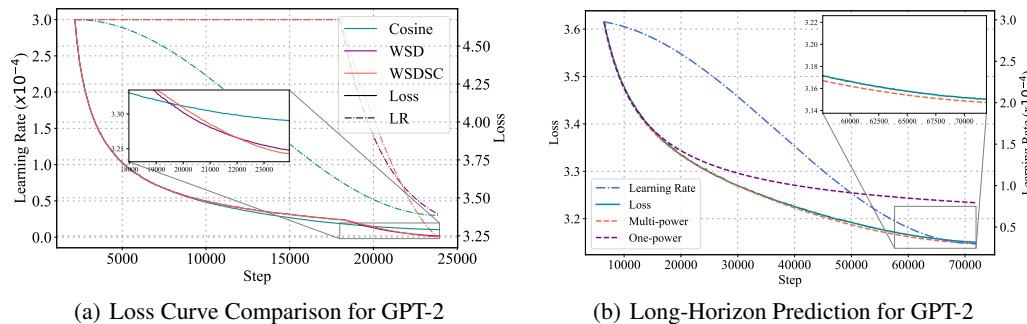

(a) Loss Curve Comparison for GPT-2      (b) Long-Horizon Prediction for GPT-2

Figure 13: Loss curves of GPT-2 models with Multi-Power Law fitted over 24000-step constant and cosine schedule losses. **(a)** Comparison between the cosine, WSD, and WSDSC schedules (see Section 5.2); **(b)** Prediction for a 72000-step cosine schedule loss curve.

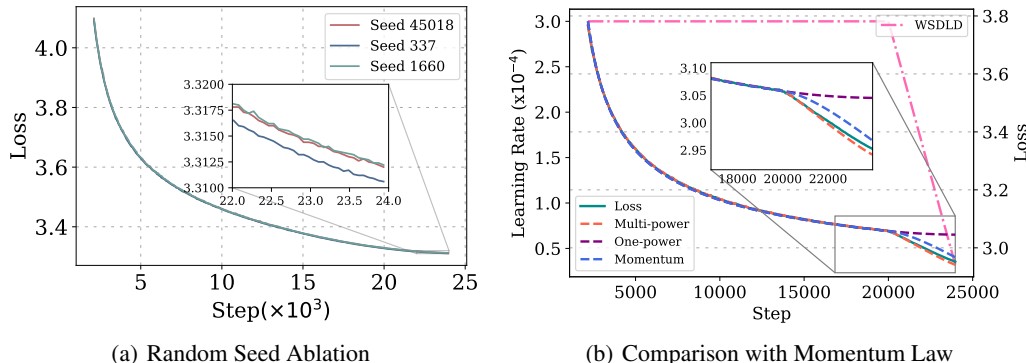

(a) Random Seed Ablation      (b) Comparison with Momentum Law

Figure 14: **(a)** Experiments with a 25M model over 24000 steps across different seeds, showing a final loss standard deviation of 0.0007 and a maximum gap of 0.0014. **(b)** Comparison between Multi-Power Law (MPL) and Momentum Law (MTL). In the decay stage, MPL achieves higher fitting accuracy and matches the curvature of the loss curve, whereas MTL fits the stable stage but predicts a counterfactual concave curve during the decay stage.

$\eta_B = 0.6\eta_A$. The peak learning rate is $3 \times 10^{-4}$, and the ending learning rate is $3 \times 10^{-5}$ for the cosine, WSD, and WSDLD schedules. For all two-stage schedules, $T_A = 8000$. All schedules include a warmup phase of 2,160 steps. Detailed descriptions of the training and test sets are summarized in Table 6.

## G.2 FITTING THE LAW

Similar to the two-stage fitting, we fit the parametric law using the Huber loss as the objective (Huber, 1992):

$$\min_{\Theta} \sum_t \text{Huber}_\delta(\log \mathcal{L}_\Theta(X_t) - \log \mathcal{L}_{\text{gt}}(X_t)), \tag{27}$$

where $\mathcal{L}_{\text{gt}}(X_t)$ denotes the ground truth of validation loss, $\mathcal{L}_\Theta(X_t)$ is the predicted loss, $\delta$ is a hyperparameter for the Huber loss, and $\Theta$ denotes parameters to fit. The total fitting loss sums up the Huber loss over the validation steps. In practice, we compute the area under the linearly interpolated polyline of the learning rate at validation steps as a surrogate for the LR sum. This approach reduces the computational cost since requiring only step numbers, learning rates, and losses at validation steps.

**Multi-Power Law.** For the Multi-Power Law (MPL), $\Theta = \{A, B, C, \alpha, \beta, \gamma, L_0\}$, and $X_t = \{\eta_1, \ldots, \eta_t\}$. We use the Adam optimizer to fit the MPL due to its flexibility, with a learning rate of $5 \times 10^{-3}$ for the index parameters ($\alpha$, $\beta$, and $\gamma$) and $5 \times 10^{-2}$ for the coefficient or constant parameters ($A$, $B$, $C$, and $L_0$). We also perform a second optimization with a learning rate of $1 \times 10^{-5}$ and $1 \times 10^{-6}$, initialized with parameters from the first optimization. Each optimization runs for $5 \times 10^4$ steps, selecting the lowest training loss result. Fitted parameters are listed in Table 7.

Table 7: Parameter values for optimized schedules across different model sizes, rounded to two decimal places.

| Model Size | $A$ | $B$ | $C$ | $\alpha$ | $\beta$ | $\gamma$ | $L_0$ |
|---|---|---|---|---|---|---|---|
| 400M | 0.66 | 614.30 | 0.16 | 0.42 | 0.88 | 0.56 | 2.52 |
| 100M | 0.59 | 521.40 | 0.24 | 0.46 | 0.60 | 0.65 | 2.79 |
| 25M | 0.51 | 446.40 | 2.07 | 0.53 | 0.41 | 0.52 | 3.17 |

In the discussion of Appendix C, we also fit simplified MPL or MPL variants in this manner, except for the momentum law (Appendix G.2). In Figure 15, we present the fitting and prediction results for a subset of experiments, with a zoom-in window highlighting predictions near the end of training. In long-horizon experiments, the zoomed-in view reveals slight discrepancies between the MPL predictions and the actual training curves, targeted for future refinement.

**Momentum Law.** For the momentum law (MTL; Appendix C), $\Theta = \{A, B, \alpha, L_0\}$, with $\lambda$ as a tunable hyperparameter. The input $X_t$ for MTL is the same as MPL's input. Following Tissue et al. (2024), we use L-BFGS to minimize Equation (27), grid-searching $\lambda \in \{0.95, 0.99, 0.995, 0.999, 0.9995\}$ and selecting the best fit based on training accuracy. Predictions are evaluated across the test set (Table 6), with comparisons to MPL in Table 1 and Figure 14. In Figure 14, we compare them specifically over the WSDLD schedule. In the decay stage, MPL not only achieves higher fitting accuracy but also aligns with the curvature of the loss curve. In contrast, MTL fits the stable stage well but predicts a counterfactual concave curve during the decay stage.

**Chinchilla Data Scaling Law.** The Chinchilla Data Scaling Law (CDSL) is similar to the one-power law mentioned in Appendix C, but uses the power of steps instead of the LR sum, with $\Theta = \{A, \alpha, L_0\}$, and $X_t = t$ (final steps only) for Equation (27). The fitting of CDSL follows Hoffmann et al. (2022) and uses the L-BFGS algorithm to minimize the Huber loss. With regard to sample efficiency (Figure 5(a)), CDSL uses cosine curves at 14960, 20080, 27760, 40560, 53360, and 72000 steps, requiring 4.8 times more compute than MPL (two 24000-step curves), with prediction errors of 0.007 (MPL) versus 0.024 (CDSL). MPL achieves less than one-third the prediction error of CDSL. In Figure 5(b), CDSL fits all intermediate steps, ignoring the effect of LR schedule and loss reductions for the comparison with MPL.

**Discussion on the Optimization Method.** We also explored the use of the L-BFGS algorithm for fitting MPL but found it highly sensitive to parameter initialization. For instance, under certain initializations, the fitted parameters may include a high $\beta$ value and a near-zero $C$. Note that $1 - (1 + Cx)^{-\beta} = 1 - \exp(-\beta \log(1 + Cx)) \approx 1 - \exp(-\beta Cx)$ in this case, making MPL resemble a multi-exponential form. In practice, this issue can be mitigated by constraining parameters such as $\beta$ and $\gamma$ to the interval $(0, 1)$. Additionally, we can initialize $C$, $\beta$, and $\gamma$ through grid search to obtain more feasible results. However, using the Adam optimizer is not without limitations, as it lacks theoretical convergence guarantees. Future work will focus on enhancing the fitting process to achieve greater robustness and stability.

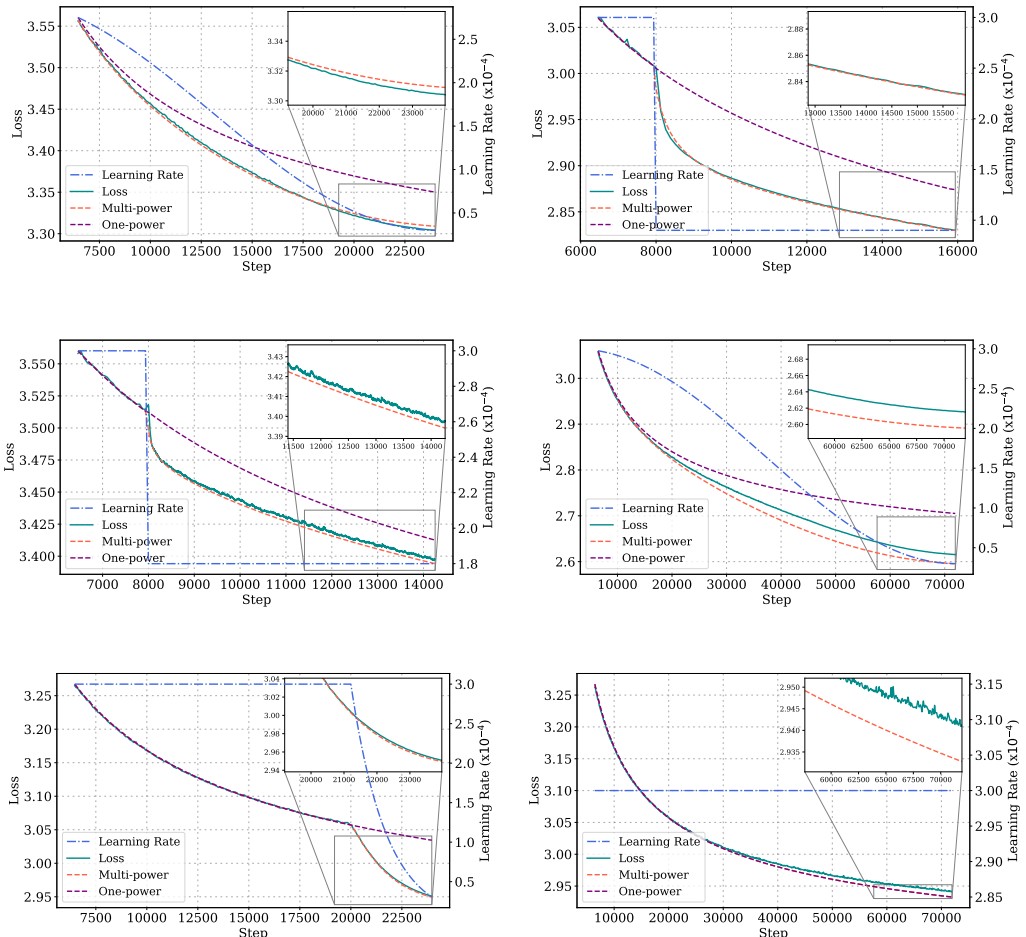

Figure 15: **Fitting and Prediction Details.** Subfigures depict loss curve fitting (training set) and prediction (test set) across various configurations, labeled as $(X, Y)$ for row $X$, column $Y$. The columns in the accompanying table indicate: **F/P** for Fitting (F) or Prediction (P), **Model Size**, **Step Length**, and **Learning Rate Schedule**. Subfigure details follow:

| $(X, Y)$ | F/P | Model Size | Step Length | LR Schedule |
|---|---|---|---|---|
| $(1, 1)$ | F | 25M | 24000 | Cosine |
| $(1, 2)$ | F | 400M | 16000 | 2-stage $(3 \times 10^{-4} \to 9 \times 10^{-5})$ |
| $(2, 1)$ | P | 25M | 16000 | 2-stage $(3 \times 10^{-4} \to 1.8 \times 10^{-4})$ |
| $(2, 2)$ | P | 400M | 72000 | Cosine |
| $(3, 1)$ | P | 100M | 24000 | WSD |
| $(3, 2)$ | P | 100M | 72000 | Constant |

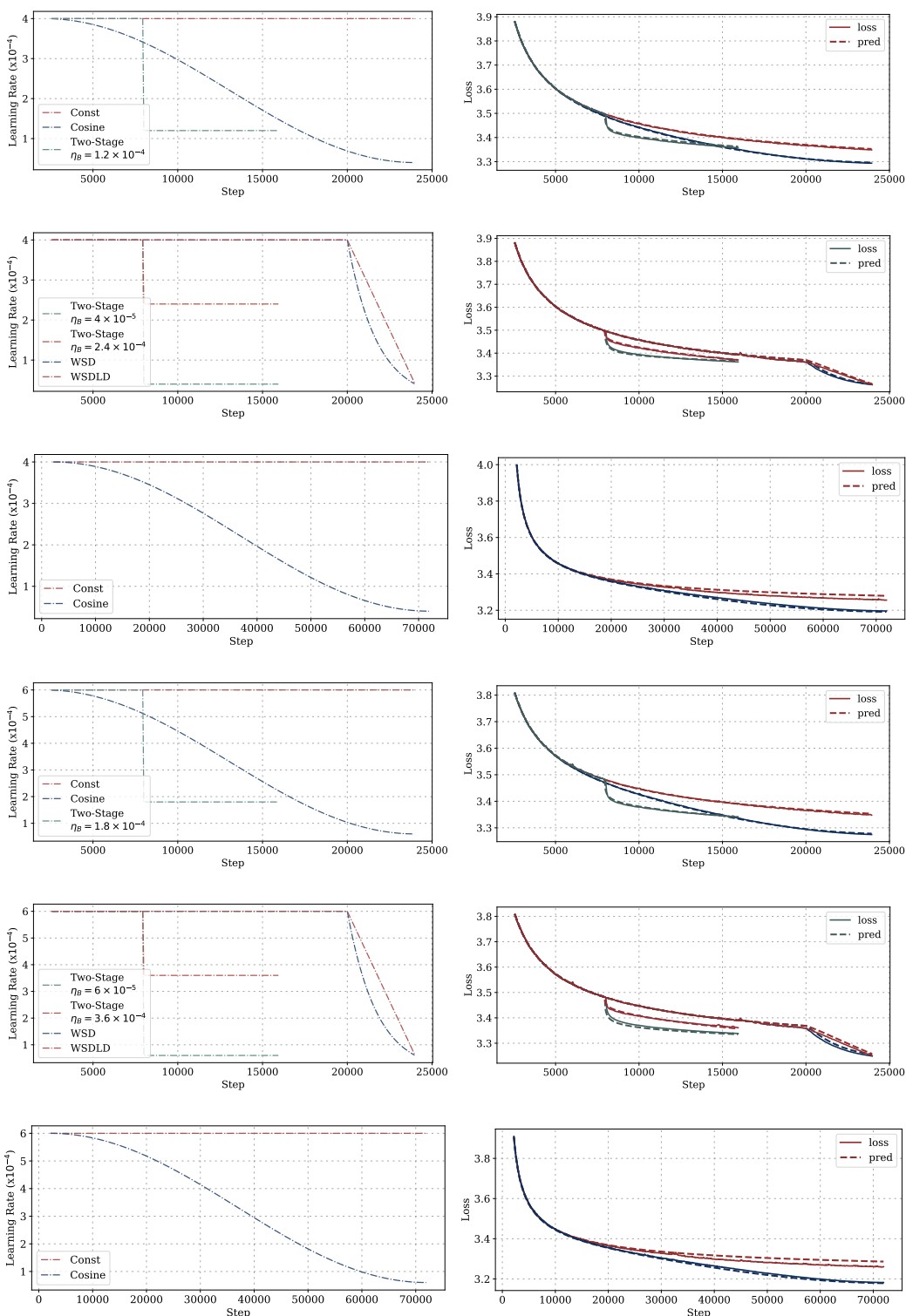

Figure 16: Ablation study on peak learning rates. **Left:** Learning rate schedules; **Right:** Corresponding loss curves. **Layout:** The first three rows show the results for a peak LR of $4 \times 10^{-4}$ while the last three rows are for the peak LR of $6 \times 10^{-4}$. Within each set of the three rows, the first row shows the fitting on the training set, the second row displays the prediction over unseen schedules and the third row demonstrates the extrapolation capability on a long horizon loss curve.

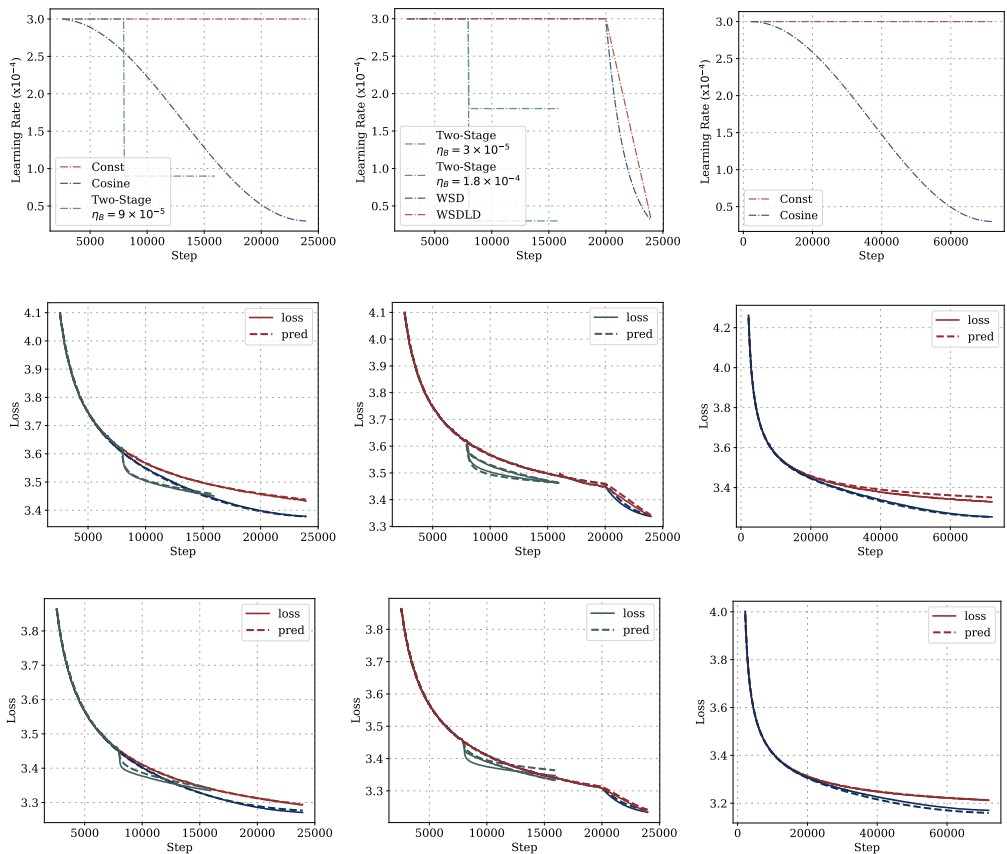

Figure 17: Ablation study on batch sizes, with $R^2$ values of 0.9977 (batch size 64) and 0.9973 (batch size 256). The subfigure layout is as follows. **Rows:** (1) Learning rate schedules, (2) Loss curves for batch size 64, (3) Loss curves for batch size 256. **Columns:** (1) Training set results, (2) Test set results (same horizon as training), (3) Test set results (extended horizon).

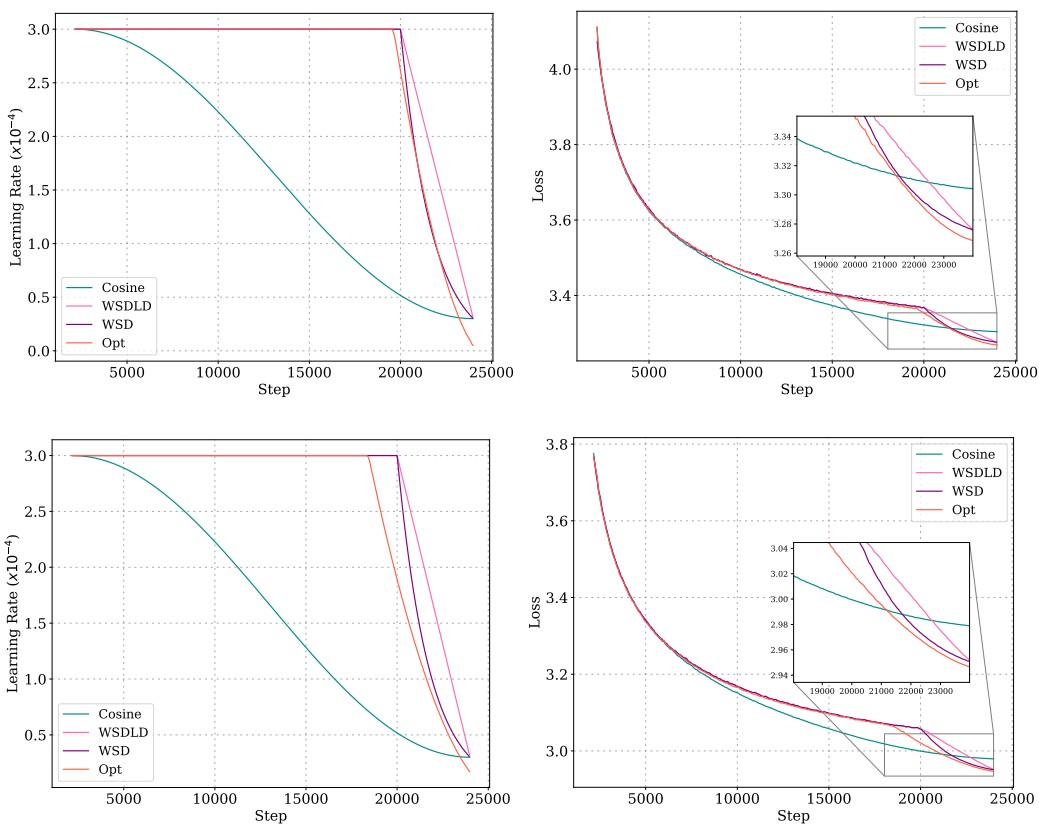

Figure 18: Comparison of our optimized LR schedules and their loss curves with cosine, WSD, and WSDLD schedules over 24000 steps. The decay step for WSD and WSDLD is set to 4000. **Upper:** 25M model; **Lower:** 100M model. **Left:** Learning rates over steps. **Right:** Losses over steps.

## H    DETAILS OF LR SCHEDULE OPTIMIZATION (SECTION 5)

**Optimizing the Surrogate Objective.**    To enhance optimization stability, we redefine the learning rate schedule $E = \{\eta_1, \ldots, \eta_T\}$ using $\Delta = \{\Delta_1, \Delta_2, \ldots, \Delta_T\}$, where $\Delta_t = \eta_{t-1} - \eta_t$ and $\eta_0$ denotes the initial peak LR. Thus, $\eta_t = \eta_0 - \sum_{k=1}^{t} \Delta_k$, establishing a one-to-one mapping between $E$ and $\Delta$. We transform the objective $\mathcal{L}_\Theta(E)$ from (13) into $\tilde{\mathcal{L}}_\Theta(\Delta)$ and alternatively optimize:

$$\min_{\Delta} \quad \tilde{\mathcal{L}}_\Theta(\Delta)$$
$$\text{s.t.} \quad \Delta_t \geq 0, \quad \forall t = 1, \ldots, T,$$
$$\sum_{t=1}^{T} \Delta_t \leq \eta_0.$$

In practice, we enforce these constraints through clipping: after each optimization step, we restrict $\Delta_t$ into $[0, \eta_0]$ and set $\Delta_t = 0$ when $\eta_t \leq \epsilon$, with $\epsilon = 10^{-10}$, to ensure numerical stability. Applied to the MPL fitted from Appendix G.1, this reformulation empirically stabilizes optimization by aligning learning rate reductions with zero initialization. For optimization, we use the Adam optimizer with a constant learning rate, grid searched from $2 \times 10^{-8}$ to $1 \times 10^{-9}$, over 50,000 to 200,000 for better convergence.

**Optimized Schedule of Longer Horizons and Different Model Sizes.**    Beyond Figure 1 and Figure 18, we validate the optimized schedules for extended horizons and different model sizes. For models ranging from 25M to 400M, we optimize LR schedules for 72000-step training based on the MPL fit over the training set. As shown in Figure 19, the resulting schedules exhibit a WSD-like shape, consisting of a stable phase and a decay phase, outperforming cosine schedules across

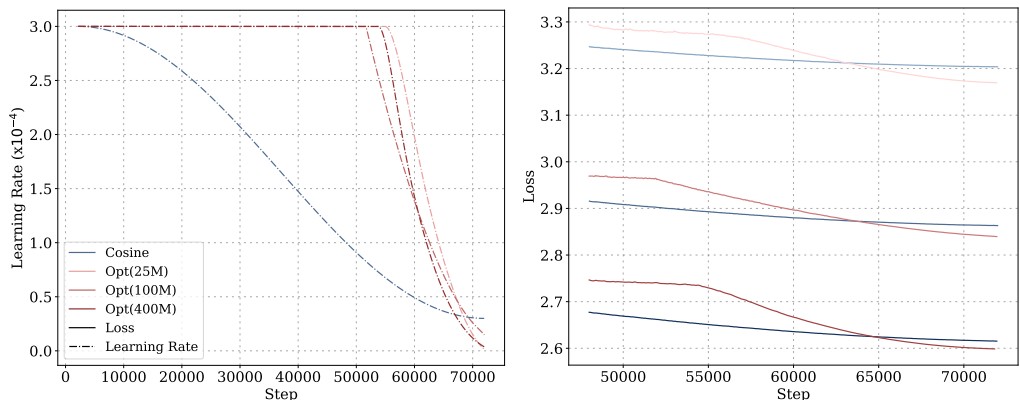

Figure 19: **Left:** Optimized and cosine LR schedules over 72000 steps for models ranging from 25M to 400M. **Right:** Corresponding loss curves for optimized and cosine schedules.

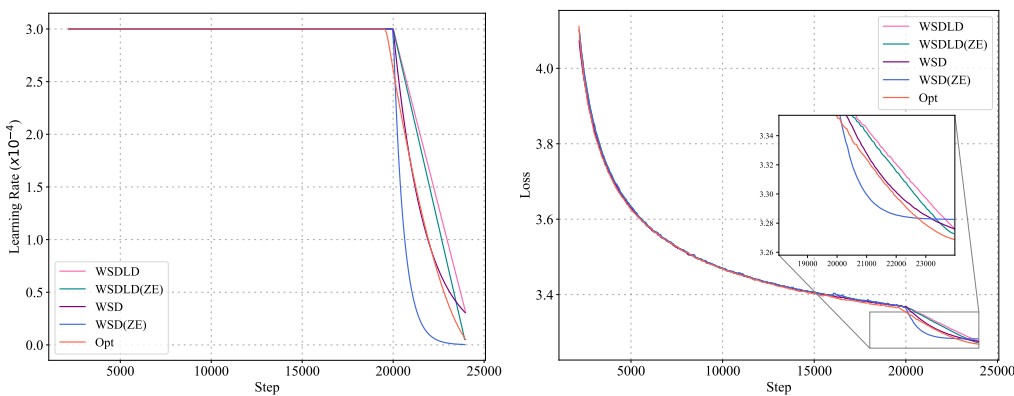

Figure 20: Comparison between the optimized schedules and WSD variants with a near-zero ending LR. WSD (ZE) and WSDLD (ZE) denote WSD and WSDLD schedules with an ending learning rate of $3 \times 10^{-7}$. **Left:** Learning rate comparison. **Right:** Loss comparison.

sizes. For the 1B model, we derive a 72000-step schedule based on the MPL fitted from 24000-step constant and cosine schedule curves, with results in Figure 6(b) confirming superiority over the cosine schedule. Additionally, for the 1B model, we evaluate the downstream performance of the MPL-induced schedule against the cosine schedule across several tasks, including LAMBADA (Paperno et al., 2016), HellaSwag (Zellers et al., 2019), PIQA (Bisk et al., 2020), ARC-easy (Gu & Dao, 2023; Clark et al., 2018), $C^3$ (Sun et al., 2020), and RTE (Wang et al., 2019). The MPL-induced schedule achieves an average score improvement of 1.03 compared to the cosine schedule, as shown in Table 2. This highlights the effectiveness of the MPL-induced schedule in enhancing model performance across diverse downstream tasks.

**Zero-Ending Learning Rate Experiments.** Optimized schedules consistently outperform WSD variants with near-zero learning rates ($3 \times 10^{-7}$, approximately 1/100 of the default setting). To test if a higher ending LR (e.g., 1/10 peak LR) degrades baseline performance, we compare the optimized schedules against WSD(LD) variants with near-zero ending learning rates and the original ones. As shown in Figure 20, the optimized schedule still outperforms these WSD variants. In addition, lower ending learning rates do not consistently improve the final loss (e.g., zero-ending WSD exceeds baseline loss), suggesting a complex interaction between the ending learning rate and the decay function. This highlights the advantage of the optimized schedule in reducing the need for extensive hyperparameter tuning.

**WSD with Sqrt-Cube Decay (WSDSC).** We derive the decay function for optimized schedules by analyzing the decay phase across Llama2 models ranging from 25M to 400M. We compute

normalized steps and learning rates (LRs) within the decay phase for schedules of varying step counts and model sizes. After averaging the normalized LRs, we perform symbolic regression against the normalized steps and approximate the decay function as $f(t - T_{\text{stable}}) = \left( \frac{T_{\text{total}} - t}{T_{\text{total}} - T_{\text{stable}}} \right)^{1.5}$. Validation experiments on 1B Llama2 and GPT models confirm its efficacy: Figure 6(b) shows that WSDSC outperforms the cosine schedule for the 1B model, though it falls short of the MPL-optimized schedule. Figure 13(a) demonstrates WSDSC's superiority over both the standard WSD and cosine schedules for GPT.

# I OPTIMAL LEARNING RATE SCHEDULE FOR MOMENTUM LAW

In this section, we derive the optimal learning rate schedules for the Momentum Law (Tissue et al., 2024):

$$L(T) = L_0 + A \cdot S_1^{-\alpha} - C \cdot S_2,$$

where $S_1 = \sum_{t=1}^{T} \eta_t$ and $S_2 = \sum_{t=1}^{T} \sum_{k=1}^{t} (\eta_{k-1} - \eta_k) \cdot \lambda^{t-k}$. $\lambda$ is a hyperparameter typically ranges from $0.99$ to $0.999$, and $L_0, A, C > 0$ are parameters.

Similar to Section 5, here we could also optimize this law to get a learning rate schedule achieving lowest final loss by solving

$$\min_{\eta_1, \eta_2, \ldots, \eta_T} \quad L_\Xi(\eta_1, \eta_2, \ldots, \eta_T) \tag{A}$$

$$\text{s.t.} \quad 0 \leq \eta_t \leq \eta_{t-1}, \ \forall 1 \leq t \leq T,$$

where $\Xi = \{L_0, A, C, \lambda\}$ represents the hyperparameters and parameters in $L(T)$. For simplicity of derivation, we introduce $\eta_0$ in front of $E$ as the maximal LR. Compared with our Multi-Power Law (MPL), this optimization problem is obviously convex, so we can characterize its minimizer more easily in math. In our result, the Momentum Law yields optimal schedules that go through a stable phase at peak LR and then directly drop to zero LR in no more than two steps. However, these kinds of schedules are clearly far from the optimal schedules in practice. As a comparison, in Section 5, MPL can induce a WSD-like schedule, which is empirically effective.

Next, we formalize the above arguments mathematically.

**Theorem 2.** *For any LR schedule $E^* := \{\eta_1^*, \ldots, \eta_T^*\}$ that minimizes the optimization problem* (A), *there exists $k \in \{0, 1, \ldots, T\}$ such that the following holds for all $t \in \{1, \ldots, T\}$:*

1. *If $t \leq k$, then $\eta_t^* = \eta_0$;*

2. *If $t \geq k + 2$, then $\eta_t^* = 0$.*

For convenience, we first prove the following lemma.

**Lemma 1.** *For a function $f(x) = x - M(1 - \lambda^x)$ with $M > 0$ and $0 < \lambda < 1$, we have the following properties:*

1. *$f(x)$ is strictly convex and has a unique minimizer over $x \in [0, \infty)$.*

2. *If $f(y) \geq 0$ for some $y \in [0, \infty)$, then $f(x) \geq f(y)$ for all $x \in [y, \infty)$.*

*Proof.* First, it is easy to check

$$f(0) = 0, \ \frac{df}{dx}(x) = 1 + M\lambda^x \log \lambda, \ \frac{d^2 f}{dx^2}(x) = M\lambda^x (\log \lambda)^2 > 0.$$

Therefore, $f(x)$ is strictly convex. Then we can discuss the property of $f(x)$ over $x \in (0, \infty)$ by discussing $\frac{df}{dx}(0)$.

(1) When $\frac{df}{dx}(0) \geq 0$, $\frac{df}{dx}(x') > \frac{df}{dx}(0) \geq 0$ for all $x' \in (0, x)$. Thus, $f(x)$ is monotonically increasing over $(0, x)$ and $f(x) > f(0) = 0$. So $x = 0$ is the unique minimizer over $x \in [0, \infty)$ and $f(x) \geq f(y)$ when $x \geq y \geq 0$.

(2) When $\frac{df}{dx}(0) < 0$, $\lim_{x \to \infty} \frac{df}{dx}(x) = 1$. Then there exists $x^* \in (0, \infty)$ such that $\frac{df}{dx}(x^*) = 0$. Thus, $f(x)$ monotonically decreases over $(0, x^*)$ and monotonically increases over $(x^*, \infty)$. Hence $x^*$ is the unique minimizer over $x \in [0, \infty)$. Moreover, $f(x^*) < f(0) = 0$ and $\lim_{x \to \infty} f(x) = \infty$, so there exists $\tilde{x} \in (x^*, \infty)$, such that $f(x) < 0$ over $(0, \tilde{x})$ and $f(x) > 0$ over $(\tilde{x}, \infty)$. Clearly, $f(x)$ monotonically increases over $x \in [\tilde{x}, \infty)$. Therefore, if $f(y) \geq 0$ for some $y \in [0, \infty)$, then $y \geq \tilde{x}$ and $f(x) \geq f(y)$ for all $x \in [y, \infty)$.

This completes the proof. □

Next, we prove Theorem 2.

*Proof for Theorem 2.* First, we reparameterize $\eta_t$ as $\eta_t = \eta_0 - \sum_{k=1}^{t} \Delta_k$, then the optimization problem (A) becomes

$$\min_{\Delta_1,\Delta_2,\ldots,\Delta_T} \quad \hat{L}_\Xi(\Delta_1, \Delta_2, \ldots, \Delta_T)$$

$$\text{s.t.} \quad \Delta_t \geq 0, \quad \forall 1 \leq t \leq T,$$

$$\sum_{i=1}^{T} \Delta_i \leq \eta_0,$$

where $\hat{L}_\Xi(\Delta_1, \Delta_2, \ldots, \Delta_T)$ is given by

$$\hat{L}_\Xi(\Delta_1, \Delta_2, \ldots, \Delta_T) = L_0 + A \cdot \left(T\eta_0 - \sum_{t=1}^{T}\sum_{k=1}^{t} \Delta_k\right)^{-\alpha} - C \cdot \sum_{t=1}^{T}\sum_{k=1}^{t} \Delta_k \lambda^{t-k}.$$

Define the Lagrangian by

$$\mathcal{L}(\Delta, \lambda, \mu) = \hat{L}_\Xi(\Delta_1, \ldots, \Delta_T) - \sum_{t=1}^{T} \lambda_t \Delta_t + \mu\left(\sum_{t=1}^{T} \Delta_t - \eta_0\right),$$

where $\lambda_1, \ldots, \lambda_T$ and $\mu$ are the Lagrange multipliers associated with the constraints $\Delta_t \geq 0$ and $\sum_{i=1}^{T} \Delta_t \leq \eta_0$, respectively. By Karush-Kuhn-Tucker (KKT) conditions, there exist $\lambda_1, \ldots, \lambda_T \geq 0$ and $\mu \geq 0$ such that the following conditions hold:

- Complementary Slackness: $\lambda_t \Delta_t = 0$ for all $t = 1, \ldots, T$ and $\mu\left(\sum_{t=1}^{T} \Delta_t - \eta_0\right) = 0$.

- Stationary: $\frac{\partial \hat{L}_\Xi}{\partial \Delta_t}(\Delta_1, \ldots, \Delta_T) - \lambda_t + \mu = 0$ for all $t = 1, \ldots, T$.

Here, we have

$$\frac{\partial \hat{L}_\Xi}{\partial \Delta_t} = \alpha A \Phi^{-\alpha-1} \cdot (T - t + 1) - C \cdot (\lambda^0 + \lambda^1 + \cdots + \lambda^{T-t})$$

$$= \alpha A \Phi^{-\alpha-1} \cdot (T - t + 1) - C \cdot \frac{1 - \lambda^{T-t+1}}{1 - \lambda}$$

$$= Kf(T - t + 1),$$

where $\Phi := T\eta_0 - \sum_{t=1}^{T}\sum_{k=1}^{t} \Delta_k$, $K := \alpha A \Phi^{-\alpha-1} > 0$, and $f(x) := x - M(1 - \lambda^x)$ with $M := \frac{C}{(1-\lambda)K} > 0$.

Note that $\Phi$ does not depend on $t$. We can rewrite the stationary condition as

$$\lambda_t = Kf(T - t + 1) + \mu.$$

By Lemma 1, $f(x)$ is strictly convex and has a unique minimizer over $x \in [0, \infty)$. Let $x^*$ be this unique minimizer. Let $f_{\min} := \min_{t \in \{1, \ldots, T\}}\{f(T - t + 1)\}$ be the minimum value of $f(T - t + 1)$, and $S$ be the set of indices that minimize $f(T - t + 1)$. Then $|S| \leq 2$ and $S \subseteq \{\lfloor T - x^* + 1 \rfloor, \lceil T - x^* + 1 \rceil\}$.

Now we discuss the following two cases by the value of $Kf_{\min} + \mu$.

**Case 1.** If $Kf_{\min} + \mu > 0$, then $\lambda_t > 0$ for all $t = 1, \ldots, T$. By the complementary slackness condition, $\Delta_t = 0$ for all $t = 1, \ldots, T$. This implies that $E^*$ is a constant schedule, $\eta_0 = \eta_1^* = \eta_2^* = \cdots = \eta_T^*$.

**Case 2.** If $Kf_{\min} + \mu = 0$, then $\lambda_t = 0$ for all $t \in S$ and $\lambda_t > 0$ for all $t \notin S$. By the complementary slackness condition, the latter implies that $\Delta_t = 0$ for all $t \notin S$. Then $E^*$ falls into one of the two categories:

1. If $S = \{s\}$ for some $s$, then $\eta_0 = \eta_1^* = \eta_2^* = \cdots = \eta_{s-1}^*$ and $\eta_s^* = \eta_{s+1}^* = \cdots = \eta_T^*$;

2. If $S = \{s - 1, s\}$ for some $s$, then $\eta_0 = \eta_1^* = \eta_2^* = \cdots = \eta_{s-2}^*$ and $\eta_s^* = \eta_{s+1}^* = \cdots = \eta_T^*$.

We claim that $\eta_T^* = 0$ if $s < T$. If not, then we have $\mu = 0$ due to the complementary slackness condition. Moreover, $s < T$ implies $T \notin S$, and then we have $0 < \lambda_T = K f(1) + \mu = K f(1)$. By Lemma 1, $f(x) \geq f(1) > 0$ for all $x \geq 1$, which implies that $\lambda_t = K f(T - t + 1) + \mu = K f(T - t + 1) > 0$ for all $t = 1, \ldots, T$, which contradicts the fact that $\lambda_t = 0$ for all $t \in S$.

Putting all these together, we conclude that $E^*$ must exhibit the pattern described in the theorem. $\square$

## J  PROOF OF THEOREM 1

The proof of Theorem 1 consists of two main parts. First, we explicitly derive the formula for any quadratic loss function after $t$ steps, without making Assumption 1. Then, we take the expectation over $\Sigma$, $H$, and the initialization $\boldsymbol{\theta}_0$ to prove the theorem.

WLOG, we assume that $\boldsymbol{H} = \mathrm{diag}(\lambda_1, \ldots, \lambda_d)$, and set $\boldsymbol{\theta}_* = 0$. Otherwise, due to the rotational and translational invariance of SGD, we can always transform the coordinate system so that the eigenbasis of $\boldsymbol{H}$ is the standard basis, and the optimal solution is at the origin. In this case, $\Sigma_{ii}$ is just the $i$-th diagonal entry of $\Sigma$ in the standard basis.

### J.1  GENERAL THEOREM FOR ALL QUADRATIC LOSS FUNCTIONS

In the first part, we establish the following theorem that characterizes the expected loss. We use $\Phi(\boldsymbol{\theta}_0, E)$ to denote the distribution of the $T$-th iteration $\boldsymbol{\theta}_T$ of SGD with initialization $\boldsymbol{\theta}_0$ and LR schedule $E := \{\eta_1, \ldots, \eta_T\}$.

**Theorem 3.** *For $\boldsymbol{\theta}_T \sim \Phi(\boldsymbol{\theta}_0, E)$ and any fixed $\eta_0 > 0$, we have the following estimate of $\mathbb{E}[\mathcal{L}(\boldsymbol{\theta}_T)]$:*

$$M(\boldsymbol{\theta}_0, E) := \frac{1}{2}\sum_{i=1}^{d}\left(\theta_{0,i}^2 \lambda_i \exp(-2\lambda_i S_1) + \eta_0 \Sigma_{ii} \cdot \frac{1 - \exp(-2\lambda_i S_1)}{2}\right)$$

$$- \frac{1}{2}\sum_{k=1}^{T}(\eta_{k-1} - \eta_k)\sum_{i=1}^{d}\Sigma_{ii}\frac{1 - \exp(-2\lambda_i S_k)}{2},$$

*where $S_k := \sum_{\tau=k}^{T}\eta_\tau$. The estimation error is bounded as*

$$|\mathbb{E}[\mathcal{L}(\boldsymbol{\theta}_T)] - M(\boldsymbol{\theta}_0, E)| \leq 5\eta_{\max}\sum_{i=1}^{d}\lambda_i^3 S_1 \exp(-2\lambda_i S_1)\theta_{0,i}^2 + \frac{15}{2}\eta_{\max}^2\sum_{i=1}^{d}\Sigma_{ii}\lambda_i,$$

*where $\eta_{\max} := \max_{0 \leq k \leq T}\eta_k$.*

To prove the theorem, we first introduce some notations and auxiliary expectations. We define

$$U(\boldsymbol{\theta}, \eta, S) := \frac{1}{2}\sum_{i=1}^{d}\left(\theta_i^2 \lambda_i \exp(-2\lambda_i S) + \eta\Sigma_{ii} \cdot \frac{1 - \exp(-2\lambda_i S)}{2}\right).$$

We decompose the expected loss $\mathbb{E}_{\boldsymbol{\theta}_T \sim \Phi(\boldsymbol{\theta}_0, E)}[\mathcal{L}(\boldsymbol{\theta}_T)]$ into a telescoping sum of $T+1$ auxiliary expectations $A_0, A_1, \ldots, A_T$:

$$\mathbb{E}_{\boldsymbol{\theta}_T \sim \Phi(\boldsymbol{\theta}_0, E)}[\mathcal{L}(\boldsymbol{\theta}_T)] = A_0 + \sum_{k=1}^{T}(A_k - A_{k-1}), \tag{28}$$

$$A_k := \mathbb{E}_{\boldsymbol{\theta}_k \sim \Phi(\boldsymbol{\theta}_0, E_{\leq k})}[U(\boldsymbol{\theta}_k, \eta_k, S_{k+1})],$$

where we define $S_{T+1} = 0$, so $A_T = \mathbb{E}_{\boldsymbol{\theta}_T \sim \Phi(\boldsymbol{\theta}_0, E)}[\mathcal{L}(\boldsymbol{\theta}_T)]$.

The above theorem needs the following two lemmas.

**Lemma 2.** *If $x \in [0, 1]$, then*

$$\exists \xi_1 \in [-10, 0] \quad s.t. \quad (1 - x)^2 = \exp(-2x)(1 + \xi_1 x^2),$$

$$\exists \xi_2 \in [-10, 0] \quad s.t. \quad (1 - 2x) = \exp(-2x)(1 + \xi_2 x^2).$$

*Proof.* The above inequalities hold for $x = 0$. For the case of $x \in (0, 1]$, we must have

$$\xi_1 = -\frac{1 - (1 - x)^2 \exp(2x)}{x^2}, \quad \xi_2 = -\frac{1 - (1 - 2x)\exp(2x)}{x^2}.$$

So it suffices to show that both $\frac{1 - (1-x)^2\exp(2x)}{x^2}$ and $\frac{1 - (1-2x)\exp(2x)}{x^2}$ lie in $[0, 10]$. Noting that $1 - 2x \leq (1 - x)^2 \leq \exp(-2x)$, we obtain the following lower bounds:

$$\frac{1 - (1 - 2x)\exp(2x)}{x^2} \geq \frac{1 - (1 - x)^2\exp(2x)}{x^2} \geq \frac{1 - \exp(-2x)\exp(2x)}{x^2} = 0.$$

Also note that $\frac{1 - (1-2x)\exp(2x)}{x^2}$ is an increasing function of $x$. So we have

$$\frac{1 - (1 - 2x)\exp(2x)}{x^2} \leq \frac{1 - (-1) \cdot \exp(2)}{1^2} \leq 10.$$

Putting these two inequalities together, we have

$$10 \geq \frac{1 - (1 - 2x)\exp(2x)}{x^2} \geq \frac{1 - (1 - x)^2 \exp(2x)}{x^2} \geq 0,$$

which completes the proof. □

**Lemma 3.** *If $\eta_{\max} \leq \frac{1}{\lambda_{\max}}$, then for all $k \in [T]$,*

$$\sum_{t=1}^{k-1} \eta_t \exp(-2\lambda_i S_t) \leq \frac{1}{2\lambda_i} \exp(-2\lambda_i S_k)$$

$$\sum_{t=1}^{k-1} \eta_t \exp(-2\lambda_i S_{t+1}) \leq \frac{4}{\lambda_i} \exp(-2\lambda_i S_k).$$

*Proof.* The first inequality follows from the fact that a lower Darboux sum is smaller than the corresponding Darboux integral

$$\sum_{t=1}^{k-1} \eta_t \exp(-2\lambda_i S_t) = \sum_{t=1}^{k-1} (S_t - S_{t+1})\exp(-2\lambda_i S_t)$$

$$\leq \int_{S_k}^{S_1} \exp(-2\lambda_i S)\mathrm{d}S$$

$$= \frac{1}{2\lambda_i}\left[\exp(-2\lambda_i S_k) - \exp(-2\lambda_i S_1)\right]$$

$$\leq \frac{1}{2\lambda_i}\exp(-2\lambda_i S_k).$$

For the second inequality,

$$\sum_{t=1}^{k-1} \eta_t \exp(-2\lambda_i S_{t+1}) = \sum_{t=1}^{k-1} \eta_t \exp(-2\lambda_i S_t)\exp(2\lambda_i \eta_t)$$

$$\leq \sum_{t=1}^{k-1} \eta_t \exp(-2\lambda_i S_t)\exp(2)$$

$$\leq \frac{\exp(2)}{2\lambda_i}\exp(-2\lambda_i S_k)$$

$$\leq \frac{4}{\lambda_i}\exp(-2\lambda_i S_k),$$

which completes the proof. □

The following lemma characterizes the difference between two consecutive auxiliary expectations $A_k$ and $A_{k-1}$.

**Lemma 4.** *If $\eta_{\max} \leq \frac{1}{\lambda_{\max}}$, then for all $k \in [T]$,*

$$A_k - A_{k-1} = -\frac{1}{2}(\eta_{k-1} - \eta_k)\sum_{i=1}^{d} \frac{1 - \exp(-2\lambda_i S_k)}{2}\Sigma_{ii} + \epsilon_k,$$

*where the error term $\epsilon_k$ is bounded by*

$$|\epsilon_k| \leq 5\sum_{i=1}^{d} \eta_k^2 \lambda_i^3 \exp(-2\lambda_i S_k)\mathbb{E}_{\boldsymbol{\theta}_{k-1}\sim\Phi(\boldsymbol{\theta}_0, E_{\leq k-1})}[\theta_{k-1,i}^2] + 5\sum_{i=1}^{d} \eta_k^3 \Sigma_{ii}\lambda_i^2 \exp(-2\lambda_i S_k).$$

*Proof.* By the definition of $A_k$ and $A_{k-1}$, we have

$$A_k - A_{k-1} = \mathbb{E}_{\boldsymbol{\theta}_k\sim\Phi(\boldsymbol{\theta}_0, E_{\leq k})}[U(\boldsymbol{\theta}_k, \eta_k, S_{k+1})] - \mathbb{E}_{\boldsymbol{\theta}_{k-1}\sim\Phi(\boldsymbol{\theta}_0, E_{\leq k-1})}[U(\boldsymbol{\theta}_{k-1}, \eta_{k-1}, S_k)]$$

$$= \mathbb{E}_{\boldsymbol{\theta}_{k-1}\sim\Phi(\boldsymbol{\theta}_0, E_{\leq k-1})}[\Delta\bar{U}(\boldsymbol{\theta}_{k-1})],$$

where

$$\Delta\bar{U}(\boldsymbol{\theta}_{k-1}) := \underbrace{\mathbb{E}_{\boldsymbol{g}_k\sim\mathcal{N}(\boldsymbol{H}\boldsymbol{\theta}_{k-1}, \boldsymbol{\Sigma})}[U(\boldsymbol{\theta}_{k-1} - \eta_k \boldsymbol{g}_k, \eta_k, S_{k+1}) \mid \boldsymbol{\theta}_{k-1}]}_{=: \bar{U}(\boldsymbol{\theta}_{k-1})} - U(\boldsymbol{\theta}_{k-1}, \eta_{k-1}, S_k).$$

Expanding $\bar{U}(\boldsymbol{\theta}_{k-1})$ based on the definition of $U$ gives

$$\bar{U}(\boldsymbol{\theta}_{k-1}) = \underbrace{\mathbb{E}_{\boldsymbol{g}_k \sim \mathcal{N}(\boldsymbol{H}\boldsymbol{\theta}_{k-1}, \boldsymbol{\Sigma})} \left[ \frac{1}{2} \sum_{i=1}^{d} (\theta_{k-1,i} - \eta_k g_{k,i})^2 \lambda_i \exp(-2\lambda_i S_{k+1}) \,\middle|\, \boldsymbol{\theta}_{k-1} \right]}_{=:\bar{U}_1(\boldsymbol{\theta}_{k-1})}$$

$$+ \underbrace{\frac{1}{2} \sum_{i=1}^{d} \eta_k \Sigma_{ii} \cdot \frac{1 - \exp(-2\lambda_i S_{k+1})}{2}}_{=:\bar{U}_2(\boldsymbol{\theta}_{k-1})} \,.$$

For $\bar{U}_1(\boldsymbol{\theta}_{k-1})$, evaluating the expectation gives

$$\bar{U}_1(\boldsymbol{\theta}_{k-1}) = \frac{1}{2} \sum_{i=1}^{d} \left( \lambda_i \exp(-2\lambda_i S_{k+1}) \left( (1 - \eta_k \lambda_i)^2 \theta_{k-1,i}^2 + \eta_k^2 \Sigma_{ii} \right) \right),$$

Then, we split $\bar{U}_1(\boldsymbol{\theta}_{k-1})$ into two parts:

$$\bar{U}_1(\boldsymbol{\theta}_{k-1}) = \underbrace{\frac{1}{2} \sum_{i=1}^{d} \lambda_i \exp(-2\lambda_i S_{k+1})(1 - \eta_k \lambda_i)^2 \theta_{k-1,i}^2}_{=:\bar{U}_{11}(\boldsymbol{\theta}_{k-1})} + \underbrace{\frac{1}{2} \sum_{i=1}^{d} \lambda_i \exp(-2\lambda_i S_{k+1}) \eta_k^2 \Sigma_{ii}}_{=:\bar{U}_{12}(\boldsymbol{\theta}_{k-1})} \,.$$

Let $\bar{U}_3(\boldsymbol{\theta}_{k-1}) := \bar{U}_{12}(\boldsymbol{\theta}_{k-1}) + \bar{U}_2(\boldsymbol{\theta}_{k-1})$. Then $\bar{U}(\boldsymbol{\theta}_{k-1}) = \bar{U}_{11}(\boldsymbol{\theta}_{k-1}) + \bar{U}_3(\boldsymbol{\theta}_{k-1})$. We can rewrite $\bar{U}_3(\boldsymbol{\theta}_{k-1})$ as

$$\bar{U}_3(\boldsymbol{\theta}_{k-1}) = \frac{1}{2} \sum_{i=1}^{d} \left( \eta_k \Sigma_{ii} \cdot \frac{1 - \exp(-2\lambda_i S_k)(1 - 2\eta_k \lambda_i)}{2} \right).$$

Since $\eta_k \lambda_i \in [0, 1]$ for all $i$, by Lemma 2, we can find $\xi_{1,i}, \xi_{2,i} \in [-10, 0]$ such that
$$(1 - \eta_k \lambda_i)^2 = \exp(-2\eta_k \lambda_i)(1 + \xi_{1,i} \eta_k^2 \lambda_i^2), \quad (1 - 2\eta_k \lambda_i) = \exp(-2\eta_k \lambda_i)(1 + \xi_{2,i} \eta_k^2 \lambda_i^2).$$
Then we can rewrite $\bar{U}_{11}(\boldsymbol{\theta}_{k-1})$ as

$$\bar{U}_{11}(\boldsymbol{\theta}_{k-1}) = \frac{1}{2} \sum_{i=1}^{d} (1 + \xi_{1,i} \eta_k^2 \lambda_i^2) \lambda_i \exp(-2\lambda_i S_k) \theta_{k-1,i}^2$$

$$= \frac{1}{2} \sum_{i=1}^{d} \lambda_i \exp(-2\lambda_i S_k) \theta_{k-1,i}^2 + \frac{1}{2} \sum_{i=1}^{d} \xi_{1,i} \eta_k^2 \lambda_i^3 \exp(-2\lambda_i S_k) \theta_{k-1,i}^2.$$

Similarly, we can rewrite $\bar{U}_3(\boldsymbol{\theta}_{k-1})$ as

$$\bar{U}_3(\boldsymbol{\theta}_{k-1}) = \frac{1}{2} \sum_{i=1}^{d} \left( \eta_k \Sigma_{ii} \cdot \frac{1 - (1 + \xi_{2,i} \eta_k^2 \lambda_i^2) \exp(-2\lambda_i S_k)}{2} \right)$$

$$= \frac{1}{2} \sum_{i=1}^{d} \left( \eta_k \Sigma_{ii} \cdot \frac{1 - \exp(-2\lambda_i S_k)}{2} \right) - \frac{1}{2} \sum_{i=1}^{d} \xi_{2,i} \eta_k^3 \Sigma_{ii} \lambda_i^2 \exp(-2\lambda_i S_k).$$

Therefore, we can rewrite $\bar{U}(\boldsymbol{\theta}_{k-1})$ as

$$\bar{U}(\boldsymbol{\theta}_{k-1}) = \frac{1}{2} \sum_{i=1}^{d} \left( \lambda_i \exp(-2\lambda_i S_k) \theta_{k-1,i}^2 + \eta_k \Sigma_{ii} \cdot \frac{1 - \exp(-2\lambda_i S_k)}{2} \right) + \tilde{\epsilon}_k(\boldsymbol{\theta}_{k-1}),$$

where

$$\tilde{\epsilon}_k(\boldsymbol{\theta}_{k-1}) := \frac{1}{2} \sum_{i=1}^{d} \xi_{1,i} \eta_k^2 \lambda_i^3 \exp(-2\lambda_i S_k) \theta_{k-1,i}^2 - \frac{1}{2} \sum_{i=1}^{d} \xi_{2,i} \eta_k^3 \Sigma_{ii} \lambda_i^2 \exp(-2\lambda_i S_k).$$

Subtracting $U(\boldsymbol{\theta}_{k-1}, \eta_{k-1}, S_k)$ from the above expression, we can obtain the following formula for $\Delta \bar{U}(\boldsymbol{\theta}_{k-1}) := \bar{U}(\boldsymbol{\theta}_{k-1}) - U(\boldsymbol{\theta}_{k-1}, \eta_{k-1}, S_k)$,

$$\Delta \bar{U}(\boldsymbol{\theta}_{k-1}) = -\frac{1}{2}(\eta_{k-1} - \eta_k) \sum_{i=1}^{d} \frac{1 - \exp(-2\lambda_i S_k)}{2} \Sigma_{ii} + \tilde{\epsilon}_k(\boldsymbol{\theta}_{k-1}).$$

Taking the expectation of $\Delta \bar{U}(\boldsymbol{\theta}_{k-1})$ over $\boldsymbol{\theta}_{k-1} \sim \Phi(\boldsymbol{\theta}_0, E_{\leq k-1})$, we have

$$A_k - A_{k-1} = \mathbb{E}_{\boldsymbol{\theta}_{k-1} \sim \Phi(\boldsymbol{\theta}_0, E_{\leq k-1})}[\Delta \bar{U}(\boldsymbol{\theta}_{k-1})]$$

$$= -\frac{1}{2}(\eta_{k-1} - \eta_k) \sum_{i=1}^{d} \frac{1 - \exp(-2\lambda_i S_k)}{2} \Sigma_{ii} + \mathbb{E}_{\boldsymbol{\theta}_{k-1} \sim \Phi(\boldsymbol{\theta}_0, E_{\leq k-1})}[\tilde{\epsilon}_k(\boldsymbol{\theta}_{k-1})].$$

Letting $\epsilon_k := \mathbb{E}_{\boldsymbol{\theta}_{k-1} \sim \Phi(\boldsymbol{\theta}_0, E_{\leq k-1})}[\tilde{\epsilon}_k(\boldsymbol{\theta}_{k-1})]$ completes the proof. $\qquad\square$

The following lemma gives an upper bound for the term $\mathbb{E}_{\boldsymbol{\theta}_{k-1} \sim \Phi(\boldsymbol{\theta}_0, E_{\leq k-1})}[\theta_{k-1,i}^2]$ that appears in Lemma 4.

**Lemma 5.** *If $\eta_{\max} \leq \frac{1}{\lambda_{\max}}$, then for all $k \in [T]$ and $i \in [d]$,*

$$\mathbb{E}_{\boldsymbol{\theta}_{k-1} \sim \Phi(\boldsymbol{\theta}_0, E_{\leq k-1})}[\theta_{k-1,i}^2] \leq \theta_{0,i}^2 \exp(-2\lambda_i(S_1 - S_k)) + \frac{4}{\lambda_i} \eta_{\max} \Sigma_{ii}.$$

*Proof.* By the update rule, for all $1 \leq t \leq k-1$, we have

$$\mathbb{E}[\theta_{t,i}^2] = (1 - \eta_t \lambda_i)^2 \mathbb{E}[\theta_{t-1,i}^2] + \eta_t^2 \Sigma_{ii}.$$

Since $(1 - \eta_t \lambda_i)^2 \leq \exp(-2\eta_t \lambda_i)$ and $\eta_t \leq \eta_{\max}$, we have the following bound:

$$\mathbb{E}[\theta_{t,i}^2] \leq \exp(-2\eta_t \lambda_i) \mathbb{E}[\theta_{t-1,i}^2] + \eta_t \eta_{\max} \Sigma_{ii}.$$

Expanding the recursion, we have

$$\mathbb{E}[\theta_{k-1,i}^2] \leq \theta_{0,i}^2 \exp(-2\lambda_i(S_1 - S_k)) + \sum_{t=1}^{k-1} \eta_t \eta_{\max} \Sigma_{ii} \exp(-2\lambda_i(S_{t+1} - S_k))$$

$$= \theta_{0,i}^2 \exp(-2\lambda_i(S_1 - S_k)) + \exp(2\lambda_i S_k) \eta_{\max} \Sigma_{ii} \sum_{t=1}^{k-1} \eta_t \exp(-2\lambda_i S_{t+1}),$$

where the first line uses the identity $\prod_{\tau=t+1}^{k-1} \exp(-2\eta_\tau \lambda_i) = \exp(-2\lambda_i(S_{t+1} - S_k))$.

Further, by Lemma 3, we have $\sum_{t=1}^{k-1} \eta_t \exp(-2\lambda_i S_{t+1}) \leq \frac{4}{\lambda_i} \exp(-2\lambda_i S_k)$. Thus, we have

$$\mathbb{E}[\theta_{k-1,i}^2] \leq \theta_{0,i}^2 \exp(-2\lambda_i(S_1 - S_k)) + \exp(2\lambda_i S_k) \eta_{\max} \Sigma_{ii} \cdot \frac{4}{\lambda_i} \exp(-2\lambda_i S_k)$$

$$= \theta_{0,i}^2 \exp(-2\lambda_i(S_1 - S_k)) + \frac{4}{\lambda_i} \eta_{\max} \Sigma_{ii},$$

which completes the proof. $\qquad\square$

**Lemma 6.** *In the setting of Lemma 4, we can bound the sum of the error terms $\epsilon_k$ as*

$$\left| \sum_{k=1}^{T} \epsilon_k \right| \leq 5\eta_{\max} \sum_{i=1}^{d} \lambda_i^3 S_1 \exp(-2\lambda_i S_1) \theta_{0,i}^2 + \frac{15}{2} \eta_{\max}^2 \sum_{i=1}^{d} \Sigma_{ii} \lambda_i.$$

*Proof.* By the upper bound of $|\epsilon_k|$,

$$\left| \sum_{k=1}^{T} \epsilon_k \right| \leq \sum_{k=1}^{T} |\epsilon_k| \leq 5 \sum_{i=1}^{d} \left( \underbrace{\sum_{k=1}^{T} \eta_k^2 \lambda_i^3 \exp(-2\lambda_i S_k) \mathbb{E}_{\boldsymbol{\theta}_{k-1} \sim \Phi(\boldsymbol{\theta}_0, E_{\leq k-1})}[\theta_{k-1,i}^2]}_{=: \mathcal{E}_{1,i}} \right.$$

$$\left. + \underbrace{\sum_{k=1}^{T} \eta_k^3 \Sigma_{ii} \lambda_i^2 \exp(-2\lambda_i S_k)}_{=: \mathcal{E}_{2,i}} \right).$$

For $\mathcal{E}_{1,i}$, we apply Lemma 5 and have

$$\mathcal{E}_{1,i} \leq \sum_{k=1}^{T} \eta_k^2 \lambda_i^3 \exp(-2\lambda_i S_k) \left( \theta_{0,i}^2 \exp(-2\lambda_i(S_1 - S_k)) + \frac{4}{\lambda_i} \eta_{\max} \Sigma_{ii} \right)$$

$$= \lambda_i^3 \exp(-2\lambda_i S_1) \theta_{0,i}^2 \sum_{k=1}^{T} \eta_k^2 + 4\eta_{\max} \lambda_i^2 \Sigma_{ii} \sum_{k=1}^{T} \eta_k^2 \exp(-2\lambda_i S_k).$$

For the first term, we have $\sum_{k=1}^{T} \eta_k \le \eta_{\max} \sum_{k=1}^{T} \eta_k = \eta_{\max} S_1$. For the second term, by Lemma 3, we have

$$\sum_{k=1}^{T} \eta_k^2 \exp(-2\lambda_i S_k) \le \eta_{\max} \sum_{k=1}^{T} \eta_k \exp(-2\lambda_i S_k) \le \frac{\eta_{\max}}{2\lambda_i} \exp(-2\lambda_i S_{T+1}) = \frac{\eta_{\max}}{2\lambda_i}.$$

Putting these bounds together, we have

$$\mathcal{E}_{1,i} \le \eta_{\max} \lambda_i^3 S_1 \exp(-2\lambda_i S_1) \theta_{0,i}^2 + 2\eta_{\max}^2 \Sigma_{ii} \lambda_i.$$

For $\mathcal{E}_{2,i}$, we have

$$\begin{aligned}
\mathcal{E}_{2,i} = \sum_{k=1}^{T} \eta_k^3 \Sigma_{ii} \lambda_i^2 \exp(-2\lambda_i S_k) &\le \eta_{\max}^2 \Sigma_{ii} \lambda_i^2 \sum_{k=1}^{T} \eta_k \exp(-2\lambda_i S_k) \\
&\le \eta_{\max}^2 \Sigma_{ii} \lambda_i^2 \cdot \frac{1}{2\lambda_i} \exp(-2\lambda_i S_{T+1}) \\
&= \frac{1}{2} \eta_{\max}^2 \Sigma_{ii} \lambda_i,
\end{aligned}$$

where the second inequality uses Lemma 3.

Putting the upper bounds of $\mathcal{E}_{1,i}$ and $\mathcal{E}_{2,i}$ together proves the lemma. $\qquad\square$

Now we are ready to prove Theorem 3.

*Proof for Theorem 3.* According to (28), we have

$$\mathbb{E}[\mathcal{L}(\boldsymbol{\theta}_T)] = A_0 + \sum_{k=1}^{T} (A_k - A_{k-1}).$$

Using Lemma 4 and Lemma 6, we have that

$$\sum_{k=1}^{T} (A_k - A_{k-1}) = -\frac{1}{2} \sum_{k=1}^{T} (\eta_{k-1} - \eta_k) \sum_{i=1}^{d} \frac{1 - \exp(-2\lambda_i S_k)}{2} \Sigma_{ii} + \epsilon,$$

where the error bound $\epsilon$ can be bounded as

$$\epsilon \le 5\eta_{\max} \sum_{i=1}^{d} \lambda_i^3 S_1 \exp(-2\lambda_i S_1) \theta_{0,i}^2 + \frac{15}{2} \eta_{\max}^2 \sum_{i=1}^{d} \Sigma_{ii} \lambda_i.$$

Putting these together with the expression of $A_0$ leads to the results in Theorem 3. $\qquad\square$

## J.2 PROOF FOR THEOREM 1

Now we take expectation over $\boldsymbol{H}$, $\boldsymbol{\Sigma}$ and $\boldsymbol{\theta}_0$ to prove Theorem 1. Throughout the proof, we use $\gamma$ to denote the lower incomplete gamma function, $\gamma(s, x) := \int_0^x t^{s-1} e^{-t} \mathrm{d}t$, and use $\Gamma$ to denote the gamma function, $\Gamma(s) := \int_0^\infty t^{s-1} e^{-t} \mathrm{d}t$, and use $\Gamma(s, x) := \Gamma(s) - \gamma(s, x)$ to denote the upper incomplete gamma function.

We first present two lemmas on gamma functions.

**Lemma 7.** *For all $a > -1$ and $C > 0$, we have*

$$\int_0^\Lambda \lambda^a \exp(-C\lambda) \mathrm{d}\lambda = \gamma(a+1, C\Lambda) C^{-a-1}. \tag{29}$$

*Proof.* We substitute $u = C\lambda$ and have

$$\begin{aligned}
\int_0^\Lambda \lambda^a \exp(-C\lambda) \mathrm{d}\lambda &= \int_0^{C\Lambda} \left(\frac{u}{C}\right)^a \exp(-u) \frac{1}{C} \mathrm{d}u \\
&= \frac{1}{C^{a+1}} \int_0^{C\Lambda} u^a \exp(-u) \mathrm{d}u \\
&= \gamma(a+1, C\Lambda) C^{-a-1},
\end{aligned}$$

which completes the proof. $\qquad\square$

**Lemma 8.** *For all $a > -1$ and $x \geq 0$,*
$$\Gamma(a, x) \leq 2(a+1)^{(a+1)} e^{-x/2}.$$

*Proof.* For all $t \geq 0$, it holds that $t^{a-1} e^{-t} \leq (a+1)^{a+1} e^{-t/2}$. To see this, it suffices to show that $g(t) := t^{a+1} e^{-t/2} \leq (a+1)^{(a+1)}$. Note that the function $g(t)$ is increasing on $[0, 2(a+1)]$ and decreasing on $[2(a+1), +\infty)$. When $t = 2(a+1)$, we have
$$g(2(a+1)) = (2(a+1))^{a+1} e^{-(a+1)} \leq (2(a+1))^{a+1} 2^{-(a+1)} = (a+1)^{a+1}.$$

Therefore, for all $t \geq 0$, we have $g(t) \leq (a+1)^{(a+1)}$. Then, for all $x \geq 0$, we have
$$\Gamma(a, x) = \int_x^{+\infty} t^{a-1} e^{-t} \mathrm{d}t \leq \int_x^{+\infty} (a+1)^{(a+1)} e^{-t/2} \mathrm{d}t = 2(a+1)^{(a+1)} e^{-x/2},$$

which completes the proof. $\qquad\square$

Now we are ready to prove Theorem 1.

*Proof for Theorem 1.* First, we recap some definitions from Assumption 1. The distribution of $\lambda$ is given by $p(\lambda) = \frac{1}{Z} \lambda^{-\nu}$. We define $\mu := \mathbb{E}[\Sigma]$. By the definition of $\mathbb{E}[\Sigma \mid \lambda]$ in Assumption 1, $\mathbb{E}[\Sigma \mid \lambda] = F \mu \lambda^{-\rho} \exp(-r\lambda)$ for some constant $F > 0$, and $\mathbb{E}[\Delta^2 \mid \lambda] = D^2 \lambda^{-\kappa}$ for some constant $D > 0$. Furthermore, we introduce the notations $\alpha := 2 - \nu - \kappa$, and $\beta := 1 - \nu - \rho$.

By the identity $\mathbb{E}[\Sigma] = \mathbb{E}\left[\mathbb{E}[\Sigma \mid \lambda]\right]$, we have $\mu = \int_0^\Lambda F \mu \lambda^{-\rho} \exp(-r\lambda) \cdot \frac{1}{Z} \lambda^{-\nu} \mathrm{d}\lambda$, from which we can obtain $F = \frac{1}{\frac{1}{Z} \int_0^\Lambda \lambda^{-\rho-\nu} \exp(-r\lambda) \mathrm{d}\lambda} = \frac{1}{\frac{1}{Z} \gamma(\beta, r\Lambda) r^{-\beta}}$. So $F = \frac{Z r^\beta}{\gamma(\beta, r\Lambda)}$.

It suffices to prove $|\mathbb{E}[\mathcal{L}(\boldsymbol{\theta}_t)] - \widehat{\mathcal{L}}(t)| = O(S_1(t)^{-\alpha-1} + \eta_{\max}^2)$ only for $t = T$. Once we prove it for $t = T$, we can easily apply the theorem for $E_{\leq t}$, which is the original schedule truncated to the first $t$ steps, to get the result for all $1 \leq t \leq T$.

Based on the estimate of $\mathcal{L}(\boldsymbol{\theta}_T)$ given in Theorem 3, we can take the expectation over $\lambda_i, \Sigma_{ii}$ and $\theta_{0,i}$ to obtain the following bound:

$$|\mathbb{E}[\mathcal{L}(\boldsymbol{\theta}_T)] - \mathbb{E}[M(\boldsymbol{\theta}_0, E)]| \leq \underbrace{\mathbb{E}\left[5\eta_{\max} \sum_{i=1}^d \lambda_i^3 S_1 \exp(-2\lambda_i S_1)\theta_{0,i}^2\right]}_{=:Q_1} + \underbrace{\mathbb{E}\left[\frac{15}{2}\eta_{\max}^2 \sum_{i=1}^d \Sigma_{ii}\lambda_i\right]}_{=:Q_2},$$

where

$$\mathbb{E}[M(\boldsymbol{\theta}_0, E)] = \underbrace{\mathbb{E}\left[\frac{1}{2}\sum_{i=1}^d \theta_{0,i}^2 \lambda_i \exp(-2\lambda_i S_1)\right]}_{=:I_1} + \underbrace{\mathbb{E}\left[\frac{1}{2}\sum_{i=1}^d \eta_0 \Sigma_{ii} \frac{1 - \exp(-2\lambda_i S_1)}{2}\right]}_{=:I_2}$$

$$- \underbrace{\mathbb{E}\left[\frac{1}{2}\sum_{k=1}^T (\eta_{k-1} - \eta_k)\sum_{i=1}^d \Sigma_{ii} \frac{1 - \exp(-2\lambda_i S_k)}{2}\right]}_{=:I_3}.$$

In the following, we bound $Q_1, Q_2, I_1, I_2, I_3$ separately with the help of Lemma 7 and Lemma 8.

For $Q_1$, we have

$$Q_1 = 5d\eta_{\max} S_1 \mathbb{E}_p\left[\lambda^3 \exp(-2\lambda_i S_1)\Delta^2\right] = \frac{5d\eta_{\max} D^2}{Z} S_1 \int_0^\Lambda \lambda^{3-\nu-\kappa} \exp(-2\lambda_i S_1)\mathrm{d}\lambda$$

$$= \frac{5d\eta_{\max} D^2}{Z} S_1 \cdot \gamma(\alpha+2, 2S_1\Lambda)(2S_1)^{-\alpha-2}$$

$$= O(\eta_{\max} S_1^{-\alpha-1}).$$

For $Q_2$, we have

$$Q_2 = \frac{15d}{2}\eta_{\max}^2 \mathbb{E}_p[\mathcal{E}\lambda] = O(\eta_{\max}^2).$$

For $I_1$, we have

$$I_1 = \frac{d}{2}\mathbb{E}_p[\Delta^2 \lambda \exp(-2\lambda S_1)] = \frac{d}{2Z}D^2 \int_0^\Lambda \lambda^{1-\nu-\kappa} \exp(-2\lambda S_1)\mathrm{d}\lambda$$

$$= \frac{d}{2Z}D^2 \gamma(\alpha, 2S_1\Lambda)(2S_1)^{-\alpha}$$

$$= \frac{d}{2^{\alpha+1}Z}D^2 \gamma(\alpha, 2S_1\Lambda)S_1^{-\alpha}.$$

By Lemma 8, we have $\gamma(\alpha, 2S_1\Lambda) = \Gamma(\alpha) - e^{-\Omega(S_1)}$. Thus, we can rewrite $I_1$ as

$$I_1 = \frac{d \cdot \Gamma(\alpha)}{2^{\alpha+1}Z}D^2 S_1^{-\alpha} + O(e^{-\Omega(S_1)}).$$

For $I_2$, we have

$$I_2 = \frac{d}{4}\eta_0\mathbb{E}_p[\mathcal{E}] - \frac{d}{4}\eta_0\mathbb{E}_p[\mathcal{E}\exp(-2\lambda S_1)]$$

$$= \frac{d}{4}\eta_0\mu - \frac{d}{4}\eta_0 \cdot \frac{F\mu}{Z}\int_0^\Lambda \lambda^{-\rho}\exp(-r\lambda) \cdot \exp(-2\lambda S_1) \cdot \lambda^{-\nu}\,\mathrm{d}\lambda$$

$$= \frac{d}{4}\eta_0\mu - \frac{dF}{4Z}\eta_0\mu\gamma(\beta, (2S_1+r)\Lambda)(2S_1+r)^{-\beta}$$

$$= \frac{d}{4}\eta_0\mu + O(\eta_{\max}S_1^{-\beta}).$$

For $I_3$, we have

$$I_3 = \frac{d}{4}\sum_{k=1}^T (\eta_{k-1}-\eta_k)\left(\mathbb{E}_p[\mathcal{E}] - \mathbb{E}_p[\mathcal{E}\exp(-2\lambda S_k)]\right)$$

$$= \frac{d}{4}\sum_{k=1}^T (\eta_{k-1}-\eta_k)\left(\mu - \frac{F\mu}{Z}\int_0^\Lambda \lambda^{-\rho}\exp(-r\lambda)\cdot\exp(-2\lambda S_k)\cdot\lambda^{-\nu}\mathrm{d}\lambda\right)$$

$$= \frac{d}{4}\sum_{k=1}^T (\eta_{k-1}-\eta_k)\left(\mu - \frac{F\mu}{Z}\gamma(\beta, (2S_k+r)\Lambda)(2S_k+r)^{-\beta}\right).$$

Replacing $F$ with $\frac{Zr^\beta}{\gamma(\beta, r\Lambda)}$, then we have

$$I_3 = \frac{d}{4}\sum_{k=1}^T (\eta_{k-1}-\eta_k)\cdot\mu\cdot\left(1 - \frac{\gamma(\beta, (2S_k+r)\Lambda)}{\gamma(\beta, r\Lambda)}r^\beta(2S_k+r)^{-\beta}\right)$$

$$= \frac{d}{4}\mu\sum_{k=1}^T (\eta_{k-1}-\eta_k)\left(1 - \frac{\gamma(\beta, (2S_k+r)\Lambda)}{\gamma(\beta, r\Lambda)}(\frac{2}{r}S_k+1)^{-\beta}\right).$$

Setting the constants $L_0, A, \alpha, B, \beta, C$ as (19), we can summarize our results for $Q_1, Q_2, I_1, I_2, I_3$ as

$$Q_1 = O(\eta_{\max}S_1^{-\alpha-1}), \qquad Q_2 = O(\eta_{\max}^2),$$

$$I_1 = AS_1^{-\alpha} + O(e^{-\Omega(S_1)}), \qquad I_2 = L_0 + O(\eta_{\max}S_1^{-\beta}),$$

$$I_3 = B\sum_{k=1}^T (\eta_{k-1}-\eta_k)\widehat{G}(S_k),$$

where

$$\widehat{G}(x) := 1 - \frac{\gamma(\beta, (2x+r)\Lambda)}{\gamma(\beta, r\Lambda)}\cdot(Cx+1)^{-\beta}.$$

Putting everything together, we have

$$|\mathbb{E}[\mathcal{L}(\boldsymbol{\theta}_t)] - \widehat{\mathcal{L}}(t)| = \left|\mathbb{E}[\mathcal{L}(\boldsymbol{\theta}_T)] - \mathbb{E}[M(\boldsymbol{\theta}_0, E)] + O(e^{-\Omega(S_1)} + \eta_{\max}S_1^{-\beta})\right|$$

$$= O(\eta_{\max}S_1^{-\alpha-1} + \eta_{\max}^2 + e^{-\Omega(S_1)} + \eta_{\max}S_1^{-\beta})$$

$$= O(\eta_{\max}S_1^{-\min\{\alpha+1,\beta\}} + \eta_{\max}^2),$$

which completes the proof. □

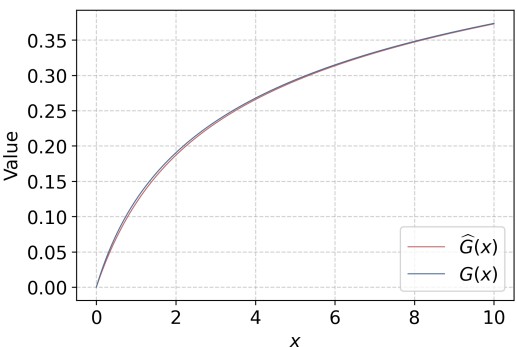

Figure 21: A comparison of $G(x)$ defined in (2) and $\widehat{G}(x)$ defined in (18). $G(x)$ follows a exact power form, while $\widehat{G}(s)$ follows a power form approximately. The gap between $G(x)$ and $\widehat{G}(x)$ converges to 0 as $x \to +\infty$. Here $\widehat{G}(x)$ is defined as in (18) with parameter $C = \frac{2}{r} = 1$, and $G(x)$ is defined as in (2) with parameters $C = (\frac{\Gamma(\beta)}{\gamma(\beta, r\Lambda)})^{-\frac{1}{\beta}}$. In both cases, we set the exponent $\beta = 0.2$.

