# OpenReview forum: "A Multi-Power Law for Loss Curve Prediction Across Learning Rate Schedules"
_ICLR.cc/2025/Conference — ICLR 2025 Poster_

### Official Review · Reviewer_ic14 · 2024-11-03

**Soundness:** 3
**Presentation:** 3
**Contribution:** 3
**Rating:** 6
**Confidence:** 3

**Summary:**

The paper presents an empirical scaling law for loss prediction while considering learning rate schedules. The formula is in a multi-power form, which includes a power law that is applied on the sum of learning rates and another power law related to learning rate decay. The rationale behind the approximation has been explained. Loss curves with both predicted and actual values on 25M, 100M, and 400M parameter models have been obtained from experiments and shown in the paper. An approach for finding an optimal learning rate scheduled based on the empirical law is also presented.

**Strengths:**

- The inclusion of learning rate schedules in scaling law.
- Experiments were conducted to study various aspects in this work.

**Weaknesses:**

- The scaling law expression in (2) is quite complicated, making it difficult to understand the intuitive meaning behind the formula.
- To apply the result, the initial learning rate still needs to be specified.
- One way of using the result, as presented in the paper, is to find the optimal learning rate schedule. However, the usefulness of such optimization in practice is not very clear. From Figure 1, the loss difference between the different learning rate schedules seems to be quite small. Only the loss is shown throughout the paper, and no benchmark evaluation result is presented, so it is not clear whether such small changes in loss would actually cause a significant difference in the model accuracy or not.
- The consideration of models with 25M, 100M, and 400M parameters, adhering to the Llama-2 architecture, may be due to limitations in the GPU resource, which is understandable. However, most practical LLMs have at least 1B parameters, so the current results are insufficient for showing that this scaling law actually works for models with practical sizes.
- Some figures, such as Figure 2, are quite hard to read and understand, due to the many curves in the figure without a clear labeling of them.

**Questions:**

- Is it possible to analyze the sensitivity of different terms in the expression (2), and try to simplify the formula while still giving a similar prediction?
- Could you add benchmark evaluation results, to see whether the loss improvement with optimized learning schedule indeed improves the accuracy?
- For figures with two y axes in the same plot, such as Figure 2, could you put each of them into two separate plots with one y axis each, to improve the readability?

---

> ### Author Response · Authors · 2024-11-23
> **Response to Reviewer ic14**
>
> Thank you for taking the time to review our paper and providing extensive feedback, which has been invaluable in improving the quality of our work. You may have noticed that we skipped some of your questions in our response; this is because these questions have been addressed collectively in **General Response**.
>
> **W1:** The scaling law expression in (2) is quite complicated, making it hard to follow the intuition.
>
> **A:**
>
> 1. We note that the **impact of LR decay on loss is inherently subtle and complex** for its influence on training speed and stability is complex and not yet well-quantified. So the somewhat intricate form of the multi-power law is to be expected. Qualitatively, a larger LR can accelerate the reduction of training loss initially but may lead to overshooting and oscillations along sharp directions in the loss landscape over time. Conversely, a smaller LR provides a more stable training process but slows down convergence, requiring more time to achieve similar results.
> 2. As clearly claimed in Section 2, the main intuition in expression (2) is that we can decompose the loss into two terms which are constant LR loss in a power law form and loss reduction term.
> 3. Regarding the derivation and intuition behind the expression in Equation (2), we have detailed the derivation process in Section 2. Through analyzing the impact of LR decay on loss across multiple cases, we derived the general formula that loss follows. The derived formula is extensively validated in Section 3,4,5.
>
> **W2:** To apply the result, the initial learning rate still needs to be specified
>
> **A:**
> 1. In most experiments, we fixed the initial learning rate in the paper for the sake of derivation simplicity. However, when studying the loss under two-stage LR schedules, we conducted experiments without fixing the initial learning rate (Section 2.2). The results demonstrate that the multi-power law holds for different initial learning rates. Furthermore, we provided a detailed analysis of how the parameters of the multi-power law vary with the initial learning rate, as shown in Figure 4.
> 2. Additionally, We add the ablation experiments on the peak learning rate in the revised version. The results show that **the multi-power law are applicable across various initial learning rate**. See Figure 16 \& Section 5, forth paragraph.
>
> **W5;Q3** Some figures with two 2 y axes is hard to read.
>
> **A:** Thanks for pointing that out. We try to fix these concerns in the revision. The main improvements are as follows.
>
> 1. Separate the twin axes in the Figure 2 (Still indexed as Figure 2). Similarly, Figure 8 \& Figure 16 are fixed in revision;
> 2. Adjust the zoom-in/out presentation, to facilitate the readers. Figure 1, 6, 7, 8, 11, 15, 17, 19, 20.
> 3. Add color bars, legends or captions for color-labeled/color-sorted curves. Figure 8, 10, 11, 13, 14.

---

> ### Author Response · Authors · 2024-11-25
> **Looking Forward to the Reviewer‘s Reply**
>
> Dear Reviewer ic14,
>
> As the deadline approaches, we would like to kindly follow up to check if our responses have adequately addressed your concerns. Your feedback would be invaluable to ensure we have fully resolved any remaining issues.
>
> In this revision, we have specifically added experiments scaling the model size, varying the model type, and incorporating downstream tasks to further support and validate our multi-power law. Additionally, we have made detailed adjustments to the presentation of figures similar to Figure 2.
>
> Thank you for your time and thoughtful feedback.

---

> > ### Comment · Reviewer_ic14 · 2024-11-25
> >
> > Thanks for your response. It seems my following questions have not been answered:
> > - Is it possible to analyze the sensitivity of different terms in the expression (2)?
> > - Could you add benchmark evaluation results, to see whether the loss improvement with optimized learning schedule indeed improves the accuracy?

---

> > > ### Author Response · Authors · 2024-11-28
> > > **Response to Reviewer ic14 [1/2]**
> > >
> > > We sincerely thank the reviewer for the timely feedback. We apologize if our previous response lacked clarity. Actually, we have already responded to the concern about benchmark evaluation results. Since several reviewers shared this common concern, we addressed it collectively in the **Downstream task evaluation of 1B Model trained with optimized LR schedule** part in **General Response**. Here, we are glad to clarify this issue for you again, including a table summarizing downstream evaluation. For your convenience, we reproduce the table below.
> > >
> > > **Q2:** Could you add benchmark evaluation results, to see whether the loss improvement with optimized learning schedule indeed improves the accuracy?
> > >
> > > **A:** **The optimized LR schedule consistently outperforms the cosine LR schedule across multiple downstream tasks**. In the experiment of the 1B model, we derived the optimized LR schedule based on its corresponding MPL. Following the evaluation criteria outlined in the Mamba paper[1], we selected the first four downstream tasks to compare the performance of the cosine and optimized LR schedules. The comparison results are shown in the table below, demonstrating that the optimized LR schedule performs better than the cosine schedule.
> > > | **Downstream Dataset** | **LAMBADA** | **HellaSwag** | **PIQA** | **ARC-E** |
> > > |-------------------------|-------------|---------------|----------|-----------|
> > > | **Cosine Schedule**     | 46.54       | 37.12         | **65.13** | 43.56     |
> > > | **Optimized Schedule**  | **48.71** ($\uparrow$ 2.17%) | **37.74** ($\uparrow$ 0.62%) | 65.07 ($\downarrow$ 0.06%) | **44.09** ($\uparrow$ 0.53%) |
> > >
> > > [1] [Mamba: Linear-Time Sequence Modeling with Selective State Spaces](https://arxiv.org/pdf/2312.00752)

---

> > > ### Author Response · Authors · 2024-11-28
> > > **Response to Reviewer ic14 [2/2]**
> > >
> > > We are sorry for not directly answering your first concern, due to time and resource limits. We now conduct extensive sensitivity checks across different terms to address your concerns. (Appendix A.1)
> > >
> > > **Q1:** Is it possible to analyze the sensitivity of different terms in the expression (2) (And try to simplify the formula while still giving a similar prediction)?
> > >
> > > **A:** The reviewer is essentially asking the following question: can we simplify the current law, or replace certain components with simpler terms to achieve similar performance? We propose and validate the following alternatives / sub-questions. These results show that our formula can be reasonably simplified, but the degree of simplification corresponds to a certain level of performance degradation.
> > >
> > > Before a detailed discussion, recap our Multi-power Law (MPL) scaling law in the form of
> > > $$\mathcal{L}(t) = L_0 + A \cdot S_1(t)^{-\alpha} - LD(t),$$ where $S_1(t) := \sum_{\tau=1}^{t} \eta_\tau$. Furthermore, we have
> > > $$LD(t) := B \sum_{k=2}^{t} (\eta_{k-1} - \eta_k) \cdot G(\eta_k^{-\gamma} S_k(t)),$$ where $S_k(t) := \sum_{\tau = k}^{t} \eta_{\tau},
> > > ~~
> > > G(x) := 1 - (C x + 1)^{-\beta}.$
> > > We conduct ablation studies as the following (sub-)questions. The results explored in the following (sub-)questions are summarized in the table below.
> > > - **subQ1:** The first naturally arising question is that **"Does the loss reduction term necessary? (essentially a one-power law, $B=0$)"**
> > >     - **A1:** We refit the MPL with the term $LD(t)=0$, corresponding to the first row in the table below. The table shows a considerable performance degradation in this case, which shows the indispensability of the term $LD(t)$.
> > >
> > > - **subQ2:** When considering the simplification of $LD(t)$, a natural starting point is to examine the sensitivity of $G(x)$. Here we pose the question that **"Can $G(x)$ be simplified as constant?"**
> > >     - **A2:** W.L.O.G., we set $G(x)$ to $1$. As shown in the table, the fitting accuracy decreases to some extent, but the fitting performance in this case is, unsurprisingly, significantly better than directly removing $LD(t)$ (second row in the following table). In fact, setting $G(x)$ to $1$ essentially means using $\sum_{k = 2}^t B (\eta_{k-1}-\eta_k)=B(\eta_1-\eta_t)$ to approximate $LD(t)$. As discussed in Appendix A.1, experiments demonstrate that $LD(t)$ can be approximated using this term across various LR schedules (Figure 9).
> > >
> > > - **subQ3:** Next, we retain $G(\eta_{k}^{-\gamma}S_k(t))$ and investigate the impact of inside power term $\eta_{k}^{-\gamma}$. **Is it necessary to introduce $\eta_{k}^{-\gamma}$?**
> > >     - **A3:** After setting $\gamma=0$, the fitting performance slightly drops as shown in the Third row in the next table. These results show that the inside power term $\eta_{k}^{-\gamma}$ helps fitting loss curves in general. One could drop this term for the sake of simplicity, however, equipping this term improves the performance without introducing much computation cost (compared to $S_k(t)$), so we keep this term in our final expression of multi-power law.
> > >
> > > - **subQ4:** **Can $\eta_k^{-\gamma}S_k(t)$ be simplified as the number of remaining steps $t-k$**, instead of the total sum of remaining LRs?
> > >     - **A4:** The results of using $x=t-k$ are shown in the fifth row. We find that using steps instead of LR sum hurt the prediction accuracy of MPL. The form of LR sum comes from our derivation of MPL in Section 2. Moreover, $x=t-k$ would imply that loss continues to decrease even when learning rates are $0$ after step $k$.
> > >
> > > - **subQ5:** Another question is **whether $G(x)$ could take forms other than the power function** specified in our formula, for example, $G(x)=(1-e^{-Cx})$.
> > >     - **A5:** The result and $G(x)=(1- e^{-Cx})$ is shown in the last second row. Note that the performance degrades even compared with the case $\gamma=0$. The results align with our observations in Section 2 that the power form fits the loss reduction of the two-stage case better than the exponential form. Moreover, one of the baselines, Momentum Law, discussed in our paper, uses a special form $G(x)=(1- e^{\log \lambda x})$ with $x=t-k+1$ and $B=\frac{B_{\text{prev}}}{1-\lambda}$, which shows degradation even compared to our variant using $x=S_k(t)$ and replacing $\lambda$ with $B$ and $C$.
> > >
> > > In conclusion, while simplifications are possible, they often result in performance degradation. Our MPL retains a balance between complexity and predictive accuracy, making it robust across different scenarios.

---

> > > > ### Author Response · Authors · 2024-11-28
> > > > **Table of Sensitivity Check of Different Terms in the Expression (2)**
> > > >
> > > > | **Alternatives**            | **$LD(t)$**                             | **$G(x)$**                        | **$x$**                     | **$R^2 \uparrow$** | **MAE $\downarrow$** | **RMSE $\downarrow$** | **PredE $\downarrow$** | **WorstE $\downarrow$** |
> > > > |------------------------------|-----------------------------------------|------------------------------------|-----------------------------|--------------------|-----------------------|------------------------|-------------------------|-------------------------|
> > > > | **B=0**                     | $0$                                     | -                                  | -                           | 0.8309             | 0.0378                | 0.0412                 | 0.0111                  | 0.0241                  |
> > > > | **$G(x)=1$** | $\eta_1-\eta_t$                    | $1$                                | -                           | 0.9797             | 0.0077                | 0.0101                 | 0.0023                  | 0.0108                  |
> > > > | **$\gamma = 0$**             | $\sum_{k=2}^t (\eta_{k-1} - \eta_k) G(x)$ | $1 - (1 + Cx)^{-\beta}$           | $S_k(t)$                    | 0.9961             | 0.0046                | 0.0053                 | 0.0014                  | 0.0041                  |
> > > > | **Multi-Power Law (Ours)**   | $\sum_{k=2}^t (\eta_{k-1} - \eta_k) G(x)$ | $1 - (1 + Cx)^{-\beta}$           | $\eta_k^{-\gamma} S_k(t)$   | **0.9975**         | **0.0039**            | **0.0046**             | **0.0012**              | **0.0040**              |
> > > > | **$x = T-k$**                | $\sum_{k=2}^t (\eta_{k-1} - \eta_k) G(x)$ | $1 - (1 + Cx)^{-\beta}$           | $T-k$                      | 0.9921             | 0.0066                | 0.0075                 | 0.0020                  | 0.0069                  |
> > > > | **$G(x) = 1 - e^{-Cx}$**     | $\sum_{k=2}^t (\eta_{k-1} - \eta_k) G(x)$ | $1 - e^{-Cx}$                     | $S_k(t)$   | 0.9934             | 0.0044                | 0.0057                 | 0.0013                  | 0.0047                  |
> > > > | **Momentum Law**             | $\sum_{k=2}^t (\eta_{k-1} - \eta_k) G(x)$ | $(1 - e^{\log \lambda x})$         | $T-k+1$                    | 0.9904             | 0.0047                | 0.0060                 | 0.0014                  | 0.0047                  |

---

> ### Author Response · Authors · 2024-12-01
>
> **Dear Reviewer `ic14`,**
>
> We sincerely thank you for your timely response and for providing valuable feedback that has greatly contributed to improving our work. We have revisited your comments and believe we have now addressed your outstanding concerns more directly.
>
> In our supplemented response, we have reorganized our explanations within the **General Response** section to provide greater clarity. Additionally, we conducted an extensive sensitivity analysis of the different terms in our multi-power law. The results demonstrate the robustness and superiority of the complete multi-power law for loss curve prediction.
>
> We deeply appreciate your professionalism and thoughtful comments. We look forward to your further feedback and hope to confirm that all your concerns have been fully resolved.
>
> Sincerely,
>
> The Authors

---

> > ### Comment · Reviewer_ic14 · 2024-12-01
> >
> > Thanks for the detailed response and analysis. I've raised my score.
> >
> > If time and resources permit, it would be nice to evaluate on some more benchmarks in addition to the four that you already have. Evaluation is generally much less resource-consuming than training. Having the results for a wide range of benchmarks and showing that the learning rate schedule can improve the performance on most of the benchmarks would make the paper stronger. You would like to select only those benchmarks where you see a reasonable improvement over random guess accuracy for both optimized and non-optimized learning rate schedules though, since the 1B model may not be able to handle the tasks for all benchmarks due to the limited model size, which is orthogonal to learning rate schedules.

---

> > > ### Author Response · Authors · 2024-12-03
> > > **Further Response to Reviewer  `ic14`**
> > >
> > > **Dear Reviewer `ic14`**,
> > >
> > > Thank you for your timely and insightful feedback. In our **general response**, we presented evaluation results on benchmarks aligned with Mamba [1] to address potential concerns about cherry-picking. In response to your request for additional benchmarks, we have **expanded our evaluation to include widely-used datasets** such as TriviaQA [2], BoolQ [3], C3 [4], RTE [5], COPA [5], MMLU [6], and CMMLU [7].
> > >
> > > As you correctly anticipated, the results on MMLU and CMMLU exhibit random patterns and do not achieve 1/4 (25%) accuracy. However, for less challenging tasks like TriviaQA, BoolQ, C3, and RTE, we observe **consistent performance improvements**. Among these, the improvement in BoolQ is particularly notable, aligning well with conjectures regarding the relationship between loss curves and emergent abilities in downstream tasks [8].
> > >
> > > | **Downstream Dataset** | **TriviaQA** | **BoolQ** | **C3** | **RTE** | **COPA** |  **MMLU** | **CMMLU** |
> > > |-------------------------|--------------|-----------|---------|----------|-----------|  --------------| --------------|
> > > | **Cosine Schedule**     | 14.30        | 54.53     | 48.44 | 52.71 | **54.0** | 23.64    | **24.94** |
> > > | **Optimized Schedule**  | **14.70** ($\uparrow$ 0.40%) | **61.41** ($\uparrow$ 6.88%) | **50.30** ($\uparrow$ 1.86%) | **53.79**($\uparrow$ 1.08%) | 53.0 ($\downarrow$ 1.0%) | **23.89** ($\uparrow$ 0.25%) | 24.84 ($\downarrow$ 0.10%) |
> > >
> > > These findings emphasize the practical benefits of our proposed multi-power law. The results demonstrate that our approach can induce a schedule that significantly improves downstream performance at a meaningful scale.
> > >
> > > Thank you again for your valuable feedback, which has helped us strengthen our work.
> > >
> > > Sincerely,
> > >
> > > The Authors
> > >
> > > [1] [Mamba: Linear-Time Sequence Modeling with Selective State Spaces](https://arxiv.org/abs/2312.00752)
> > >
> > > [2] [TriviaQA: A Large Scale Distantly Supervised Challenge Dataset for Reading Comprehension](https://arxiv.org/abs/1705.03551)
> > >
> > > [3] [BoolQ: Exploring the Surprising Difficulty of Natural Yes/No Questions](https://arxiv.org/abs/1905.10044)
> > >
> > > [4] [Investigating Prior Knowledge for Challenging Chinese Machine Reading Comprehension](https://arxiv.org/abs/1904.09679)
> > >
> > > [5] [SuperGLUE: A Stickier Benchmark for General-Purpose Language Understanding Systems](https://arxiv.org/abs/1905.00537)
> > >
> > > [6] [Measuring Massive Multitask Language Understanding](https://arxiv.org/abs/2009.03300)
> > >
> > > [7] [CMMLU: Measuring massive multitask language understanding in Chinese](https://arxiv.org/abs/2306.09212)
> > >
> > > [8] [Understanding Emergent Abilities of Language Models from the Loss Perspective](https://arxiv.org/abs/2403.15796)

---

### Official Review · Reviewer_ytrq · 2024-11-03

**Soundness:** 3
**Presentation:** 3
**Contribution:** 4
**Rating:** 8
**Confidence:** 4

**Summary:**

The authors formulate and fit an intricate multi-power law (MPL)
function to estimate a subset of learning rate schedules using only
very simple staged LR schedules, i.e., those with constant LR over
different sections of time, for its derivation. The main contribution
is the inclusion of learning rate decay in the formulation of the
scaling law using the empirically motivated MPL term. It is important
to note that the empirical law was fit with extremely few examples,
requiring only 9 training runs for strong fits on unseen data at test
time.

The method is used to propose an optimised WSD-like schedule with a
smoother transition between the constant ("stable") and decay phase.

**Strengths:**

The paper's findings and the design of the empirical law are extremely
exiting, even with limitations such as only being able to work with a
fixed maximum learning rate. Similarly, being able to extrapolate to
longer horizons is both important in practice but a testament to the
design of the MPL.

**Weaknesses:**

Despite highlighting the flexibitily of the method, it is only tested
on few unseen LR schedules.

It would be nice to quantify how well/badly extrapolation to other
model sizes works.

**Questions:**

### Please address

Page 1, line 052f:
I do not understand: the statement is made here (and in section 5.1)
that only monotonously decreasing schedules ("we restrict our
attention to the class of LR schedules that decay the LR over time")
are used. However, in sections 2 and 3 and appendix C.1, you state
that warmup was included in every experiment and taken into
consideration in practice.

Page 10, line 516f:
You state that the final learning rate for the "tuned" WSD schedule
was fixed to 1/10 of its peak. I don't think this makes for a fair
comparison, since your proposed optimized version decays to a far
lower final learning rate. This is consistent with results on learning
rate schedules in the LLM setting, which found that a decay to very
low learning rates (more specifically, zero) is best.

### Minor comments

Page 1, line 053:
Should this be "$\eta_{t} >= \eta_{t + 1}$" instead of "$\eta_{t} >=
\eta_{t - 1}$"?

Page 3, line 123:
I believe this should this be "as illustrated in Figure 2(b)", not "as
illustrated in Figure 1(b)".

Page 10, line 512f:
"So the optimization of the WSD schedule is still disturbing."
Please rephrase this sentence, it is unclear to me what you mean here.
Do you want to imply that optimizing the WSD schedule is difficult?

---

> ### Author Response · Authors · 2024-11-23
> **Response to Reviewer ytrq**
>
> Thank you for your appreciation of the multi-power law. We have added evaluations on larger model sizes, different model types, and the performance of optimized LR schedules on downstream datasets to validate the flexibility of the multi-power law. You can find these details in **General Response**.
>
> **Q1：** How do you treat warm-up stage in your derivation and experiments?
>
> **A:** We incorporate the effect of warm-up stage into the constant loss term ($\mathcal{L}_{const}=L_0+AS_1^{-\alpha}$) in our results. We add the sum of the learning rate during the warm-up phase to $S_1(t)$ to represent the learning rate accumulation at time $t$ in the formula. The loss reduction term in the formula are unchanged, except that they are computed on the learning rates after the warm-up phase. Using this adjusted formula, we fit the loss curves observed in the experiments (Appendix E.3).
>
> **Q2:** Tuning WSD to 1/10 peak learning rate is not fair. Better way is to tune WSD to 0.
>
> **A:**
> 1. **Mutil-power law outperforms WSD even when we tune the WSD ending learning rate to 0**. The optimized schedules outperform WSD variants with "zero" ending learning rates. As shown in Figure 21, we compare WSD(LD) variants with near-zero ending learning rates, the optimized schedules, and the original WSD(LD) schedules. In this experiment, the ending learning rate is set to $3 \times 10^{-7}$, which is $1/100$ of the previous setting. **Notably, a lower ending learning rate does not consistently lead to improved final loss.** For example, the final loss of the WSD schedule increases with a near-zero ending learning rate. This suggests a complex interaction between the ending learning rate and the decay function, highlighting the challenges of jointly optimizing these hyperparameters in WSD schedules. In this context, the optimized schedule demonstrates its advantage by reducing the need for extensive hyperparameter tuning in WSD variants. Results details are shown in Figure 21 (Appendix H).
> 2. Actually, we follows the conventions used in previous literatures[1, 2, 3] on cosine learning rate and WSD, we tune WSD to LR / 10 as WSD paper also didn't decay the LR to 0.
> 3. Tuning the final LR introduces an extra hyperparameter and more training cost. For using our law, we just need to run at most 3 runs to fit the parameters, requiring less tuning.
>
> ### Other Comments
> Thanks for your careful reading, and all typos in your comments have been fixed. Specifically, in line 512, as you said, we mean "optimizing the WSD schedule is difficult" since this learning rate schedule includes many hyper-parameters.
>
> [1] [Scaling Laws for Neural Language Models](https://arxiv.org/abs/2001.08361)
>
> [2] [Scaling Laws and Compute-Optimal Training Beyond Fixed Training Durations](https://arxiv.org/abs/2405.18392)
>
> [3] [Minicpm: Unveiling the potential of small language models with scalable training strategies](https://arxiv.org/abs/2404.06395)

---

### Official Review · Reviewer_H6ae · 2024-11-04

**Soundness:** 3
**Presentation:** 2
**Contribution:** 2
**Rating:** 5
**Confidence:** 3

**Summary:**

This paper introduces a multi-power law for loss curve prediction of large language models. By using power law to fit loss curve of different learning rate in each step, it can accurately predict the loss curves for unseen schedules of different shapes and horizons in different sizes of Llama-2 models. Moreover, by minimizing the predicted final loss across learning rate schedules, it is able to find a schedule that outperforms the widely-used cosine learning rate schedule.

**Strengths:**

1.	It achieves good loss curve prediction performance for 25M, 100M and 400M Llama-2 models and unseen learning rate schedules.
2.	It can also help to find a better learning rate schedule than cosine learning schedule.

**Weaknesses:**

1.	It is empirical law which is not well-proofed as well as fully verified. It would be better to include more kinds of LLMs and larger model sizes since it can benefit more for large model training and new model structure exploration.
2.	It only considers the relationship between learning rate and the loss curve. However, many other hyper-parameters will also affect the loss curve, such as model size, batch size, initialization seeds, etc. It would be better to take a comprehensive view so that the prediction could be more practical.
3.	It is interesting that the multi-power law can also be used to tune optimal learning rate schedule for each case. It would be better to give more results on different kinds of models and larger model sizes.
4.	It seems the parameters A, B, C, \alpha, \beta, \gama,and L0 need to be tuned per model (or even per combination of other hyper-parameters). The tuning process needs to collect data from different learning rates. Each optimization takes over 5 × 10^4 steps. The tuning cost is a little high.

**Questions:**

Please address the questions in weaknesses.

---

> ### Author Response · Authors · 2024-11-23
> **Response to Reviewer H6ae**
>
> We greatly appreciate the efforts you put in, and the thoughtful feedback on our work. Your comments have been invaluable in improving our paper. We skip some weakness and questions you proposed, since we have answered in the **General Response**, including **W1**, **W3**, asking about verification experiments on larger model sizes and extra model types.
>
>
> **W2:** Take other hyperparameters into consideration, e.g. model size, batch size, initial seeds, etc. to make this law more practical.
>
> **A:**
> 1. When considering the impact of LR schedule only, we have conducted very extensive experiments, including ablations over peak LR, diffenrent LR schedule types, different data size and different model sizes to support our proposed law. More discussions including larger model model size of 1B, as well as the non-monotomic schedules are ablated. Details of ablation see Section 5.
>
> 2. Due to the time and resource constraints, we are not able to thoroughly verify our law in every single possible setting. But we believe our current set of experiments have already provided strong empirical evidence for justifying our proposed law. We believe the LR-dependent data scaling law like our MPL is an important step towards general scaling laws.
>
>
> **W4:** The tuning process needs to collect data from different learning rates. Each optimization takes over $5 \times 10^4$ steps. The tuning cost is a little high.
>
> **A:**
>
> 1. The sample efficiency is indeed our advantage over the previous works. We can use $2$-$3$ loss curves to predict the whole loss curves while the previous scaling laws, like Chinchilla [1], require to collect more than $10$ curves to fit the final loss given fixed model and fixed hyperparameter setting. As shown in the left of Figure 7, MPL fitted on two 24,000-step loss curves accurately predicts the final loss at 128,000 steps (over $5$ times length) trained with a cosine schedule. MPL achieved lower extrapolation error while using only $1/4$ computational resources required for Chinchilla fitting.
> 2. We set optimization step as $5\times 10^{4}$ to ensure the convergence of the fitted parameters. The computational cost of our law fitting is much lower than anticipated. The parameter space for the MPL ($L_0$, $A$, $B$, $C$, $\alpha$, $\beta$, $\gamma$) is only 7-dimensional, and the loss computation involves predictions over training steps, typically fewer than 100k. As a result, tuning the multi-power law (MPL) is lightweight compared to standard deep learning training.
> The optimization of the learning rate (LR) schedule may seem more computationally intensive, since its parameter dimensionality corresponds to the length of the loss curve. However, this process only requires final-step loss predictions, making it highly efficient. Consequently, MPL fitting and schedule optimization rarely require GPU acceleration.
>
> [1] [Training Compute-Optimal Large Language Models](https://arxiv.org/abs/2203.15556)

---

> > ### Author Response · Authors · 2024-11-25
> > **Looking Forward to the Reviewers' Reply**
> >
> > Dear Reviewer `H6ae`,
> >
> > As the deadline draws near, we wanted to kindly follow up to see if our responses have adequately addressed the concerns you raised. We would greatly appreciate any feedback to ensure that all issues have been resolved to your satisfaction.
> >
> > In this revision, we have included additional experiments by scaling the model size, varying the model type, and incorporating downstream tasks to address your points of concern specifically.
> >
> > Thank you for your time and feedback.

---

> ### Author Response · Authors · 2024-12-01
> **Looking Forward to the Reviewer's Feedback**
>
> **Dear Reviewer `H6ae`,**
>
> We sincerely thank you for your valuable feedback, which has greatly contributed to improving our work. As the rebuttal deadline approaches, we kindly follow up to check if our responses have adequately addressed your concerns. Your feedback would be greatly appreciated to ensure all issues have been resolved to your satisfaction.
>
> In the revised version, we have incorporated additional experiments to address your concerns more thoroughly:
> - We conducted experiments with various model types, including both open-source and self-run results, and larger model sizes (e.g., a 1B model and 144B tokens) to validate the multi-power law's applicability in practice.
> - We included evaluations on downstream tasks to examine the practical benefits of law-inducing schedules.
> - We comprehensively explored hyperparameters, including peak learning rates, batch sizes, and random seeds, through extensive ablation studies.
>
> We believe these additional experiments and discussions effectively demonstrate the capacity and generalizability of the multi-power law.
>
> Thank you again for your time and thoughtful comments. We sincerely look forward to any further feedback you may have on our responses.
>
> Sincerely,
>
> The Authors

---

> ### Author Response · Authors · 2024-12-03
> **Looking Forward to the Reviewers' Reply**
>
> **Dear Reviewer `H6ae`**,
>
> We greatly appreciate your thoughtful comments. As the deadline approaches, we sincerely look forward to your response. We fully understand how busy you may be and have summarized the supplemented experiments addressing your major concerns as follows:
>
> 1. Test our multi-power law across **different model types**, including both open-source models (e.g., Olmo 7B) and self-run experiments (e.g., GPT-2);
> 2. Scaling up experiments to a **larger scale** (1B-parameter models and 144B tokens);
> 3. Assessing large-scale results using the induced schedule on **downstream tasks**;
> 4. **Ablation studies on hyperparameters**, including peak learning rates, batch sizes, and random seeds.
>
> We believe these comprehensive results address your main concerns. We hope they meet your expectations and contribute to a constructive discussion. We kindly request that you consider our detailed response in your further discussions with the AC, even if your current schedule does not allow for additional feedback.
>
> Thank you once again for your time and valuable insights.
>
> Sincerely,
>
> The Authors

---

> > ### Comment · Reviewer_H6ae · 2024-12-03
> >
> > Thank you for your detailed response to my review. It addresses some of my concerns. However, regarding to the major concerns of Q1 and Q2, the paper could be further improved. Therefore, I will keep my score unchanged.

---

> > > ### Author Response · Authors · 2024-12-04
> > > **Response to Reviewer `H6ae`**
> > >
> > > **Dear Reviewer H6ae,**
> > >
> > > We sincerely appreciate your feedback and the time and effort you have dedicated to reviewing our paper. Your comments are invaluable, and we hope this response addresses your concerns and provides further clarification about our multi-power law (MPL).
> > >
> > > ---
> > >
> > >  **Q1**: *"It is an empirical law that is neither well-proven nor fully verified. It would be better to include more types of LLMs and larger model sizes, as this could provide greater insights into large-scale model training and the exploration of new model structures."*
> > >
> > > **Response**:
> > >
> > > **A1.1**: **The MPL has been extensively validated through experiments and generalizes effectively.** In our **General Response**, we conducted experiments on larger models up to 1B parameters, matching the scale of widely used models (like Llama-3.2-1B \& OLMo-1B). We also evaluated diverse architectures like GPT-2 [1] and OLMo-7B [2]. As many modern architectures are derivatives of Llama 2 and GPT-2, we believe this establishes a solid foundation for the applicability of MPL.
> > >
> > > **A1.2**: In our rebuttal revision, we have provided a **theoretical analysis specifically for quadratic loss functions**, capturing scenarios such as linear regression. Our analysis of the loss convergence demonstrates that multi-power laws naturally arise when the Hessian and noise covariance matrices exhibit power-law eigenvalue decay. Due to space constraints, the detailed proofs can be found in **Theorem 2 and Corollary 1** in **Appendix K**. In our next revision, we will expand the discussion of these theoretical results in the main text to enhance clarity and emphasis.
> > >
> > > ---
> > >
> > > **Q2**:  *"However, many other hyperparameters, such as model size, batch size, and initialization seeds, can also affect the loss curve. A more comprehensive perspective would make the prediction more practical."*
> > >
> > > **Response**:
> > >
> > > In addition to our initial response, we would like to clarify that incorporating other hyperparameters in the scaling law is an interesting but very challenging future direction.
> > >
> > > - In previous work, like Chinchilla Law, they consider the setting as follows: (1) fix the LR schedule; (2) only predict the final loss; (3) fix all other hyperparameters, like batch size, peak learning rate, and model architecture.
> > > Actually, even for more recently proposed scaling laws, they usually only focus on one aspect of the LLM-training issues (e.g., the data-set transferability [3], precision [4]). Finding a scaling law that can capture all the hyperparameters is still very challenging in the field.
> > > - **Our MPL is already less restrictive** than prior scaling laws (e.g., Chinchilla Law) and uniquely predicts both: (1) across training schedules and (2) throughout the entire loss curve **for the first time**. Moreover, MPL demonstrates **higher sample efficiency**, achieving approximately **$1/4$ of the prediction error** with only **$1/4$ of the compute** compared to Chinchilla Law. This efficiency reduces the cost of exploring additional hyperparameter dimensions in future research.
> > > - As a necessary first step toward a general scaling law for all hyperparameters, we consider one of the most important hyperparameters in training, which is the LR schedule. Many other hyper-parameters can be connected to the LR. For example, the relationship between batch size and the learning rate is studied in [5,6]. Under certain parameterizations of LLMs, one can tune the LR to the optimal value for a fixed model, then this optimal LR can automatically transfer to larger model sizes [7,8,9]. Initialization has also been connected to the learning rate [9]. Therefore, we argue that studying the LR could bridge all hyperparameters, which motivated our work.
> > > We hope this response clarifies our contributions and adequately addresses your concerns. Thank you again for your feedback and suggestions.
> > >
> > > Sincerely,
> > >
> > > The Authors
> > >
> > > [1] [Language Models are Few-Shot Learners](https://arxiv.org/abs/2005.14165)
> > >
> > > [2] [OLMo: Accelerating the Science of Language Models](https://arxiv.org/abs/2402.00838)
> > >
> > > [3] [Loss Prediction: Scaling Laws for All Datasets](https://arxiv.org/pdf/2411.12925)
> > >
> > > [4] [Scaling Laws for Precision](https://arxiv.org/abs/2411.04330)
> > >
> > > [5] [Don't Decay the Learning Rate, Increase the Batch Size](https://arxiv.org/abs/1711.00489)
> > >
> > > [6] [Surge Phenomenon in Optimal Learning Rate and Batch Size Scaling](https://arxiv.org/abs/2405.14578)
> > >
> > > [7] [Tensor Programs V: Tuning Large Neural Networks via Zero-Shot Hyperparameter Transfer](https://arxiv.org/abs/2203.03466)
> > >
> > > [8] [DeepSeek LLM: Scaling Open-Source Language Models with Longtermism](https://arxiv.org/abs/2401.02954)
> > >
> > > [9] [Scaling Exponents Across Parameterizations and Optimizers](https://arxiv.org/pdf/2407.05872)

---

### Official Review · Reviewer_Uu5m · 2024-11-04

**Soundness:** 2
**Presentation:** 3
**Contribution:** 2
**Rating:** 5
**Confidence:** 4

**Summary:**

This paper introduces a multi-power law for predicting loss curves in LLMs. The law is empirically derived, with all experiments conducted using Llama-2 models of various sizes (25M, 100M, 400M). The proposed law predicts training loss curves under decaying learning rates across different iterations. Building on Chinchilla's scaling law, it incorporates additional terms to account for the impact of learning rate decay, which Chinchilla’s law does not explicitly address. The authors fit the multi-power law to three loss curves under distinct learning rate schedules, demonstrating that it can predict loss curves for unseen schedules, validated using WSD. Additionally, they applied the law to optimize learning rate schedules, identifying one that resembles WSD but achieves faster convergence.

**Strengths:**

1. The proposed multi-power law shows potential for predicting loss curves for unseen learning rate schedules, though it has not been sufficiently validated.

2. The proposed law could be leveraged to optimize learning rate schedules, potentially accelerating LLM training.

3. The paper provides valuable insights into pretraining dynamics, offering guidance for designing learning rate schedules that can simplify the complex task of learning rate tuning.

**Weaknesses:**

1. The theoretical justification for the proposed multi-power law is limited. The formulation of the law in Equation 2 lacks rigorous derivation or justification, raising doubts about its generalizability to unseen learning schedules.

2. The paper does not clearly state the fundamental assumptions underlying the proposed law. It is unclear whether the additional terms in Chinchilla's law (the second and third terms in Equation 2) should be model-independent, batch-size-independent, or optimizer-independent. The paper appears to assume that the impact of learning rate schedules remains independent of these factors, a point with which the reviewer may disagree.

3. The experimental section does not explore other types of learning rate schedules, such as cyclic or increasing schedules, leaving it unconvincing that the proposed law applies universally to various schedulers.

4. It remains unclear if the learning rate scheduler derived from the proposed law consistently outperforms baseline schedulers. While the paper demonstrates reduced training loss at specific iterations with the proposed scheduler, it does not clarify if this advantage holds at convergence.

5. The scope of application for the proposed law appears restrictive. The parameters of the law need to be fitted under the same task, model, batch size, and optimizer before yielding an improved learning rate schedule. It seems that the model, batch size, and optimizer must remain unchanged to apply the law effectively.

6. There is a typo on line 491, and Figure 8 lacks clarity. The curves for the baseline methods are difficult to distinguish due to missing labels.

**Questions:**

1. How does the proposed law generalize to other model types, such as high-parameter models like Llama-7B? Currently, it has only been tested on models with 25M, 100M, or 400M parameters.

2. The paper claims that the proposed approach is effective for long-horizon predictions, but the validation is limited to training curves with cosine and constant learning rate schedulers. Could you add plots supporting this claim for additional learning rate schedulers?

---

> ### Author Response · Authors · 2024-11-23
> **Major Concern Response to Reviewer Uu5m**
>
> We would like to thank the reviewer for appreciating our multi-power law and noting its application in optimizing learning rate schedules to accelerate LLM pretraining. We also appreciate the acknowledgment that our work provides valuable insights into pretraining dynamics.
>
> Through the review, we noticed that the reviewer may have misunderstood the scope of application we claimed for our multi-power law. Below we first clarify on this key point, then address the other concerns.
>
> ### Major Concern: The Scope of Application of Our Law
>
> **W2, W3, \& W5:** In Weaknesses 2, 3 and 5, the reviewer raises concerns on the scope of application for our multi-power law: **(W2)** Is it model-independent, batch-size-independent, or optimizer-independent? **(W3)** Is it applicable to cyclic or increasing schedules? **(W5)** The scope of application appears restrictive because the model, batch size, and optimizer must remain unchanged.
>
> **Response:**
>
> To clarify, as mentioned in the introduction, our primary focus is on the LR schedules that decay the LR over time. For these LR schedules, we believe that **the form** of our multi-power law can be readily applied to a wide range of practical pretraining settings, but **the parameters** of the laws do depend on training details, such as the model architecture, batch size, optimizer, etc.
>
> **The Form is Correct Across Various Settings.**
> 1. For a specific pretraining setup, after fitting the parameters in the multi-power law with a few loss curves, the law can generalize to other LR schedules with different shapes and horizons. Our law is even accurate enough to be used to optimize learning rate schedules for minimizing the final pretraining loss. See Section 3, 4.
> 3. Our experiments showed that the generalizability of MPL and superiority of optimized schedules are true for various model sizes (e.g., 1B models) and architectures (e.g., Llama-2, GPT-2 and OLMo [1]), though for each setting we need to fit the parameters separately. See Section 5.
>
> **The Parameters Do Depend on Training Details.**
> We acknowledge that the parameters in the proposed multi-power law indeed depend on other training details, such as the batch size.
> 1. Compared to the previous Chinchilla scaling law, our law is already less restrictive because it can transfer across LR schedules. Our MPL unlocks two additional  dimensions: our fitted law can predict (1) across schedules and (2) throughout the whole loss curve.  We believe the LR-dependent data scaling law is an important step towards general scaling laws.
> 2. Our law is more sample-efficient than previous methods. In fact, with additional experiments, our multi-power law can uses only two observed loss curves from constant LR schedules and cosine LR schedules to accurately predict the loss curves of various unseen LR schedules, as well as to forecast the loss for longer training horizons with high precision. As shown in Figure 7, our MPL can achieve better prediction accuracy on 128,000-step loss of the Cosine schedule, with only $1/4$ compute resource. See Section 3.2 in revision.
>
> **Response to W2&W5:** As clarified above, the form is independent while the parameters dependent on these factors. We have extended the LR-dependency and full loss curve prediction from previous work, while the previous work like Chinchilla Law fix these factors with other parameters also fixed.
>
> **Response to W3: Additional results on non-decreasing LR schedules.**
> As mentioned in our introduction (the third paragraph,line 53), we focus on the schedule that non-increases the learning rate over time.
> 1. The main rationale is as follows. In large model training, using a cosine LR schedule or the recently proposed WSD LR schedule has become common practice. The LR schedules mentioned above, which are used in practical applications, typically start with a warm-up phase followed by a non-increasing decay pattern.
> 2. Nevertheless, we have added loss prediction experiments for a cyclic LR schedule [2] and an irregular non-decreasing LR schedule as shown in Figure 6. These experiments demonstrate that **the multi-power law can effectively predict loss under these two non-decreasing LR schedules, even though our derivation is based on decay LR schedules**. This highlights the strong generalization ability of the multi-power law. See Section 3.1, third paragraph.
>
> While there is much more to uncover behind the proposed law, we would like to emphasize that **we are the first to systematically propose a scaling law adaptive to learning rate schedules**. As for other training details you mentioned, we believe they represent promising directions for future work.
>
> [1] [OLMo: Accelerating the Science of Language Models](https://arxiv.org/abs/2402.00838)
>
> [2] [Cyclical Learning Rates for Training Neural Networks](https://arxiv.org/abs/1506.01186)

---

> ### Author Response · Authors · 2024-11-23
> **Other Concern Response to Reviewer Uu5m**
>
> **W1:** The theoretical justification for the proposed multi-power law is limited. The formulation of the law in Equation 2 lacks rigorous derivation or justification，doubting its generalizability.
>
> **A:** We appreciate the reviewer's interest in scaling law theory and concerns regarding the theoretical justification of our paper.
> 1. Even though we don't have rigorous justification, we do extensive experiments, including ablations over peak LR, diffenrent LR schedule types, differnet data size and different model sizes to support our proposed law to support our proposed law. Futher, more discussions including larger model model size of 1B, as well as the non-monotomic schedules are ablated. See Section 3, Section 5.
>
> 2. It is generally very hard to prove theorems for the optimization process in deep learning under realistic assumptions. Even for linear regression, purely proving the Chinchilla scaling law in theory is already very non-trivial and needs 40+ pages [3] Nevertheless, in the rebuttal revision, we present a theoretical analysis for quadratic loss functions. We demonstrate that the multi-power law can indeed arise when the Hessian and noise covariance matrices have a power-law decay in their eigenvalues (Section 5, last paragraph \& Appendix K \& Appendix L)
>
> **W4:** The paper demonstrates reduced training loss at specific iterations with the proposed scheduler but does not clarify if this advantage holds at convergence.
>
> **A:**
>
> 1. Clarify the setting: due to the high cost of training, practitioners typically focus on minimizing the final loss within a fixed compute budget, which corresponds to fixing the total number of steps ($T$). Notably, even for models like LLaMA-2, the loss does not plateau at the end of training.
> 2. Learning rate schedules are always defined for a given $T$. It is generally impossible to extend a schedule beyond its predefined duration—for example, cosine and WSD schedules are explicitly tied to a specific $T$. In particular, when the ending LR is $0$, the training can not go on with non-positive LR.
> 3. Regarding our experiments, the data-to-model ratio in our setting is even larger than that of LLaMA-2. For instance, our 0.025B model is trained on more than 36B data tokens, compared to 2T tokens for 70B LLaMA-2.
>
> **W6:** There is a typo on line 491, and Figure 8 lacks clarity.
>
> **A:**
> 1. Thanks for pointing this out. In line 491 we mean we set the $\eta_{0}=\eta_{\max}$, in which $\eta_{\max}=3 \times 10^{-4}$.
> 2. The fixed Figure 8 is indexed as Figure 20 (right) in revision. In Figure 20 (right), the pink dash-dotted line at the top represents the WSD schedule with linear-decay (WSDLD) and the green solid line represents the corresponding loss curve. The blue dashed lines below represents the predictions of the momentum law. The orange dashed line represents the multi-power law prediction while the purple dashed line represents the loss prediction of the auxiliary process mentioned in Section 2.1 for comparison. As shown, the momentum law can fit the loss in the stable stage, but predicts a counterfactual concave curve in the decay stage, whereas the multi-power law accurately predicts the loss curve. We sincerely apologize for these confusions caused by our oversight.
>
> **Q2:** The effectiveness of the proposed approach at long horizon is only validated on cosine and constant learning rate schedulers. Could you add experiments on other learning rate schedulers?
>
> **A:** We have supplemented our study with experiments on cyclic, and an irregular piecewise linear LR schedule over a longer horizon (72,000 steps). As shown in Figure 7, **the multi-power law accurately predicts long-horizon loss across these unseen LR schedules**. See Section 3.1, third paragraph.
>
> [3] [Scaling Laws in Linear Regression: Compute, Parameters, and Data](https://arxiv.org/abs/2406.08466)

---

> > ### Author Response · Authors · 2024-11-25
> > **Looking Forward to the Reviewer's Reply**
> >
> > Dear Reviewer `Uu5m`,  As the deadline approaches, we would like to kindly follow up to check if our responses have sufficiently addressed your concerns. Your feedback would be greatly appreciated to ensure we have effectively resolved any outstanding issues.
> >
> > For this revision, we have specifically included experiments scaling the model size, varying the model type, and adding downstream tasks, and a theoretical analysis part to show how the multi-power law might arise, to respond to the points you raised.
> >
> > Thank you very much for your time and thoughtful input.

---

> ### Author Response · Authors · 2024-12-01
> **Looking Forward to the Reviewer's Feedback**
>
> **Dear Reviewer `Uu5m`,**
>
> We sincerely thank you for your insightful comments, which has greatly contributed to the improvement of our work. As the rebuttal deadline approaches, we kindly follow up to ensure that our responses have adequately addressed your concerns. Your thoughtful comments have provided invaluable guidance, and we deeply appreciate your time and effort.
>
> In the revised version, we have clarified our experimental settings and extended our evaluation. Specifically:
> - We have tested non-monotonic schedule types, such as cyclic and polyline, over longer horizons.
> - Ablation studies have been added, including scaling the model size, varying the model type, and evaluating on downstream tasks.
> - We have varied additional hyperparameters, such as batch sizes and peak learning rates, to demonstrate that the form of our multi-power law holds across diverse settings, while the parameters of the law depend on specific configurations.
> - A supplemental theoretical analysis has also been included, providing insights into how the multi-power law might arise, directly addressing the points you raised.
>
> Thank you again for your time and thoughtful input. We look forward to any additional feedback you may have and would be grateful for your further comments.
>
> Sincerely,
>
> The Authors

---

> ### Comment · Reviewer_Uu5m · 2024-12-03
>
> Thank you for your response. It definitely clears some of my concerns. However, the lack of theoretical proof for the proposed scaling law is still concerning. Comprehensively considering all these factors, I will increase my rating to 5.

---

> > ### Author Response · Authors · 2024-12-03
> > **Response to Reviewer Uu5m's Concern Regarding Theoretical Analysis**
> >
> > Thanks for your interest in theory.
> > 1. In fact, **it is generally very hard to prove theorems for the optimization process in deep learning under realistic assumptions.** Even for linear regression, rigorously demonstrating the Chinchilla scaling law in theory (for both data and model scaling) is already very non-trivial and needs 40+ pages [1]. Analyzing nonlinear models is even harder and is quite open in deep learning theory. For this reason, **most previous works have only focused on linear regression [2,3,4,5,6,7,8] or even simpler models [9,10]**.
> > 2. In our rebuttal revision, we presented a theoretical analysis specifically for quadratic loss functions, which can capture the case of linear regression. **We did a careful analysis of the loss convergence and showed that multi-power laws can indeed emerge when the Hessian and the noise covariance matrices exhibit a power-law decay in their eigenvalues**, a common assumption used in previous works [1,2,3,4,5,6]. Due to space constraints, please see Theorem 2 and Corollary 1 in Appendix K for more details. In our next revision, we commit to expanding the discussion of these results in the main text for greater clarity and emphasis.
> > 3. We acknowledge that optimizing arbitrary loss functions may not always lead to the multi-power law. The key theoretical question is: Can we identify specific properties of the loss landscape that (1) empirically hold for large language models (LLMs), and (2) theoretically induce the multi-power law? **Our theoretical results provide a valuable hint by suggesting that the multi-power law may be related to the power-law spectrum of the Hessian and noise covariance matrices.**
> > 4. While the quadratic case is indeed a simplification and does not fully capture the complexities of LLM pretraining, **our scaling law has been thoroughly validated through experiments and performs effectively beyond the quadratic case.** We argue that these results already establish a solid foundation for future theoretical investigations.
> > 5. For future work, **it would be interesting to leverage existing theoretical tools to analyze (1) deep linear nets [11], or (2) general neural nets under the assumption of river-valley loss landscape [12]**, and to understand what could make multi-power law emerge and whether there exist scaling laws that might yield an even better approximation of the pretraining loss. Given the complexity and length of these analyses, it is challenging to comprehensively cover such theoretical results within a single paper. We leave them as future works.
> >
> > [1] [Scaling Laws in Linear Regression: Compute, Parameters, and Data](https://arxiv.org/abs/2406.08466)
> >
> > [2] [Spectral bias and task-model alignment explain generalization in kernel regression and infinitely wide neural networks](https://www.nature.com/articles/s41467-021-23103-1.pdf)
> >
> > [3] [Asymptotic learning curves of kernel methods: empirical data versus teacher-student paradigm](https://iopscience.iop.org/article/10.1088/1742-5468/abc61d/pdf)
> >
> > [4] [Generalization error rates in kernel regression: The crossover from the noiseless to noisy regime](https://proceedings.neurips.cc/paper_files/paper/2021/file/543bec10c8325987595fcdc492a525f4-Paper.pdf)
> >
> > [5] [A solvable model of neural scaling laws](https://arxiv.org/pdf/2210.16859)
> >
> > [6] [Loss-to-Loss Prediction: Scaling Laws for All Datasets](https://arxiv.org/pdf/2411.12925)
> >
> > [7] [Scaling and renormalization in high-dimensional regression](https://arxiv.org/pdf/2405.00592)
> >
> > [8] [Spectrum dependent learning curves in kernel regression and wide neural networks](https://proceedings.mlr.press/v119/bordelon20a/bordelon20a.pdf)
> >
> > [9] [Learning Curve Theory](https://arxiv.org/pdf/2102.04074)
> >
> > [10] [Two Phases of Scaling Laws for Nearest Neighbor Classifiers](https://arxiv.org/abs/2308.08247)
> >
> > [11] [How Feature Learning Can Improve Neural Scaling Laws](https://arxiv.org/abs/2409.17858)
> >
> > [12] [Understanding Warmup-Stable-Decay Learning Rates: A River Valley Loss Landscape Perspective](https://arxiv.org/abs/2410.05192)

---

### Official Review · Reviewer_g7Pt · 2024-11-09

**Soundness:** 2
**Presentation:** 3
**Contribution:** 2
**Rating:** 6
**Confidence:** 3

**Summary:**

In this paper authors introduce an empirical power law that predicts the pretrianing loss for LLMs as a given training step. Method is tested across the set of learning rate schedules: constant, cosine and step decay. Using the proposed power law authors were able to find the schedule that outperforms widely used cosine learning rate schedule.

**Strengths:**

Improving the predictability of LLMs at different scales has been a very important task for many years. This paper attempts to make the projections very granular, in particular, authors propose an empirically tuned power law that allows to predict the training loss for a given model at every individual step across the set of learning rates. Within the range of tested LR schedules, authors also demonstrated that their method improves training loss compared to the popular cosine learning rate schedule.

**Weaknesses:**

I think this work has a lot of potential, but it is important to conduct additional testing and analysis of the proposed method:

1. Given that the training loss demonstrates an improvement for the provided method, it is important to understand how the gain in training loss quality effects validation loss. In case the proposed method has a lower generalization capacity, it might not be beneficial to train with it even provided the improvement in the training loss.
2. Several prior works tried to control training dynamics and make LLMs more predictable with various levels of granularity. For example, MuP paper: https://arxiv.org/abs/2203.03466 that proposes a list of adjustments in the architecture to improve predictability of the LLMs at scale. It would be very interesting to understand the benefits of the proposed method in the scenario when combined with techniques like MuP.
3. During optimization of LLMs it is usually very important to look at the batch size together with learning rate. Here is the paper that proposes how the critical batch size should be estimate assuming the perfectly tuned learning rate: https://arxiv.org/abs/1812.06162. I think it is important to understand the temperature of the training in the proposed method and how the benefits would be affected once the batch size is tuned.
4. One other thing that I struggled with was understanding what is the actionable outcome from this work. If the authors propose a new learning rate schedule derived from the power law equations, it should be rigorously tested across a set of downstream tasks, or at least authors should demonstrate improvements in terms of the validation loss and not just the training loss.
5. Finally, I would recommend authors testing their methodology against the existing scaling laws that allow us to understand benefits of this method when scaling the pretraining FLOP budget.

**Questions:**

It would be good to see the quality change on the validation or test set in addition to the training set.
If you have quality on the downstream tasks, that would be very important to look at as well for the proposed method.
From these, I would also want to see how marginal the improvements are by looking at the stddev values for the validation loss due to the randomness.

---

> ### Author Response · Authors · 2024-11-23
> **Response to Reviewer g7Pt [1/2]**
>
> We sincerely thank for your effort and thoughtful feedback on our work. Your comments and suggestions have greatly contributed to improving the quality and clarity of our paper.
>
> **W1, W4 \& Question:** It is important to understand how the gain in training loss quality effects the validation loss and downstream performance.
>
> **A:** We would like to clarify that we are considering the typical pretraining setting where each training step is taking a fresh data sample from a data stream (i.e., one-pass training).
> 1. The training is essentially done on the population distribution, so there is no generalization gap between the training and test loss.
> 2. We held out a validation set sampled from the same distribution as the training set and **use the validation loss instead of the training loss in all our plots** except that for the OLMo experiments we directly used their released curves.
> 3. we have added extensive experiments in downstream tasks and the results of our optimized LR schedule present better performance than the cosine LR schedule as shown in the General Response.
>
> **W2:**  It would be very interesting to understand the benefits of the proposed method in the scenario when combined with techniques like MuP.
>
> **A:** The reviewer is essentially asking the following question: what the difference between our approaches and the well-known MuP in the LR-tuning is, and whether the benefit of our approach can be replaced by MuP.
> - Our work is orthogonal to MuP. We compare our work with MuP as follows:
>     1. MuP enables learning rate (LR) transfer across model scales. The current practice usually only searches for the optimal value of the **initial LR** for a specific type of schedule, such as the cosine schedule.
>     2. Our method attempts to leverage the derived scaling law to understand and approximate the optimal **LR shape**.
> - Our approch of optimizing the LR schedule with multi-power law can be combined with MuP as follows: one can use our approach to search for an optimal LR schedule for small models, then may transfer the optimal schedule to a large scale without extra tuning.
>
> We would like to thank the reviewer for raising this practical question, and we believe it would be an promising future direction even though it falls outside the scope of our paper.
>
> **W3:**  It is important to understand the temperature of the training in the proposed method and how the benefits would be affected once the batch size is tuned
>
> **A:** The reviewer is essentially asking the following question: if the batch size in the training runs with baseline schedules is optimally tuned to $B'$, does our optimized schedule still benefit much?
>
> 1. If the batch size of baseline runs is changed $B'$, we could just apply the multi-power law over a few training runs with batch size $B'$ and fit the parameters in the law. Then newly-fit law could induce a new schedule.
> 2. Our experiments with different batch sizes (e.g., 128 sequence batch size is used in the experiments in Section 3, and the 1B experiments use 512 sequence batch size in Section 5) suggest that the applicability of our law does not depend on the choice of batch size.
> 3. It is noteworthy that we are optimizing the schedule while fixing the peak learning rate, and our optimized schedule decays from the peak value only in the late phase. In this case, it is believable that the optimal batch size strongly depends on the peak learning rate and very weakly depends the shape of LR decay. In the paper that the reviewer pointed out[1], they also only studied the relationship between peak LR and batch size. In this perspective, the tuning of batch size is orthogonal to our discussion. Nevertheless, we agree that scaling the batch size and LR schedules simultaneously is a promising future direction.
>
> [1] [An Empirical Model of Large-Batch Training](https://arxiv.org/abs/1812.06162)

---

> > ### Comment · Reviewer_g7Pt · 2024-12-03
> >
> > 1. Thanks for adding downstream tasks comparisons. It would be great if you can also add the std values for these tasks to understand what amount of deviation can be present due to the randomness.
> >
> > 2. Thanks for clarifying the differences between your approach and MuP. I still believe that it would be great to understand whether this method can work with me MuP as good as it works without it. This can greatly increase the usability of the proposed power law at large scale training.
> >
> > I am changing my rating to 6. It is important to battle test this method in the scenarios that are closer to the LLM trainings at scale as the future directions. When both peak learning rate and batch sizes are tuned optimal (and in case of MuP transferred automatically to a larger scale). It can significantly increase credibility of this work and allow this method to be widely adopted.

---

> > > ### Author Response · Authors · 2024-12-04
> > > **Response to Reviewer `g7Pt`**
> > >
> > > **Dear Reviewer `g7Pt`,**
> > >
> > > We sincerely thank you for acknowledging our efforts to improve and clarify our work. While we believe your suggestions are optional, we greatly value them. We will consider incorporating the standard deviation of evaluation results and using MuP in our experiments in the future revision or subsequent work.
> > >
> > > Again, thanks for your insightful suggestions and thoughtful engagement in the discussion.
> > >
> > > Sincerely,
> > >
> > > The Authors

---

> ### Author Response · Authors · 2024-11-23
> **Response to Reviewer g7Pt [2/2]**
>
> **W5:** I would recommend authors testing their methodology against the existing scaling laws that allow us to understand benefits of this method when scaling the pretraining FLOP budget.
>
> **A:** Details see Figure 7 \& Section 3.2, first paragraph in revision.
> 1. **When scaling the pretaining FLOP buget and predicting the final loss, MPL shows higher sample efficiency and higher prediction accuracy compared with Chinchilla scaling law**. As shown in the left panel of Figure 7, MPL is fitted to two 24,000-step loss curves and used to predict the final loss of a 128,000-step loss curve trained with a cosine schedule. MPL achieved lower extrapolation error while using only $1/4$ of the computational resources required for Chinchilla fitting. Second, we evaluate MPL and Chinchilla fits on the 7B OLMo curve trained with a linear schedule. As shown in the right panel of Figure 7, the MPL aligned closely with the OLMo training loss, whereas the Chinchilla fit showed significant deviations.
> 2. Our MPL extends the Chinchilla law in two dimensions: after fitting, our law can be applied to (1) different schedules and (2) the whole loss curve prediction. **To our best knowledge, we are the first to systematically propose a scaling law for the full training time loss prediction**.
>
> **Q:** How marginal the improvements are by looking at the stddev values for the validation loss due to the randomness.
>
> **A:** The variance of final loss due to the random seed is minor compared to the improvements of both prediction error and the optimized schedule.
> - To address the potential impact of randomness, we added an ablation study on randomness (See Section 5, last second paragraph). Specifically, we selected three random seeds and trained a 25M-parameter model for 24,000 steps using cosine learning rate schedules. As illustrated in Figure 20, the standard deviation of the resulting loss values is less than $0.001$, establishing a lower bound for prediction errors.
> As shown in Figure 7, the improvement of prediction accuracy from Chinchilla Laws much exceeds the randomness margin. The final losses trained with optimized schedules also surpass the baselines multiple times of this margin.
> - The improvement of such scale matters due to the following reasons:
>     1. The reduction in validation loss indicates the improvement in the downstream tasks. In our 1B experiment, validation loss improvement of the optimized schedule is about $0.013$ and deduces a $2.17$\% accuracy progress in the LAMBADA task. See Section 5, third paragraph in revision.
>     2. Although the largest model size in the experiments we ran is 1B, for models of much larger scale, reducing the validation loss by a tiny amount could even induce an emergent behavior on downstream tasks[2].
>
> [2] [Understanding Emergent Abilities of Language Models from the Loss Perspective](https://arxiv.org/abs/2403.15796)

---

> ### Author Response · Authors · 2024-11-25
> **Looking Forward to the Reviewer's Reply**
>
> Dear reviewer  `g7Pt`, As the deadline approaches, we kindly ask if our responses have adequately addressed your concerns. We would greatly appreciate your feedback to ensure we have fully resolved any outstanding issues.
>
> Notably, this revision and response include the clarification of using validation in our paper and more convincing experiments including scaling the model size, varying the model type, and adding downstream tasks to address your specific concerns. Thank you for your time and consideration.

---

> ### Author Response · Authors · 2024-12-01
> **Looking Forward to the Reviewer's Feedback**
>
> **Dear Reviewer `g7Pt`,**
>
> We sincerely thank you for your detailed and constructive feedback, which has greatly contributed to improving our work. As the rebuttal deadline approaches, we kindly seek your further input to ensure our responses have addressed your concerns. Your insights are invaluable in helping us clarify our contributions and refine our settings.
>
> In this revision, we have:
> - Clarified the use of validation loss in our methodology,
> - Conducted additional experiments to address your specific suggestions, including scaling the model size, varying the model type, and adding downstream task evaluations to verify the improvement of our method.
>
>
>
> Additionally,to directly address your query:
> > **W3:** How might the benefits of your approach be affected by batch size tuning?
>
> we performed batch size ablation experiments to validate that **the form of our multi-power law holds consistently across different batch sizes**. We compared the validation loss of our optimized schedule against commonly used schedules. As shown in the table below, our optimized schedule consistently outperforms the Cosine schedule by a significant margin and exceeds the WSD and WSDLD schedules, even without elaborate tuning. This demonstrates that **the benefits of the scheduling dimension are robust across various batch sizes.**  (Section 5, fifth paragraph & Appendix I, last paragraph)
>
> | Batch Size     | Cosine  | WSD    | WSDLD  | Optimized-schedule       |
> |----------------|---------|--------|--------|-----------|
> | 64             | 3.3783  | 3.3372 | 3.3382 | **3.2734** |
> | 128 (default)  | 3.3041  | 3.2760 | 3.2764 | **3.2687** |
> | 256            | 3.2711  | 3.2340 | 3.2334 | **3.2308** |
>
> We hope that these additional experiments and clarifications have addressed your concerns. Thank you again for your time and comments. We look forward to any further feedback you may have.
>
>
> Sincerely,
>
> The Authors

---

### Comment · Area_Chair_6M1B · 2024-11-21
**No author response yet**

Dear Submission13754 Authors,

ICLR encourages authors and reviewers to engage in asynchronous discussion up to the 26th Nov deadline. It would be good if you can post your responses to the reviews soon.

---

### Author Response · Authors · 2024-11-23
**General Response**

We sincerely thank all the reviewers for taking the time to review our paper and for engaging with the content of our work. We appreciate your interest in the multi-power law (MPL), and your insights and feedback have been crucial in helping us improve the paper. We have slightly reorganized the paper and added additional experiments based on the reviewers' suggestions.

We notice that reviewers `Uu5m`, `H6ae`, `ic14`, and `ytrq` have a strong concerning in verifying MPL on larger model sizes, while reviewers `g7Pt`, `ic14`, and `H6ae` share a common focus on evaluating the optimized LR schedule on downstream tasks. We conduct additional experiments from three perspectives—model size, model type, and downstream tasks—to provide a more comprehensive verification of MPL and address the reviewers' concerns.

**Q: Verification of Multi-Power Law in larger model size. (`Uu5m`; `H6ae`; `ic14`)**

**A: We have added experiments with 1B model size, and MPL still holds in the setting with 1B model size and induces a schedule outperforming the Cosine schedule.** In these experiments, we trained models using cosine and constant LR schedules of 24,000 steps and used the resulting loss curves to fit MPL. Then we showed the MPL could predict the longer-horizon (72,000 steps) loss curve with high accuracy. We also conduct experiments to test the final validation loss of the optimized schedule of long horizon (72,000 steps), consistently outperforming the Cosine schedules. The results demonstrate that MPL retains strong generalization capabilities for 1B models. (Figure 8 \& Section 5, third paragraph \&  Appendix I, second paragraph in revision)

**Q: Downstream task evaluation of 1B Model trained with optimized LR schedule. (`g7Pt`; `ic14`; `H6ae`)**

**A: The optimized LR schedule consistently outperforms the cosine LR schedule across multiple downstream tasks**. In the experiment of 1B model, we derived the optimized LR schedule based on its corresponding MPL. Following the evaluation criteria outlined in the Mamba paper[1], we selected the first four downstream tasks to compare the performance of the cosine and optimized LR schedules. The comparison results are shown in the table below, demonstrating that the optimized LR schedule performs better than the cosine schedule. (Table 2 \& Section 5, third paragraph \& Appendix I, second paragraph in revision)

| Downstream Dataset| LAMBADA     |HellaSwag  |PIQA       |ARC-E     |
| -----------       | ----------- |-----------|-----------|----------|
| cosine schedule            | 46.54       |37.12      |**65.13**  |43.56     |
| optimized schedule       | **48.71**($\uparrow$ 2.17%)   |**37.74**($\uparrow$ 0.62%) | 65.07($\downarrow$ 0.06%)     |**44.09**($\uparrow$ 0.53%)

**Q: Verification of Multi-Power law in Different Model Types. (`Uu5m`; `H6ae`)**

**A: MPL still holds across different model types including GPT2[2] and OLMo[3] (model size 7B)**. In the previous experiments, we used Llama2 architecture. We additionally conduct two sets of experiments to validate that the multi-power law holds across the model architecture. (Section 5, second paragraph \& Appendix I, first paragraph in revision)
1. We pretrain GPT2 models and fit the law over 24000-step Cosine and Constant schedules. Then we validate the accuracy of unseen schedules and longer horizons and the superiority of the optimized schedules. MPL outperforms other baselines, consistently match loss curves in longer horizon with unseen LR schedule as shown in Figure 21.
2. We fit the multi-power law over the open-source training curve of 7B OLMo which uses a linear decay schedule. As a comparison, we also show the fitting of vanilla Chinchilla Law, which shows significant deviations, while the MPL shows better alignment with the OLMo curve. See Figure 7, right side.

[1] [Mamba: Linear-Time Sequence Modeling with Selective State Spaces](https://arxiv.org/pdf/2312.00752)

[2] [Language Models are Few-Shot Learners](https://arxiv.org/abs/2005.14165)

[3] [OLMo: Accelerating the Science of Language Models](https://arxiv.org/abs/2402.00838)

---

### Author Response · Authors · 2024-11-25
**Summary of Revisions**

We sincerely thank all reviewers for their valuable feedback and constructive suggestions. Based on your comments, we have enhanced the paper and clarified our contributions as follows (section references correspond to the revised version):

- **Section 1**: We reorganized the introduction for clarity, explicitly defined our problem setting, and summarized our contributions in a clear, enumerated list.
- **Section 3.1**: We extended the experiments to include **complex and non-monotonic schedules** of **long horizon** such as cyclic, random polyline schedules (Figures 6). (`Uu5m`, `ytrq`)
- **Section 3.2**: We added a detailed **comparison with Chinchilla Scaling Laws**, demonstrating the superior sample efficiency and prediction accuracy of the multi-power law (MPL) compared to the Chinchilla Law (Figure 7). (`g7Pt`, `Uu5m`, `H6ae`)
- **Section 4.2**: We provided extensive validation of the **superiority of the optimized schedules**, including comparisons with zero-ending-LR variants of baseline schedules (Figures 1, 8, 17, 18, 19). To facilitate practical usage, we also introduced and evaluated approximate optimized schedules (Figures 8, 10, 21). (`g7Pt`, `ytrq`)
- **Section 5**: We validated the robustness of the MPL across different hyperparameter settings:
    - **Model Types** (Figure 7, 21): Added experiments with GPT2 and open-source 7B OLMo models. (`Uu5m`, `H6ae`)
    - **Model Size** (Figure 8): Validated the law and optimized schedule using a 1B model trained on 144B tokens. (`Uu5m`, `H6ae`, `ic14`, `ytrq`)
    - **Downstream Task Evaluation** (Table 2): Showed that 1B models trained with optimized schedules achieve improved downstream task performance compared to baseline schedules. (`g7Pt`, `H6ae`, `ic14`)
    - **Peak Learning Rate** (Figure 16): Validated the MPL's applicability with peak learning rates of $4 \times 10^{-4}$ and $6 \times 10^{-4}$.
    - **Batch Size** (Figure 22): Demonstrates that the MPL consistently holds across varying batch sizes of $64$ and $256$. (`g7Pt`, `Uu5m`, `H6ae`)
    - **Random Seed** (Figure 20): Demonstrated that random seed variations have a negligible effect on results. (`g7Pt`, `H6ae`)
    - **Theoretical Discussion**: We included additional discussion on the quadratic case to address reviewer comments. (`Uu5m`)

We believe these revisions address your concerns and strengthen the paper. Thank you for your thoughtful reviews!

---

### Author Response · Authors · 2024-12-01
**Request for Assistance in Facilitating Reviewer Feedback**

**Dear Area Chair,**

We hope this message finds you well. We sincerely seek your assistance regarding our submission. We believe we have thoroughly addressed all concerns raised by the reviewers and are eager to engage in further discussion to ensure any remaining issues are resolved.

However, we regret that we have not yet received follow-up feedback from the majority of reviewers. We would appreciate it if you could let us know how to handle this situation or assist with further feedback from reviewers.

Thank you for your time, effort, and consideration.

Best regards,

The Authors

---

> ### Comment · Area_Chair_6M1B · 2024-12-01
> **Please respond to author rebuttal**
>
> Dear Reviewers,
>
> Only 1 out of 5 reviewers have responded to the author rebuttal by the original deadline of Nov 26th. There is a new deadline which is Dec 2nd. Please respond timely as reviewer response and acknowledgement is a critical part of the ICLR review process.

---

### Meta-Review · Area_Chair_6M1B · 2024-12-17

**Metareview:**

The paper extends scaling laws to account for learning rates (LRs), with special attention paid to how LRs change over time according to a schedule. Using the new scaling laws, the paper proposes a new LR schedule that approaches recent (2024) state of the art results.

Reviewers agreed that accounting for LRs in scaling laws was a valuable problem to solve. While the initial version of the paper only presented results on 400M-parameter or smaller models (which was a major concern shared by all reviewers), the authors have since added results on 1-B and 7B-parameter models.

Because the proposed scaling laws are more complex than previous ones, multiple reviewers requested additional analyses to validate the reasoning behind these choices. Not all concerns were fully addressed; at the same time, some requests (e.g. theoretical analysis) are also atypical for scaling law papers.

Overall, the paper solves an interesting problem and presents results at adequate scale, even if the analysis is imperfect in places.

**Additional Comments On Reviewer Discussion:**

Most reviewers were concerned about the small scale of results (400M), but the authors addressed this with 1B and 7B results.

Reviewers raised diverse questions about the analysis of the proposed scaling laws, and the authors made a strong attempt to address some but not all of these questions. Given the wide scope of the questions, I think this outcome is adequate and not a cause for rejection.

---

### Decision · Program_Chairs · 2025-01-22

Accept (Poster)